# CURATE: Automatic Curriculum Learning for Reinforcement Learning Agents through Competence-Based Curriculum Policy Search

## Abstract

Due to fundamental exploration challenges without informed priors or specialized algorithms, agents may be unable to consistently receive informative rewards, leading to inefficient learning. To address these challenges, we introduce CURATE, an automatic curriculum learning algorithm for reinforcement learning agents designed for difficult target task distributions. Through "exploration by exploitation," CURATE dynamically scales the task difficulty to match the agent's current competence. By exploiting its current capabilities that were learned in easier tasks, the agent improves its exploration in more difficult tasks. Our key insight is that the performance increase in tasks that are close to those used for training is inversely proportional to their difficulty, and an agent that chooses a nearby distribution of the easiest unsolved tasks at any given time can automatically induce an easiest-to-hardest curriculum. To achieve this, CURATE conducts policy search in the task space to learn the best task distribution for training the agent. As the agent's mastery grows, the learned curriculum adapts in an approximately easiest-to-hardest and task-directed fashion, efficiently culminating in an agent that can solve the target tasks. Our experiments across three domains of varying task parameterization and dimensionality demonstrate that CURATE learns highly effective curricula, matching or exceeding prior curriculum methods in target task performance. Moreover, CURATE curricula are effective beyond solving the difficult target tasks, yielding broadly capable agents.

## 1 Introduction

The advent of reinforcement learning (RL) (Sutton & Barto, 2018; Kaelbling et al., 1996) with deep neural networks (LeCun et al., 2015; Goodfellow et al., 2016) has ushered in a promising era of impressive milestones in sequential decision making for deep RL (Mnih et al., 2013; 2015; Silver et al., 2018; Vinyals et al., 2019; OpenAI et al., 2019b;a; Li, 2018; 2023). Yet, without models, inductive biases, expert trajectories, or dense rewards, model-free deep RL algorithms are markedly sample inefficient due to fundamental challenges with exploration. Initially, the RL agent's actions are essentially random, requiring many interactions with the environment before the agent can learn useful behaviors that accrue rewards. However, agent learning can be structured through *curriculum learning* (Bengio et al., 2009), which specifies how training data should be sequenced in order to achieve two broad aims (Wang et al., 2022): to guide training (i.e., increase learning sample efficiency) and to denoise training (i.e., improve learning robustness and generalization through focus on high-confidence training data regimes). Indeed, the effectiveness of introducing concepts in an orderly, structured fashion also has support from cognitive neuroscience (Elman, 1993) and effective pedagogy such as problem-based learning and assisted discovery learning (Schmidt et al., 2007; Loyens et al., 2008; Alfieri et al., 2011; Khan et al., 2011), where knowledge arises from both learner self-discovery of innovations and timely instructor interventions.

For reinforcement learning, the advantages of improving sample efficiency, generalization, and exploration through a curriculum are generally recognized (Narvekar et al., 2020; Portelas et al., 2021; Parker-Holder et al., 2022b). Indeed, achieving the goal of *automatic curriculum learning* — automatically learning the optimal curricula for *any* domain — would have far-reaching implications for the field of reinforcement learning, leading to the *de facto* standard for training RL agents and the

significant impact it would entail. However, an automated way of selecting the optimal curriculum remains an open problem, as previous literature suggests that, in the words of Bengio et al. (2009), "some curriculum strategies work better than others." Indeed, insight from evolutionary algorithms for open-ended learning suggest that innovations and autocurricula can arise in a nonlinear, spontaneous fashion (Wang et al., 2019; 2020; Zhang et al., 2024; Faldor et al., 2025). Similarly, implicit curricula that yield zero-shot transfer and generalization can emerge from Unsupervised Environment Design (UED) (Dennis et al., 2020) and Dual Curriculum Design (DCD) (Jiang et al., 2021a) methods. However, for reformulating single-task RL as multi-task RL with a one-dimensional curriculum, it has been argued that solving tasks in an easy-to-hard fashion is optimal (Li et al., 2023), but it is unclear how this insight extends to multiple dimensions. Therefore, it is not generally obvious in what sequence the tasks should be visited for an optimal curriculum. In light of these questions, curricula are often constructed manually by human designers in an *ad hoc* fashion, leading to handcrafted curricula that are tailor-made for specific domains but do not generalize to others.

To answer these questions, we introduce CURATE (**Cur**riculum **A**gent for **T**argeted **E**xploration),[1] an automatic curriculum learning algorithm for training a model-free, on-policy reinforcement learning agent to solve a difficult target task distribution, often with sparse rewards. Our approach overcomes fundamental exploration challenges by conducting *exploration by exploitation*, as coined by Leibo et al. (2019).[2] Specifically, CURATE adapts the difficulty of the training tasks to the agent's current capabilities, or *competence*, through curriculum policy search, analogous to how children self-design curricula (Dahmani et al., 2025). Initially, the agent has not learned useful behaviors, so relatively easier tasks ensure that random exploration is (relatively more) viable. Then, as the agent's competence grows, more difficult training tasks are selected to match the current capabilities of the agent. In other words, the agent improves its ability to explore in more challenging tasks by exploiting its current capabilities that were gained from previous easier tasks. Our key insight is that tasks that are nearby those used for training will exhibit performance increases inversely proportional to their difficulty (App. H.1), and by choosing the easiest (i.e., highest return) unsolved tasks at any time, CURATE can induce an easiest-to-hardest curriculum. In this way, CURATE trains an RL agent through an approximately easiest-to-hardest progression, quickly training the agent to complete the target task distribution at the end of the curriculum. This progression can also yield greater performance more widely beyond the target task, broadening CURATE's applicability.

Our contributions are as follows. First, we introduce CURATE, an automatic curriculum learning algorithm designed for solving a difficult target task distribution that learns an approximately easiest-to-hardest sequencing of training tasks using competence-based curriculum policy search. Second, our findings across three domains that are diverse in task parameterization (discrete, continuous) and dimensionality (1D to 8D) show that learned CURATE curricula match or outperform prior curriculum baselines in training capable agents. Third, we introduce the Procgen Curriculum Suite, a version of Procgen that defines a structured task space for each game, enabling rigorous and systematic benchmarking of curriculum learning in this domain.

## 2 RELATED WORK

**Curriculum learning for reinforcement learning** As formalized by Bengio et al. (2009), curriculum learning concerns how to meaningfully organize data for training machine learning models, including those used for reinforcement learning. For comprehensive surveys in curriculum learning for RL, please refer to Narvekar et al. (2020), Portelas et al. (2021), and Parker-Holder et al. (2022b). Both Graves et al. (2017) and Matiisen et al. (2019) introduce automatic curriculum learning methods based on nonstationary multi-armed bandit algorithms based on measures of progress, e.g. learning progress (Oudeyer et al., 2007; Oudeyer & Kaplan, 2007). Wang et al. (2019; 2020) show that curricula can emerge from co-evolving environments and agents. Portelas et al. (2020) introduce ALP-GMM, a Gaussian mixture model in the parameter space of the environment. Algorithms from the Unsupervised Environment Design (Dennis et al., 2020) and Dual Curriculum Design (Jiang et al., 2021a) frameworks yield implicit curricula that emerge from unsupervised learning. Li et al. (2023) propose that, under certain assumptions with one-dimensional curricula,

---

[1]Upon acceptance, we will open-source CURATE, our codebase, and the Procgen Curriculum Suite.

[2]Leibo et al. (2019) describe "exploration by exploitation" as a form of exploration in agents that continuously adapt to exploit their abilities in non-stationary environments. In our work, our environments are not non-stationary, but the training distribution is made non-stationary by the learned curriculum policy.

solving tasks from easiest to hardest is optimal. Zhang et al. (2024) and Faldor et al. (2025) show that autocurricula can emerge when leveraging foundation model priors. Our algorithm, CURATE, is most similar to Portelas et al. (2020) and Li et al. (2023). CURATE also maintains a task distribution similar to ALP-GMM, but CURATE curricula are driven by seeking out the easiest set of unsolved tasks in the direction of the target tasks. In this way, CURATE leads to approximately easiest-to-hardest curricula that extend Li et al. (2023) to multiple dimensions.

**Solving difficult tasks using curricula and exploration methods**   The question of exploration, i.e., how to train RL agents to solve difficult tasks with sparse rewards, has overlap between methods in curriculum learning and exploration. For curriculum learning, Florensa et al. (2017) introduce a curriculum learning algorithm that modifies the agent starting state, starting from a goal state and growing in reverse. This method was generalized in BaRC (Ivanovic et al., 2019) through integrating prior knowledge, e.g., physical priors. Recently, Tao et al. (2024) combine a reverse curriculum over expert demonstrations with forward curriculum learning. For exploration, approaches can be categorized based on their methodology, e.g., counts (Burda et al., 2019; Henaff et al., 2022), curiosity (Pathak et al., 2017; 2019; Raileanu & Rocktäschel, 2020), memory (Badia et al., 2020), or information theory (Seo et al., 2021). For more information, please refer to Amin et al. (2021) and Ladosz et al. (2022) for surveys. Both curriculum learning and exploration algorithms improve training efficiency by "densifying" the reward signal for the agent. CURATE achieves this by adapting the task difficulty such that the agent receives consistently available extrinsic rewards.

**Unsupervised Environment Design and Dual Curriculum Design**   First introduced by Dennis et al. (2020), the Unsupervised Environment Design (UED) paradigm provides a framework wherein parameters of an underspecified environment are varied by a teacher (e.g., as in PAIRED (Dennis et al., 2020)) to produce distributions over environments for a student learner. Jiang et al. (2021a) unify the UED framework with prior work in replaying experiences with Prioritized Level Replay (PLR) (Jiang et al., 2021b) to form the Dual Curriculum Design (DCD) framework. In so doing, Jiang et al. (2021a) introduce REPAIRED (replay-augmented PAIRED) and an extension of PLR, Robust PLR (also stylized as PLR$^{\perp}$), in which gradient updates only occur on replayed levels. Later, Parker-Holder et al. (2022a) introduce ACCEL, an evolutionary-based algorithm that randomly mutates levels starting from environments of minimal complexity. Other UED algorithms include MAESTRO (Samvelyan et al., 2023), the work of Mediratta et al. (2023) to stabilize PAIRED, and ReMiDi (Beukman et al., 2024). Our algorithm, CURATE, can be placed within the DCD framework by functioning as a teacher that offers levels that are at the leading edge of the student's competence. However, we do not refer to CURATE as a UED algorithm, as CURATE is concerned with finding the best distribution of predefined tasks instead of the design of individual tasks.

## 3 PRELIMINARIES

### 3.1 UNDERSPECIFIED POMDPs

The agent learns within an Underspecified Partially Observable Markov Decision Processs (UPOMDP) framework (Dennis et al., 2020). The UPOMDP defines a distribution of Partially Observable Markov Decision Process (POMDP) tasks (Åström, 1965; Kaelbling et al., 1998) as determined by the selection of environment parameters. The UPOMDP is defined as follows:

$$\mathcal{M} = \langle \mathcal{A}, O, \Theta, \mathcal{S}^{\mathcal{M}}, \mathcal{T}^{\mathcal{M}}, \mathcal{I}^{\mathcal{M}}, \mathcal{R}^{\mathcal{M}}, \gamma \rangle \tag{1}$$

where $a \in \mathcal{A}$ is a set of actions, $o \in O$ is a set of observations, $\theta \in \Theta$ is a set of environment parameters, and $\gamma$ is a discount factor for future rewards. The remainder of the UPOMDP tuple is defined with respect to the chosen environment parameters $\theta$ and are thus superscripted by $\mathcal{M}$. Therefore, for the POMDP $\mathcal{M}_{\theta}$ specified by $\theta$, $s \in \mathcal{S}^{\mathcal{M}} : \mathcal{S} \times \Theta$ is a set of states from state space $\mathcal{S}$ that are not observable to the agent, $\mathcal{T}^{\mathcal{M}} : \mathcal{S} \times \mathcal{A} \times \Theta \to \mathcal{S}$ defines the transition function, $\mathcal{I}^{\mathcal{M}} : \mathcal{S} \times \Theta \to O$ is the observation (i.e., introspection) function, and $R \in \mathcal{R}^{\mathcal{M}} : \mathcal{S} \times \mathcal{A} \times \Theta \to \mathbb{R}$ is the reward function. A task is considered solved if its reward meets or exceeds a solved threshold $R_S$. Through reinforcement learning, the agent learns a policy $\pi(a|o)$ from maximizing the objective

$J(\pi)$, the expected sum of discounted rewards over trajectories $\tau$ with maximum timesteps $T$:

$$J(\pi) = \mathop{\mathbb{E}}_{\tau \sim \pi} [\sum_{i=0}^{T} \gamma^i R_i] \tag{2}$$

In principle, the UPOMDP framework allows a temporally-varying trajectory of environment parameters, but in practice, we are concerned with environment parameters that only specify the construction of the task state space $\mathcal{S}$ via the underspecified state space $\mathcal{S}^{\mathcal{M}}$. In this view, our use of UPOMDPs is similar to Contextual MDPs (Abbasi-Yadkori & Neu, 2014; Hallak et al., 2015; Modi et al., 2018). We also assume that the environment parameter space $\Theta$ is disentangled, i.e., each dimension controls a single factor of variation.

### 3.2 Curriculum learning within UPOMDPs

Our problem addresses automatic curriculum learning within a UPOMDP for solving a particular target task distribution that is initially challenging or impossible for the agent to complete. Under the assumption that the environment parameters $\Theta$ are disentangled, curriculum learning can be conducted within the *axes of generalization* of the UPOMDP's *task space*, i.e., the space of environment parameters $\Theta$. In this work, tasks are ordered by difficulty within the axes of generalization, so that increasing $\theta$ generally yields more difficult tasks. We define task difficulty in terms of the returns obtained by the agent, such that easier tasks tend to induce higher returns (up to a maximum amount) and more difficult tasks lead to lower returns (above a minimum amount). Although these assumptions on the structure of the curriculum space are strong, they permit the curriculum policy search that drives CURATE. The easiest tasks occur at $\theta_e = \min(\Theta)$, and the hardest tasks occur at $\theta_t = \max(\Theta)$, defining the parameters for the target task distribution. Therefore, the POMDP that specifies the target task distribution is $\mathcal{M}_{\theta_t}$. The time-varying sequence of tasks that arises from a curriculum learning algorithm is called a curriculum $\mathcal{C}$.

## 4 Methodology

The goal of CURATE is to automatically learn a curriculum $\mathcal{C}$ for training a control policy $\pi$ to complete a difficult target task distribution $\mathcal{M}_{\theta_t}$. To do this, CURATE conducts policy search using sample-based evaluations to determine a curriculum policy that best shapes the distribution of tasks used for training the agent. Intuitively, the curriculum policy learns a local distribution of unsolved tasks with high returns that can be sampled for training. As the agent becomes more proficient and begins solving these tasks, this distribution shifts towards more difficult unsolved tasks in the direction of the target task distribution, approximating an easiest-to-hardest curriculum that has been shown to be optimal under certain conditions (Li et al., 2023). The curriculum policy $\pi_c(\theta; \mu_\theta, \Sigma_\theta)$ is represented by a Gaussian distribution over environment parameters $\theta$ with mean $\mu_\theta$ and covariance $\Sigma_\theta$. The training procedure is summarized in Sec. 4.1. Section 4.2 describes the curriculum update step, UPDATECURRICULUM.

### 4.1 Training RL policies with curriculum learning

This section summarizes the procedure for training the RL agent as described in Alg. 1 (App. B). This training procedure is designed to close the RL training loop around the target task distribution $\mathcal{M}_{\theta_t}$, such that RL training ends with only the minimum number of frames needed to solve $\mathcal{M}_{\theta_t}$. First, the control policy $\pi$ is initialized randomly. Then, the curriculum policy $\pi_c$ is initialized by the Gaussian distribution that approximates a uniform distribution over the task space. Thereafter, the curriculum policy is updated prior to training with the (initially random) control policy $\pi$ via UP-DATECURRICULUM (Sec. 4.2). For each iteration in the training loop, tasks are sampled from the curriculum policy $\pi_c$ by first sampling environment parameters $\theta_i$, which are in turn transformed into tasks. Then, a trajectory dataset $\mathcal{D}$ is generated with mean training reward $R_\mathcal{D}$ from rollouts of $\pi$ in the sampled tasks. Next, $\pi$ is updated by the reinforcement learning algorithm by performing gradient updates of policy parameters using the dataset $\mathcal{D}$. Although any on-policy reinforcement learning algorithm could be used in principle, we use Proximal Policy Optimization (PPO) (Schulman et al., 2017) due to its favorable performance and stability with both discrete and continuous domains.

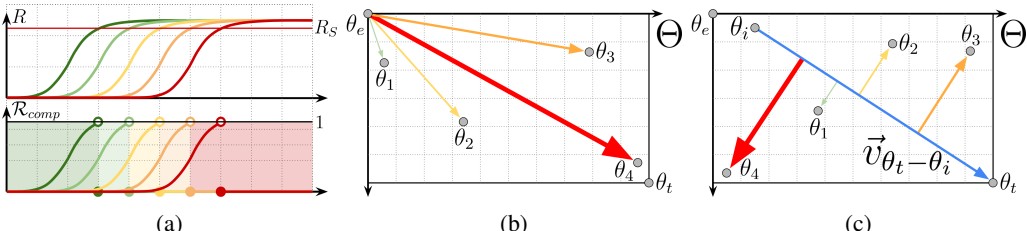

Figure 1: (a) Illustration of $\mathcal{R}_{comp}$. (b) Illustration of $\mathcal{L}_{diff}$. (c) Illustration of $\mathcal{L}_{dist}$.

Following the policy update, the curriculum policy $\pi_c$ is updated via UPDATECURRICULUM if the training reward $R_\mathcal{D}$ meets or exceeds the task solved threshold $R_S$. This trigger indicates that the agent has mastered proficiency in its current training distribution and is ready for more challenging tasks. Curriculum learning can also be triggered if a maximum number of timesteps since the last curriculum update has been reached. This prevents training stagnation if the current tasks are too difficult for the agent. Finally, the agent is evaluated on the target task distribution $\mathcal{M}_{\theta_t}$ to obtain a task evaluation reward $R_t$. We typically conduct stochastic, rather than deterministic, policy evaluation. If the agent solves the target task ($R_S \leq R_t$), then training concludes successfully. Otherwise, training continues while the number of maximum training frames has not been reached.

## 4.2 UPDATING THE CURRICULUM USING CURRICULUM POLICY SEARCH

This section summarizes UPDATECURRICULUM, the curriculum update procedure that is fully described in Alg. 2 (App. B). UPDATECURRICULUM is a nonlinear optimization within environmental parameter space to learn $\pi_c$ by optimizing the curriculum objective $J_c(\pi_c)$:

$$J_c(\pi_c) = \mathbb{E}_{\theta_j \sim \pi_c, \mathcal{M}_{\theta_j} \sim \theta_j, R_j \sim \mathcal{M}_{\theta_j}(\tau \sim \pi)}[\nu_j] \tag{3}$$

$$\nu_j = \begin{cases} \mathcal{R}_{comp} - \mathcal{L}_{diff}, & \text{initial update} \\ \mathcal{R}_{comp} - \mathcal{L}_{dist}, & \text{otherwise} \end{cases} \tag{4}$$

$$\mathcal{R}_{comp} = \frac{R_j}{R_S} \mathbb{1}_{R_j < R_S} \tag{5}$$

$$\mathcal{L}_{diff} = \lambda_\theta ||\vec{v}_{\theta_j - \theta_e}||_2 \tag{6}$$

$$\mathcal{L}_{dist} = \lambda_d ||\vec{v}_{\theta_j - \theta_i} \perp \vec{v}_{\theta_t - \theta_i}||_2 \tag{7}$$

where $\mathcal{M}_{\theta_j}$ is a task distribution sampled from $\pi_c$ via parameters $\theta_j$, $R_j$ is the reward obtained by evaluating $\pi$ on $\mathcal{M}_{\theta_j}$, $\nu_j$ is the curriculum reward, $\mathcal{R}_{comp}$ is an objective that rewards competence in unsolved tasks, $\mathcal{L}_{diff}$ is a regularization loss that penalizes difficult tasks, and $\mathcal{L}_{dist}$ is a loss that induces tasks to remain close to $\vec{v}_{\theta_t - \theta_i}$, the vector that connects the initial task distribution $\theta_i$ learned in the initial update to the target task distribution $\theta_t$. Hyperparameters $\lambda_\theta$ and $\lambda_d$ are automatically determined based on heuristics (Sec. 4.3). The major components of $J_c$ (i.e., $\mathcal{R}_{comp}$, $\mathcal{L}_{diff}$, $\mathcal{L}_{diff}$) are illustrated in Fig. 1 and discussed further. Please also see App. L for experiments that investigate the sensitivity of $R_S$.

**Regularization loss** During the initial curriculum update, a regularization loss $\mathcal{L}_{diff}$ (Eq. 6, Fig. 1b) is applied proportional to the distance of the sampled task parameters $\theta_j$ from the easiest task parameters $\theta_e$ via the vector $\vec{v}_{\theta_j - \theta_e}$. This loss addresses cases where samples consistently return zero reward (e.g., at the start of training, all tasks may be too difficult), leading to a small bias towards easier tasks. If this regularization did not exist, then the curriculum would remain close to a uniform distribution if all sampled tasks return zero reward. For all other curriculum updates beyond the initial update, this loss is not used, as it otherwise induces curriculum pressure away from the target task distribution and slows the agent's progression.

**Orthogonal distance loss** For curriculum updates except the initial update, a loss $\mathcal{L}_{dist}$ (Eq. 7, Fig. 1c) is applied to improve the progression of the agent. A loss is applied that is proportional to the magnitude of $\vec{v}_{\theta_j - \theta_i} \perp \vec{v}_{\theta_t - \theta_i}$, the vector rejection of $\vec{v}_{\theta_j - \theta_i}$ from $\vec{v}_{\theta_t - \theta_i}$, where $\vec{v}_{\theta_j - \theta_i}$ is

the vector from initial parameters $\theta_i$ to sampled parameters $\theta_j$ and $\vec{v}_{\theta_t - \theta_i}$ is the vector from initial parameters $\theta_i$ to target parameters $\theta_t$. This loss provides *task directedness*, preventing the agent from meandering through the task space by providing curriculum pressure to stay near the shortest distance between the starting point and the target tasks. Note that initial parameters $\theta_i$ are generally not the easiest task parameters $\theta_e$; $\theta_i$ is the value of the curriculum policy mean $\mu_\theta$ after the initial update and thus is the starting point of the learned curriculum. This loss is not calculated in the initial update, as $\theta_i$ is only learned at the end of the initial update. Please refer to App. G for an ablation experiment that shows $\mathcal{L}_{dist}$ is necessary for certain domains.

**Overview of method**  UPDATECURRICULUM conducts evaluations to assess the current proficiency of the agent in a sample-efficient manner without exhaustive search of the task space. First, the initial parameter distribution for the curriculum policy $\pi_c$ is provided as $(\mu_\theta, \Sigma_\theta)$. Then, for each of $N_r$ rounds, the agent draws $N_s$ parameter samples from the curriculum policy parameter distribution $(\mu_\theta, \Sigma_\theta)$. For each parameter sample $\theta_j$, the corresponding task $\mathcal{M}_{\theta_j}$ is generated, and the agent is evaluated on this task to yield reward $R_j$. However, this reward is not used directly for the curriculum learning reward $\nu_j$. Instead, it is assessed whether it meets or exceeds the threshold $R_S$, i.e., the task is solved. If so, the agent receives zero curriculum reward for this task, as the agent has mastered this task. Otherwise, the curriculum reward is first assessed as $R_j / R_S$. This reward signal $\mathcal{R}_{comp}$ induces the agent towards the easiest (i.e., highest return) tasks that have not yet been solved based on the agent's current competence (App. H.1, Fig. 12). In this way, $\mathcal{R}_{comp}$ is a mathematical characterization of the objective used by children when self-generating curricula (Dahmani et al., 2025). Thereafter, a loss is calculated to update $\nu_j$: $\mathcal{L}_{diff}$ for the initial update and $\mathcal{L}_{dist}$ otherwise. Then, the parameter samples $\theta_j$ and curriculum reward $\nu_j$ are appended to buffers. These buffers are used by Relative Entropy Policy Search (REPS) (Peters et al., 2010) to yield an updated Gaussian distribution that maximizes the curriculum reward, subject to an information loss bound based on Kullback-Leibler divergence (Kullback & Leibler, 1951). Lastly, the continuous curriculum parameters $(\mu_\theta, \Sigma_\theta)$ are discretized to yield the updated curriculum policy $\pi_c$. This process repeats iteratively $N_r$ times before returning $\pi_c$ at the conclusion of the curriculum update.

### 4.3 AUTOMATIC DETERMINATION OF HYPERPARAMETERS

To improve CURATE's utility as an automatic curriculum learning algorithm, the initial value of $(\mu_\theta, \Sigma_\theta)$ and selected hyperparameters (including $\lambda_\theta$ and $\lambda_d$) are automatically determined by heuristics according to the task space. For more information, please refer to App. B.1.

## 5 EXPERIMENTAL RESULTS

In this section, we critically analyze CURATE curricula and systematically evaluate CURATE against a variety of curriculum baselines, where performance is quantified by sample efficiency: how much training data are required to train an RL agent to achieve a certain level of performance. Specifically, we seek to answer the three following research questions. Q1: What are the characteristics of learned CURATE curricula, and how does these properties translate to sample efficiency? Q2: How does CURATE improve relative to prior curriculum methods with respect to the target tasks? Q3: Can CURATE train broadly capable RL agents beyond just the target tasks?

**Experimental domains**  Figure 2 provides an overview of the experimental domains. The domains have a rich diversity of task parameterization (discrete or continuous), task space dimensionality (1D to 8D), actions (discrete or continuous), and observations (state-based or image-based). More information about the experimental domains can be found in App. D.

*MiniGrid MultiRoom* (Chevalier-Boisvert et al., 2023) is a one-dimensional task space with sparse rewards. The agent must navigate a series of rooms to reach the goal, whereupon the agent receives a time-discounted reward (and zero reward otherwise). The specific domain is a reimplementation of MultiRoom-Random-N4 (Jiang et al., 2021b), except with the typical MiniGrid observation space of field-of-view state and agent direction. The task space is one-dimensional, where $\Theta_1 = [1, 4]$ represents the number of rooms.

The *Procgen Curriculum Suite* (PCS) contains 16 action-based games with image-based observations and a range of task space dimensionality, from one-dimensional to four-dimensional. The

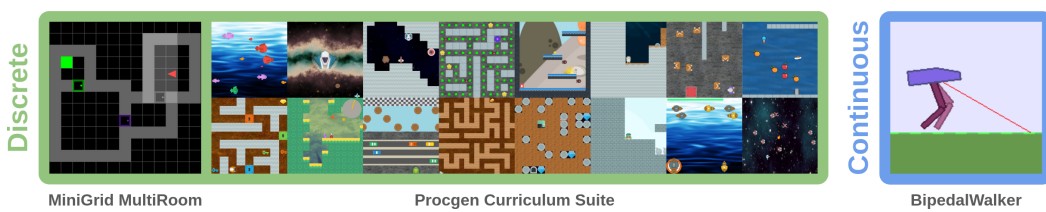

Figure 2: CURATE is evaluated on three experimental domains broadly grouped by task parameterization: discrete (MiniGrid MultiRoom, Procgen Curriculum Suite) and continuous (BipedalWalker).

games in the Procgen Curriculum Suite are adaptations of the same games first introduced by Cobbe et al. (2020) by introducing a structured task space for each game and using the associated environment parameters as inputs to the level generation. We investigate 14 of the 16 games, excluding 2 that were too difficult. Our representative game from PCS for studying Q1 is Leaper, which only offers rewards upon solving a task. However, a majority of Procgen games offer intermediate feedback, which may reduce or eliminate the need for curricula. Therefore, we "sparsify" all games in PCS that usually have intermediate feedback, such that the agent only receives a reward upon completing a task. To indicate this, games are referred to with "Terminal" appended to their names, e.g., BigFish Terminal, or BigFish-T to be concise.

*BipedalWalker* (Brockman et al., 2016) is a continuous control domain where the agent must ambulate while crossing difficult terrain. Unlike in our other two domains, BipedalWalker is a dense reward environment. However, training directly on the target domain is not meaningfully informative, so curricula are beneficial in this domain. We use the implementation of BipedalWalker used in Parker-Holder et al. (2022a). The task space is eight-dimensional, covering a range of parameters that control the terrain.

**Train and test procedure** All methods use PPO (Schulman et al., 2017) with the Adam optimizer (Kingma & Ba, 2015) for training the control policy $\pi$. The optimizer runs continuously and is not reset during training. After every control policy update, the agent is evaluated on the target task distribution $\mathcal{M}_{\theta_t}$. If the return achieved in $\mathcal{M}_{\theta_t}$ is at least $R_S$, training ends. Otherwise, training continues up to a maximum allowable frames.

**Baselines** We assess CURATE against a variety of baselines, including non-learning baselines and baselines that learn implicit curricula.

Our curriculum baselines without learning are Domain Randomization (DR) (Tobin et al., 2017), Incremental Curriculum (IC), and Target (NC). DR represents a random curriculum. IC is a structured baseline that represents a hand-designed, easiest-to-hardest curriculum that incrementally increases the task difficulty using an expert-chosen increment step size. Please refer to App. C for how IC is algorithmically generated. IC can also be considered a pseudo-oracle and an approximation of ground truth. Lastly, NC represents only training on the target tasks without a curriculum.

Our UED and DCD baselines that learn implicit curricula are Robust PLR (PLR$^\perp$) (Jiang et al., 2021b) and ACCEL Jiang et al. (2021a). PLR$^\perp$ focuses on student replay of levels, extending PLR (Jiang et al., 2021b) by only updating the agent on replayed levels. ACCEL (Jiang et al., 2021a) randomly mutates levels replayed by the student and starts from the easiest set of tasks.

## 5.1 Q1: ANALYSIS OF CURATE CURRICULA

What are the characteristics of learned CURATE curricula? To answer this question, we investigate the resulting CURATE curricula from experiments with MultiRoom and Leaper. We find that CURATE curricula 1) start in tasks that are relatively easier; 2) focus on a small, relevant distribution of tasks at a time; 3) yield an approximately easiest-to-hardest progression; and 4) remain task-directed in the presence of multidimensional task spaces. We visualize representative curricula for MultiRoom (Fig. 3a) and Leaper (Fig. 3b) for the median trial over a set of (10: MultiRoom, 6: Leaper) trials. Please refer to App. F for a comprehensive analysis.

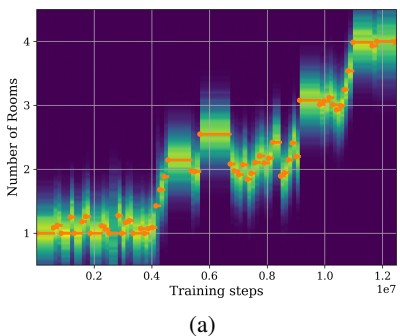 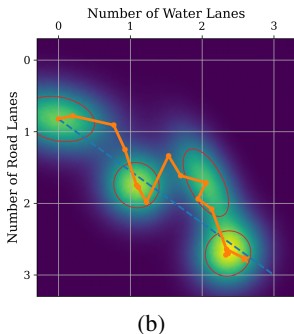

(a)                                (b)

Figure 3: Learned CURATE curricula for (a) MiniGrid MultiRoom and (b) PCS Leaper.

Table 1: Sample efficiency for MiniGrid MultiRoom in terms of student PPO updates required to solve $\mathcal{M}_{\theta_t}$ within the maximum allowable steps ($100 \times 10^6$). Summary statistics for 10 trials are shown in terms of Success Rate, Mean: mean steps with $\pm$ one standard deviation (STD), Median: median steps with $\pm$ one interquartile range (IQR), Min, and Max. Trials that do not solve the task still count towards summary statistics. Student PPO updates are $\times 10^3$. The best approaches within 1 STD/IQR are **bolded**, and the second best approaches within 1 STD/IQR are underlined.

| Method | Success Rate | Mean | Median | Min | Max |
|---|---|---|---|---|---|
| CURATE (ours) | 1.000 (10) | 1.642 $\pm$ 0.333 | 1.530 $\pm$ 0.488 | 1.162 | 2.273 |
| CURATE, $R_S = 0.3$ | 1.000 (10) | **1.130 $\pm$ 0.123** | **1.102 $\pm$ 0.143** | 0.991 | 1.433 |
| Incremental | 1.000 (10) | **1.066 $\pm$ 0.126** | **1.036 $\pm$ 0.162** | 0.856 | 1.304 |
| Domain Rand. | 1.000 (10) | 1.861 $\pm$ 0.292 | 1.805 $\pm$ 0.320 | 1.390 | 2.397 |
| Robust PLR | 1.000 (10) | 2.203 $\pm$ 0.241 | 2.131 $\pm$ 0.414 | 1.920 | 2.594 |
| ACCEL | 1.000 (10) | 2.334 $\pm$ 0.224 | 2.297 $\pm$ 0.330 | 2.026 | 2.715 |
| Target | 0.000 (10) | 12.207 $\pm$ 0.000 | 12.207 $\pm$ 0.000 | 12.207 | 12.207 |

## 5.2 Q2: EVALUATION OF CURATE AGAINST BASELINES FOR TARGET TASKS

Do the characteristics of CURATE curricula lead to beneficial performance on the target tasks as compared to other curriculum approaches? For both MultiRoom and PCS, we find that CURATE outperform naïve curricula baselines (DR, NC) and our learning baselines that yield implicit curricula (PLR$^\perp$, ACCEL). Notably, the performance improvement is significant when compared to the learning baselines, PLR$^\perp$ and ACCEL, in terms of training steps.

For MultiRoom, sample efficiency results are shown in Tab. 1, and Fig. 4 shows the learning curves for the target tasks. We find that IC is strong in this domain and outperforms a standard configuration of CURATE. However, we find that using $R_S = 0.3$ for CURATE significantly boosts performance, matching the performance of IC. We find that the primary cause for IC's performance over CURATE with $R_S = 0.7$ is the presence of a distribution shift: between tasks with 1 room and tasks with 2 rooms, the agent must now learn to open doors. The difference in evaluation return between these two distributions is significant, which leads CURATE to prefer offering 1 room tasks until these have been completely solved before offering 2 room tasks. Conversely, IC needs only to solve the task distribution of 1 rooms once before progressing to 2 room tasks. We note that these curriculum "bottlenecks" which arise in one-dimensional task spaces are opportunities to further improve CURATE's performance, such as through management of $R_S$ by using a lower value. For more figures, please see App. H.

For the Procgen Curriculum Suite, some games (BigFish-T, BossFight-T, Climber-T, Plunder-T, StarPilot-T) had target tasks that too difficult to be solvable. Therefore, we primarily conduct comparisons based on the returns in the target tasks. Furthermore, we observed that IC can experience catastrophic forgetting. Thus, for PCS, IC uses off-curriculum sampling, whereby 25% of the parallelized environments use previously solved tasks.

Figure 5 shows the composite return in the target tasks for 14 of 16 games. We visualize by training steps only, as there is a diversity of PPO batch sizes across the games. Overall, we find that CURATE

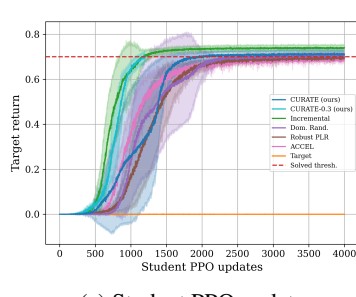 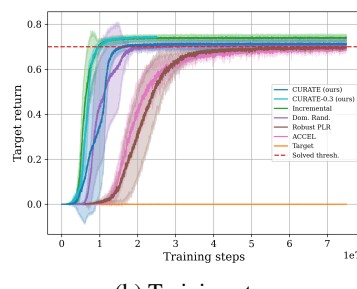

(a) Student PPO updates        (b) Training steps

Figure 4: Target return curves for MultiRoom in terms of (a) student PPO updates and (b) training steps. CURATE-0.3 refers to using CURATE, but with a lower threshold of $R_S = 0.3$, but evaluating whether the target tasks have been solved using the usual threshold of 0.7.

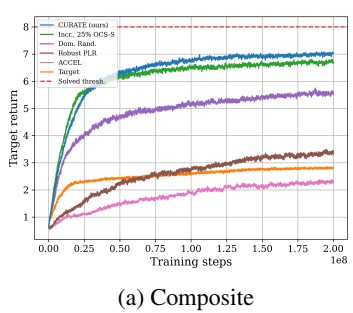 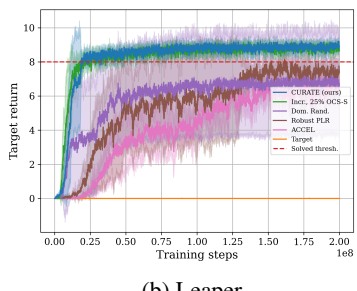

(a) Composite        (b) Leaper

Figure 5: (a) Composite return in the target tasks for 14 of the 16 games in the Procgen Curriculum Suite. For clarity, standard deviation is not shown. Chaser-T and Dodgeball-T are not included due to their difficulty, as their easiest versions yielded no training return for neither CURATE nor IC. (a) Representative target return curve for PCS Leaper.

and IC offer comparable performance that outperforms other methods. IC is slightly more sample efficient earlier in training, but CURATE offers higher-end return.

For games where the target tasks were solvable, we conduct an analysis to compare the performance of CURATE against IC in terms of sample efficiency (i.e., training steps to solve the target tasks), where the lowest-median approach is better if the other approach is greater than 1 IQR of the lowest-median approach. We find that for three games (CaveFlyer-T, Jumper, Ninja), CURATE is superior. However, for FruitBot-T, IC provides significantly greater sample efficiency. Lastly, for five games (CoinRun, Heist, Leaper, Maze, Miner-T), both approaches were comparable. Thus, we conclude overall that, although the approaches are similar, CURATE is generally more perfomant for PCS.

Lastly, we also conclude that for two games (Jumper, Maze), a curriculum is not necessary, or in some cases, detrimental. For Jumper, we find that training in the target tasks is better than either CURATE or IC, whereas for Maze, these three methods are similar. CURATE assumes that the use of a curriculum is beneficial, such that bootstrapping the solving of the target tasks by transfer from previously solved tasks is better than just training on the target tasks alone. However, an automatic curriculum learning algorithm that is truly general must work for cases where the optimal strategy is training in the target tasks. We leave this for future work.

### 5.3 Q3: BROADENING CURATE'S CAPABILITIES TO GREATER TASK DISTRIBUTIONS

Our evidence suggests that CURATE is highly capable for automatic curriculum learning for difficult, target tasks. However, in BipedalWalker, we find compelling evidence that CURATE curricula can train generally capable agents that succeed on broader task distributions than the target tasks

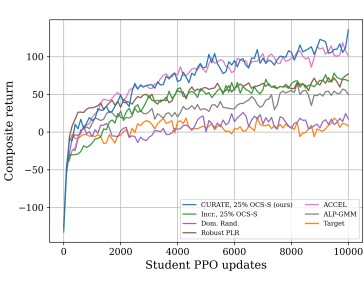
(a) Student PPO updates

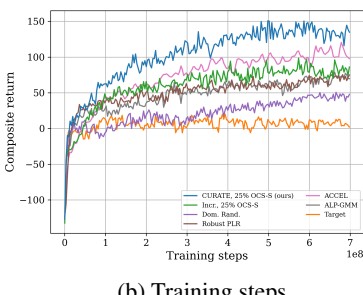
(b) Training steps

Figure 6: Composite test return curves for BipedalWalker in terms (a) student PPO updates and (b) training steps. The composite test is calculated as the average of seven test environments. Curves are calculated by time averaging over each of the 4 trials. For clarity, standard deviation is not shown.

alone. Due to the vast size of the task space, CURATE uses 25% off-curriculum sampling of previously solved tasks to boost generalization and mitigate forgetting.

Figure 6 shows the composite test return for our approaches over seven environments: Bipedal-Walker, BipedalWalkerHardcore, the four terrain challenges introduced by Parker-Holder et al. (2022a) (PitGap, Roughness, Stairs, Stump), and BipedalWalkerMax, which is our target environment with maximum terrain difficulty. Although no approaches reached positive target return, including CURATE, we find that the curricula learned by CURATE trains a generally capable agent that matches ACCEL per PPO update or exceeds it in terms of training steps. We find that CURATE can even solve the PitGap challenge, the only approach that can solve any terrain challenge. For figures of the test results, please see Sec. K.

## 6 CONCLUSION

We present CURATE, an automatic curriculum learning approach for training a model-free, on-policy reinforcement learning agent to complete a difficult target task distribution. CURATE navigates a curriculum through policy search in the task space to establish the best task distribution that matches the agent's current competence. In so doing, CURATE's "exploration by exploitation" approach addresses fundamental exploration challenges through curriculum learning. Moreover, CURATE is effective in discontinuous curriculum spaces without requiring optimal initializations or starting in the easiest tasks. Our results show that CURATE either matches or outperforms a diversity of prior curriculum methods at training agents that are not only highly performant in the difficult tasks, but in certain domains, are broadly capable for a wide range of tasks. Through this work, we hope that CURATE and the Procgen Curriculum Suite spark greater interest towards realizing a general automatic curriculum learning algorithm and the transformative impact it would entail for reinforcement learning.

**Limitations** Although CURATE offers promising performance for automatically learning curricula, it is important to note CURATE's assumptions. CURATE requires that the curriculum space is 1) defined, 2) structured in difficulty order along certain axes of task variation, and 3) available for the agent. These assumptions permit the curriculum policy search that powers CURATE. We believe that future work in addressing these assumptions would be impactful to not only CURATE, but the curriculum learning field as a whole. CURATE also assumes that task evaluations are not limited (e.g., if the target task distribution can only be attempted once). In principle, CURATE can be modified to use training returns to update the training task distribution rather than task evaluations, but we leave this for future work.

**Future work** In future work, we will explore avenues to address CURATE's assumptions. We are primarily interested in how the task space structure can be learned, rather than given as in this work. We also believe that inferring the difficulty of a task without access to the environment parameter variables would be impactful for determining the proper curriculum sequencing.

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

## A  OVERVIEW OF CURATE

Figure 7 illustrates the intuition behind CURATE.

## B  CURATE ALGORITHMS

Algorithm 1 (CURATE) describes the training procedure used to train RL agents using CURATE in this work. Algorithm 2 (UPDATECURRICULUM) describes the policy search procedure that CURATE uses to learn the curriculum policy $\pi_c$ during training.

### B.1  AUTOMATIC DETERMINATION OF CURATE HYPERPARAMETERS

To strengthen CURATE's utility as an automatic curriculum learning algorithm, six hyperparameters are automatically determined based on heuristics of the task space $\Theta$, easiest tasks $\theta_e = \min(\Theta)$, and hardest tasks $\theta_t = \max(\Theta)$.

1. The method INITIALIZERANDOMCURRICULUMPOLICY takes as input $\Theta$ and returns $\mu_\theta \leftarrow (\theta_e + \theta_t)/2$ as the center of the task space and $\Sigma_\theta$ as the Gaussian distribution that best approximates a uniform distribution over $\Theta$ via an optimization procedure. The resulting initial $(\mu_\theta, \Sigma_\theta)$ is then used for the initial CURATE curriculum update.

2. The curriculum advancement on solve $\Delta\mu_\theta$ hyperparameter is calculated as $\Delta\mu_\theta \leftarrow (\theta_t - \theta_e)/10$. This hyperparameter provides a slight bias towards progression after the training tasks are solved.

3. The curriculum covariance for update $\Sigma_{\theta_u}$ hyperparameter controls how broad the initial distribution is at the beginning of the curriculum update (for all updates except the first). It is calculated as $\Sigma_{\theta_u} \leftarrow \mathrm{diag}(((\theta_t - \theta_e)/5)^2)$.

---

**Algorithm 1:** CURATE: CURRICULUM AGENT FOR TARGETED EXPLORATION

---

**Input:** target task $\mathcal{M}_{\theta_t}$, task solved threshold $R_S$, maximum number of training frames $f_{max}$,
      number of parallelized workers $N_v$, curriculum advancement on solve $\Delta\mu_\theta$, curriculum
      covariance for update $\Sigma_{\theta_u}$, maximum frames between curriculum updates $\Delta f_{sync}$

**Initialize:** training indicator $train \leftarrow$ True, target task solved indicator
  $converged \leftarrow$ False, number of training frames $f \leftarrow 0$, control policy
  $\pi \leftarrow$ INITIALIZERANDOMPOLICY(), curriculum policy and parameters
  $\pi_c, \mu_\theta, \Sigma_\theta \leftarrow$ INITIALIZERANDOMCURRICULUMPOLICY(), previous curriculum update
  frame $f_{prev} \leftarrow 0$

---

// Initial curriculum policy update
$\pi_c, \mu_\theta, \Sigma_\theta \leftarrow$ UPDATECURRICULUM($\pi_c, \mu_\theta, \Sigma_\theta, \pi, use\_diff \leftarrow$ True)
$\theta_i \leftarrow \mu_\theta$
**while** $train$ **do**
  | // Sample tasks from the curriculum
  | $\mathcal{M}_{\boldsymbol{\theta_c}} \leftarrow \emptyset$
  | **for** $i = 1$ **to** $N_v$ **do**
  |   | $\theta_i \sim \pi_c$
  |   | $\mathcal{M}_{\theta_i} \leftarrow$ TASKGENERATOR($\theta_i$)
  |   | $\mathcal{M}_{\boldsymbol{\theta_c}} \xleftarrow{+} \mathcal{M}_{\theta_i}$
  | **end**
  | // Collect experience
  | $\mathcal{D}, R_\mathcal{D} \leftarrow$ ROLLOUTAGENTONPARALLELTASKS($\pi, \mathcal{M}_{\boldsymbol{\theta_c}}$)
  | // Update policy
  | $\pi \leftarrow$ UPDATEAGENT($\pi, \mathcal{D}$)
  | $f = f +$ NUMFRAMES($\mathcal{D}$)
  | // Update curriculum policy
  | **if** $R_S \leq R_\mathcal{D}$ **then**
  |   | $\pi_c, \mu_\theta, \Sigma_\theta \leftarrow$ UPDATECURRICULUM($\pi_c, \mu_\theta + \Delta\mu_\theta, \Sigma_{\theta_u}, \pi, use\_dist \leftarrow$ True$, \theta_i$)
  |   | $f_{prev} \leftarrow f$
  | **else if** $\Delta f_{sync} \leq (f - f_{prev})$ **then**
  |   | $\pi_c, \mu_\theta, \Sigma_\theta \leftarrow$ UPDATECURRICULUM($\pi_c, \mu_\theta, \Sigma_{\theta_u}, \pi, use\_dist \leftarrow$ True$, \theta_i$)
  |   | $f_{prev} \leftarrow f$
  | // Evaluate agent on target task
  | $R_t \leftarrow$ EVALUATEAGENT($\pi, \mathcal{M}_{\theta_t}$)
  | // Determine whether to continue training
  | **if** $R_S \leq R_t$ **then**
  |   | $train \leftarrow$ False
  |   | $converged \leftarrow$ True
  | **if** $f_{max} \leq f$ **then**
  |   | $train \leftarrow$ False
**end**

---

**Result:** control policy $\pi$, target task solved indicator $converged$, number of frames $f$

---

---

**Algorithm 2:** UPDATECURRICULUM: Curriculum update for CURATE

---

**Input:** curriculum policy $\pi_c$, initial curriculum policy mean $\mu_{\theta_0}$, initial curriculum policy covariance $\Sigma_{\theta_0}$, control policy $\pi$, task solved threshold $R_S$, easiest environment parameters $\theta_e$, target environment parameters $\theta_t$, regularization hyperparameter $\lambda_\theta$, orthogonal distance loss hyperparameter $\lambda_d$, number of rounds $N_r$, samples per round $N_s$, relative entropy bound $\epsilon$, minimum temperature $\eta$, enable regularization loss $use\_diff \leftarrow \texttt{False}$, enable orthogonal distance loss $use\_dist \leftarrow \texttt{False}$, initial environment parameters $\theta_i \leftarrow \texttt{None}$

**Initialize:** $\mu_\theta \leftarrow \mu_{\theta_0}, \Sigma_\theta \leftarrow \Sigma_{\theta_0}$

---

**for** $i = 1$ **to** $N_r$ **do**
    // Reset buffers
    $\boldsymbol{\theta_{eval}} \leftarrow \emptyset$
    $\boldsymbol{\nu_{eval}} \leftarrow \emptyset$
    **for** $j = 1$ **to** $N_s$ **do**
        // Sample task
        $\theta_j \sim \mathcal{N}(\mu_\theta, \Sigma_\theta)$
        $\mathcal{M}_{\theta_j} \leftarrow$ TASKGENERATOR$(\theta_j)$
        // Evaluate agent on sampled task
        $R_j \leftarrow$ EVALUATEAGENT$(\pi, \mathcal{M}_{\theta_j})$
        // Calculate curriculum reward based on competence
        **if** $R_j < R_S$ **then**
            $\nu_j \leftarrow R_j/R_S$
        **else**
            $\nu_j \leftarrow 0$
        // Calculate regularization loss, if applicable
        **if** $use\_diff$ **then**
            $\vec{v}_{\theta_j - \theta_e} \leftarrow \theta_j - \theta_e$
            $\nu_j \leftarrow \nu_j - \lambda_\theta ||\vec{v}_{\theta_j - \theta_e}||_2$
        // Calculate orthogonal distance loss, if applicable
        **if** $use\_dist$ **then**
            $\vec{v}_{\theta_j - \theta_i} \leftarrow \theta_j - \theta_i$
            $\vec{v}_{\theta_t - \theta_i} \leftarrow \theta_t - \theta_i$
            $\nu_j \leftarrow \nu_j - \lambda_d ||\vec{v}_{\theta_j - \theta_i} \perp \vec{v}_{\theta_t - \theta_i}||_2$
        // Append to buffers
        $\boldsymbol{\theta_{eval}} \overset{+}{\leftarrow} \theta_j$
        $\boldsymbol{\nu_{eval}} \overset{+}{\leftarrow} \nu_j$
    **end**
    // Run REPS and update curriculum policy
    $\mu_\theta, \Sigma_\theta \leftarrow$ REPSUPDATE$(\boldsymbol{\theta_{eval}}, \boldsymbol{\nu_{eval}}, \epsilon, \eta)$
    $\pi_c \leftarrow$ DISCRETIZEGAUSSIAN$(\mu_\theta, \Sigma_\theta)$
**end**

---

**Result:** updated curriculum policy $\pi_c$, updated curriculum policy mean $\mu_\theta$, updated curriculum policy covariance $\Sigma_\theta$

---

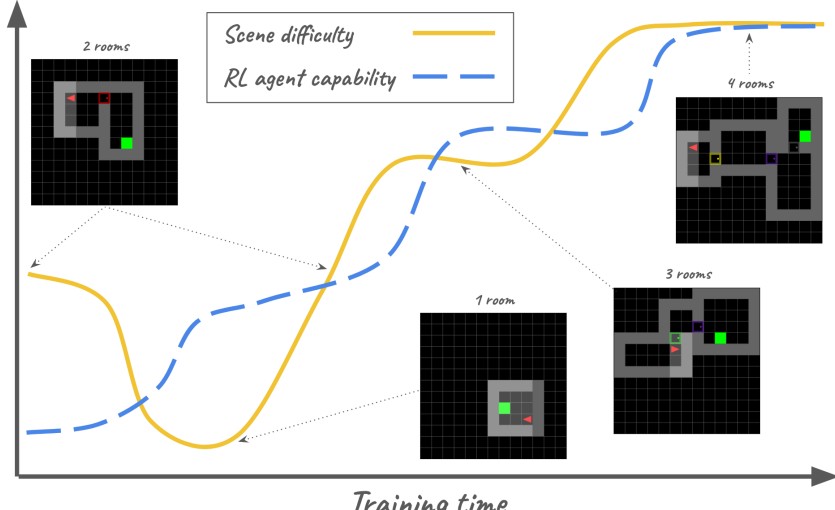

Figure 7: The CURATE algorithm automatically learns a curriculum for training an RL agent to complete a target task distribution that is initially too difficult for the agent. CURATE sequences the RL agent's training data by altering the difficulty of the training task distribution. The RL agent's current capability, or competence, is a measure of its performance in relatively more difficult tasks. In this visualization, the tasks offered by CURATE are initially too difficult, leading to a simplification of tasks. Once the RL agent begins solving these simple tasks, CURATE dynamically adjusts the training data accordingly to offer harder tasks. Finally, the agent solves the target task distribution at the end, indicating that training can conclude. Scenes are from the MiniGrid MultiRoom domain (Sec. 5).

4. The regularization hyperparameter $\lambda_\theta$ controls how much to penalize tasks based on difficulty in the initial curriculum update. The calculation is $\lambda_\theta \leftarrow 0.1||\vec{v}_{\theta_t-\theta_e}||_2$, where $\vec{v}_{\theta_t-\theta_e} \leftarrow \theta_t - \theta_e$. This means that the penalty on the curriculum reward will be no worse than 0.1.

5. The orthogonal distance hyperparameter $\lambda_d$ controls how tightly the curriculum should follow a straight line. Let $\theta_d$ be the environment parameters that have the largest distance from $\vec{v}_{\theta_t-\theta_e}$. Then, $\lambda_d \leftarrow 0.5||\vec{v}_{\theta_d-\theta_e} \perp \vec{v}_{\theta_t-\theta_e}||_2$.

## C  INCREMENTAL CURRICULUM ALGORITHM

The incremental curriculum baseline in Sec. 5 approximates a handcrafted curriculum that a domain expert may design. Specifically, the incremental curriculum sequentially visits tasks in increasing order of difficulty, approximating a straight line that forms the shortest path curriculum through the curriculum space. (Note that the incremental curriculum does not directly traverse the shortest path curriculum, as the incremental curriculum only increments one dimension of the environment parameters at a time to prevent training instability.) We generate the incremental curriculum algorithmically using Alg. 3 (GENERATEINCREMENTALCURRICULUM), given the environment parameters of the easiest task $\theta_e$ and the environment parameters of the target task $\theta_t$. Figure 8 shows the incremental curriculum that was used for Procgen Curriculum Suite Leaper. The incremental curriculum for MiniGrid MultiRoom is $\{1, 2, 3, 4\}$.

Algorithm 3 yields the lowest error approximation of a straight line through the curriculum subject to sequential, cyclic increments of each environment parameter, starting from the easiest task parameters $\theta_e$ until the target task parameters $\theta_t$ are reached. Intuitively, each dimension of the environment parameters is incremented by a multiple of the respective dimension of $\Delta_\theta$ (which for our experiments is always one). This multiple is the minimum multiple that yields a curriculum point $\theta'$ that has dimension $d$ greater than the same dimension of curriculum point $\theta'_{SP}$, which is the projection onto the shortest path curriculum in the direction of the increment. An incremental cur-

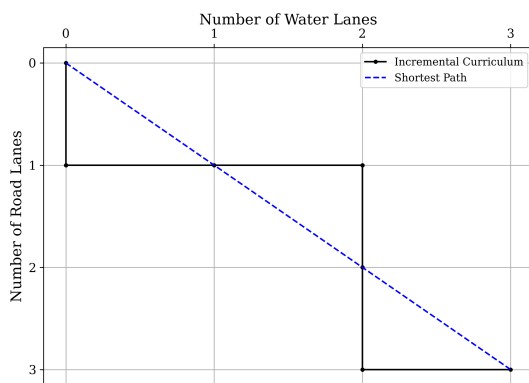

Figure 8: Incremental curriculum for Procgen Curriculum Suite Leaper.

riculum is generated for each possible starting increment of the first dimension, then the incremental curriculum with the least error is returned.

We note that proper selection of $\Delta_\theta$ is key to a well-performing incremental curriculum. If the increment is too low, learning will be inefficient. Conversely, if the increment is too high, then there will be risk of training destabilization by catastrophic forgetting due to a large step change in the training distribution. For our current experiments, we keep $\Delta_\theta$ as 1 for all dimensions. However, this value is not optimal in general, and the best setting is likely domain-specific.

## D  Experimental details

**Implementation**   Our work is implemented within the Dual Curriculum Design (DCD) codebase (Jiang et al., 2021a).[1] We use the official implementations of PLR$^\perp$ and ACCEL as provided in this codebase.

**Task solved threshold**   The task solved threshold $R_S$ indicates when a task has been solved based on its reward. An agent that receives a reward of at least $R_S$ on a task is said to have solved a task. This threshold is used to determine when training is no longer needed in a few ways in this work:

1. When the evaluation reward obtained on the target task distribution meets or exceeds $R_S$, the RL training procedure concludes.

2. CURATE uses $R_S$ to calculate the rewards $\nu$ based on competence (Eq. 5) used for the curriculum policy, which favors learning the set of easiest tasks not yet solved.

3. $R_S$ is used by the incremental curriculum baseline to indicate when it is time to advance to the next set of tasks in the curriculum.

We assume that $R_S$ is provided as part of the task definition. In practice, we train an RL agent using a random curriculum (i.e., domain randomization) to obtain what the maximum achievable reward in the target task distribution is. Then, we set $R_S$ slightly below that value.

**Maximum number of training frames**   The maximum allowable frames $f_{max}$ provides an upper limit to how long the RL agent is trained. Generally, it is determined as 2-5 times the average frames required for a random curriculum (i.e., domain randomization) to reach a target reward of at least $R_S$.

### D.1  MiniGrid MultiRoom Navigation

MiniGrid MultiRoom requires the RL agent to master grid-based navigation within the MiniGrid domain (Chevalier-Boisvert et al., 2023). In MultiRoom, the agent must navigate through a series of

---

[1]https://github.com/facebookresearch/dcd

---

**Algorithm 3:** GENERATEINCREMENTALCURRICULUM: Generate incremental curriculum

---

**Input:** easiest environment parameters $\theta_e$, target environment parameters $\theta_t$

**Initialize:** environment parameter dimensionality $N_d \leftarrow \dim(\theta_e)$, environment parameter increment $\Delta_\theta \leftarrow \vec{1}$, shortest path curriculum vector $\vec{v}_{SP} \leftarrow \theta_t - \theta_e$, shortest path curriculum vector magnitude $\hat{v}_{SP} \leftarrow \vec{v}_{SP}/||\vec{v}_{SP}||_2$, number of curriculum steps per dimension $\boldsymbol{N_\Delta} \leftarrow \{\mathrm{ceil}(\vec{v}_{SP}[d]\,/\Delta_\theta[d]) \mid d \in [1, N_d]\}$, total number of curriculum steps $N_{\Sigma_\Delta} \leftarrow \sum_{d=1}^{N_d} \boldsymbol{N_\Delta}[d]$, candidate incremental curricula $\boldsymbol{\mathcal{C}_i} \leftarrow \emptyset$, candidate incremental curricula errors $\boldsymbol{e_i} \leftarrow \emptyset$

---

```
// Loop over all possible initial increments
```
**for** $N'_{\Delta,init} = 1$ **to** $\boldsymbol{N_\Delta}[1]$ **do**
$\quad \theta' \leftarrow \theta_e$
$\quad \theta'_{SP} \leftarrow \theta_e$
$\quad N'_\Delta \leftarrow 0$
$\quad \boldsymbol{\mathcal{C}'_i} \leftarrow \{\theta_e\}$
$\quad e'_i \leftarrow 0$
$\quad$ **while** $N'_\Delta < N_{\Sigma_\Delta}$ **do**
$\quad\quad$ **for** $d = 1$ **to** $N_d$ **do**
$\quad\quad\quad$ **while** $((\theta'[d] \leq \theta'_{SP}[d])$ **or** $(N'_\Delta < N'_{\Delta,init}))$ **and** $(\theta'[d] < \theta_t[d])$ **do**
```
                    // Increment environment parameter along specified
                       dimension
```
$\quad\quad\quad\quad \theta'[d] \leftarrow \min(\theta'[d] + \Delta_\theta[d], \theta_t[d])$
```
                    // Update steps taken and add to incremental
                       curriculum
```
$\quad\quad\quad\quad N'_\Delta \leftarrow N'_\Delta + 1$
$\quad\quad\quad\quad \boldsymbol{\mathcal{C}'_i} \overset{+}{\leftarrow} \theta'$
```
                    // Calculate error with respect to shortest path
```
$\quad\quad\quad\quad \vec{v}_{\theta'} \leftarrow \theta' - \theta_e$
$\quad\quad\quad\quad h_e \leftarrow \vec{v}_{\theta'} \cdot \vec{v}_{SP} \,/\, ||\vec{v}_{SP}||_2^2$
$\quad\quad\quad\quad \vec{v}_{proj} \leftarrow h_e \cdot \vec{v}_{SP}$
$\quad\quad\quad\quad \vec{v}_\perp \leftarrow \vec{v}_{\theta'} - \vec{v}_{proj}$
$\quad\quad\quad\quad e'_i \leftarrow e'_i + ||\vec{v}_\perp||_2$
$\quad\quad\quad$ **end**
```
                // Update point on the shortest path
```
$\quad\quad\quad h_{SP} \leftarrow (\theta'[d] - \theta_e[d]) \,/\, \hat{v}_{SP}[d]$
$\quad\quad\quad \theta'_{SP} \leftarrow h_{SP} \cdot \hat{v}_{SP} + \theta_e$
$\quad\quad$ **end**
$\quad$ **end**
```
    // Append to buffers
```
$\quad \boldsymbol{\mathcal{C}_i} \overset{+}{\leftarrow} \boldsymbol{\mathcal{C}'_i}$
$\quad \boldsymbol{e_i} \overset{+}{\leftarrow} e'_i$
**end**
```
// Choose the incremental curriculum with the least error
```
$i_h \leftarrow \mathrm{argmin}(\boldsymbol{e_i})$
$\boldsymbol{\mathcal{C}_{i,min}} \leftarrow \boldsymbol{\mathcal{C}_i}[i_h]$

---

**Result:** incremental curriculum with least error approximation to the shortest path $\boldsymbol{\mathcal{C}_{i,min}}$

---

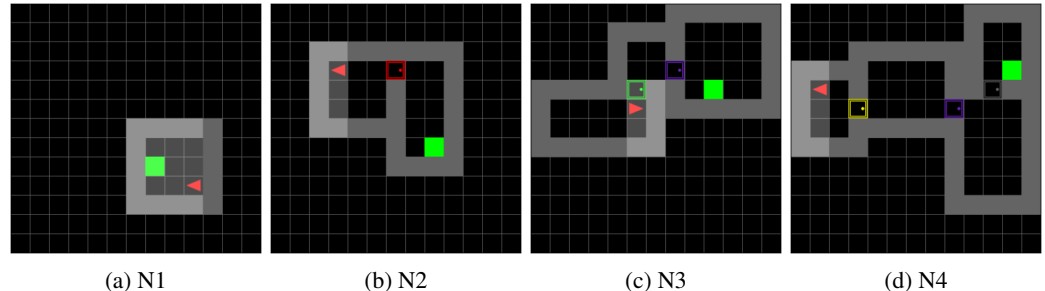

|                |                |                |                |
| :------------: | :------------: | :------------: | :------------: |
|    (a) N1      |    (b) N2      |    (c) N3      |    (d) N4      |

Figure 9: Variation in initial scenes for MultiRoom based on selection of environment parameters. Each figure represents an example task within the task distribution corresponding to the chosen environment parameters. For example, (a) represents a task with $\theta_1 = 1$ room.

rooms that are sequentially connected with doors separating the rooms. The agent always starts in the first room, and the goal always exists in the last room. The environment is a reimplementation of MultiRoom-Random-N4 (Jiang et al., 2021b). However, we use the typical MiniGrid observation space as described below.

**Observation space**  The agent receives two observations:

1. A field-of-view observation consisting of the states of the world within a 7 x 7 grid within the agent's line of sight. The agent cannot see through walls.

2. The direction the agent is facing, represented as an integer with one of four values that each represent a different direction.

**Action space**  The agent uses discrete actions with action space $|\mathcal{A}| = 7$. The actions include turning left, turning right, going forward, picking up an object, dropping an object, toggling the activation for an object, and doing nothing. As there are no objects for the agent to pick up in this domain, the actions for picking up and dropping an object have no effect. Toggling the activation in front of a door will either open it (if closed) or close it (if opened).

**Reward**  The agent receives a time-discounted reward when solving the level by reaching the goal; zero reward is received otherwise. A task is considered solved if the agent receives at least 0.7 reward.

**Task space**  MultiRoom is a one-dimensional task space, where the curriculum axis $\theta_1 = [1, 4]$ controls the number of rooms in each task. Figure 9 shows example tasks from each parameter in this task space.

**Target task distribution**  For MultiRoom, the agent must solve a task distribution consisting of $\theta_t = 4$ rooms.

D.2  PROCGEN CURRICULUM SUITE

Procgen, as introduced by Cobbe et al. (2020), tasks RL agents to master different types of discrete control games. For our experiments, we select the representative game of Leaper.

In our work, each game is adapted such that each level can be changed by specifying causal interventions in the environment parameters to change the initial level state. For example, the intervention $do(\theta_1 = 1, \theta_2 = 3)$ on level seed 0 in Leaper would yield the same level as without interventions, except with 1 road lane and 3 water lanes. Please refer to Fig. 10 for a visualized example. Note that for Leaper, intervention on these parameters may change other aspects of the initial state, such as the initial placement of cars and logs. Therefore, for Leaper, partial entanglement exists between $\Theta$ and other variables in the environment.

We implement our extensions of Procgen within the Procgen fork used by Jiang et al. (2021b)[2]

**Observation space**  The agent receives a 64 x 64 RGB image observation of the game.

**Action space**  The agent uses discrete actions with action space $|\mathcal{A}| = 15$. The actions generally correspond to eight directional actions, six special actions, and one action that does nothing. The actions are game-specific; please refer to Cobbe et al. (2020) for a complete description.

**Distribution mode**  We use the easy distribution mode of Procgen to avoid extra computational resources that would be required for the hard distribution mode.

**Reward**  The rewards for Procgen games are game-specific. For Leaper, the agent receives 10 reward when reaching the goal (zero otherwise).

**Task space: Leaper**  The curriculum axes specify the number of road lanes ($\theta_1 = [0, 3]$) and number of water lanes ($\theta_2 = [0, 3]$).

**Target task distribution**  Generally, the target task distribution for each game contains the hardest levels that would be obtained in each game under the easy distribution mode (i.e., $\theta_t = \max(\Theta)$). In other words, no level is harder than what would have been possible to experience when randomly sampling levels from Procgen.

### D.3 HYPERPARAMETERS

Table 2 presents the experimental hyperparameters. For MiniGrid MultiRoom, we use the hyperparameters from Jiang et al. (2021a) for their MiniGrid experiments. For Procgen, we generally use the same hyperparameters as the Procgen experiments in Jiang et al. (2021b) for the easy distribution. However, we use the episode length as defined by each game, and set the PPO rollout length to the nearest power of two. Then, we select minibatches per epoch such that each minibatch has 2048 samples, the same as in Jiang et al. (2021b).

For PLR$^{\perp}$, we generally use the same hyperparameters as Jiang et al. (2021a) for MultiRoom and Jiang et al. (2021b) for Procgen. Our ACCEL hyperparameters come from Jiang et al. (2021a).

## E  PROCGEN CURRICULUM SUITE

This work introduces the *Procgen Curriculum Suite*, an extension of Procgen (Cobbe et al., 2020) where each of the 16 games are assigned a structured task space $\Theta$ of varying dimensionality, from one-dimensional to four-dimensional (Tab. 3). This task space definition is intended to 1) make games easier to learn with a curriculum and 2) promote benchmarking for advancing the field of curriculum learning by proposing standard task spaces. Moreover, defining these task spaces facilitates future work in learning these task spaces by comparing learned task spaces against these human-curated task spaces.

Under the hood, the procedure generation is changed by allowing causal interventions in specific parameters of interest. For example, the game of Leaper has a two-dimensional task space: 1) $\theta_1$: number of road lanes and 2) $\theta_2$: number of water lanes. Specifying that distributions of levels should be generated with 2 water lanes is done by setting $do(\theta_2 = 2)$ in the generation process. We note that for some games, these causal interventions are sparse and local. Thus, they are disentangled with the rest of the environment factors of variation. This allows for precise evaluation of causal counterfactuals, opening up new possibilities for future work. However, for other games, the environment generation process has some degree of entanglement.

Intuitively, the Procgen Curriculum Suite is conceptually similar to C-Procgen (Tan et al., 2024), which also exposes the parameters used for procedural generation. Although C-Procgen exposes significantly more parameters, it remains an open question for which parameters in particular are

---

[2] https://github.com/minqi/procgen

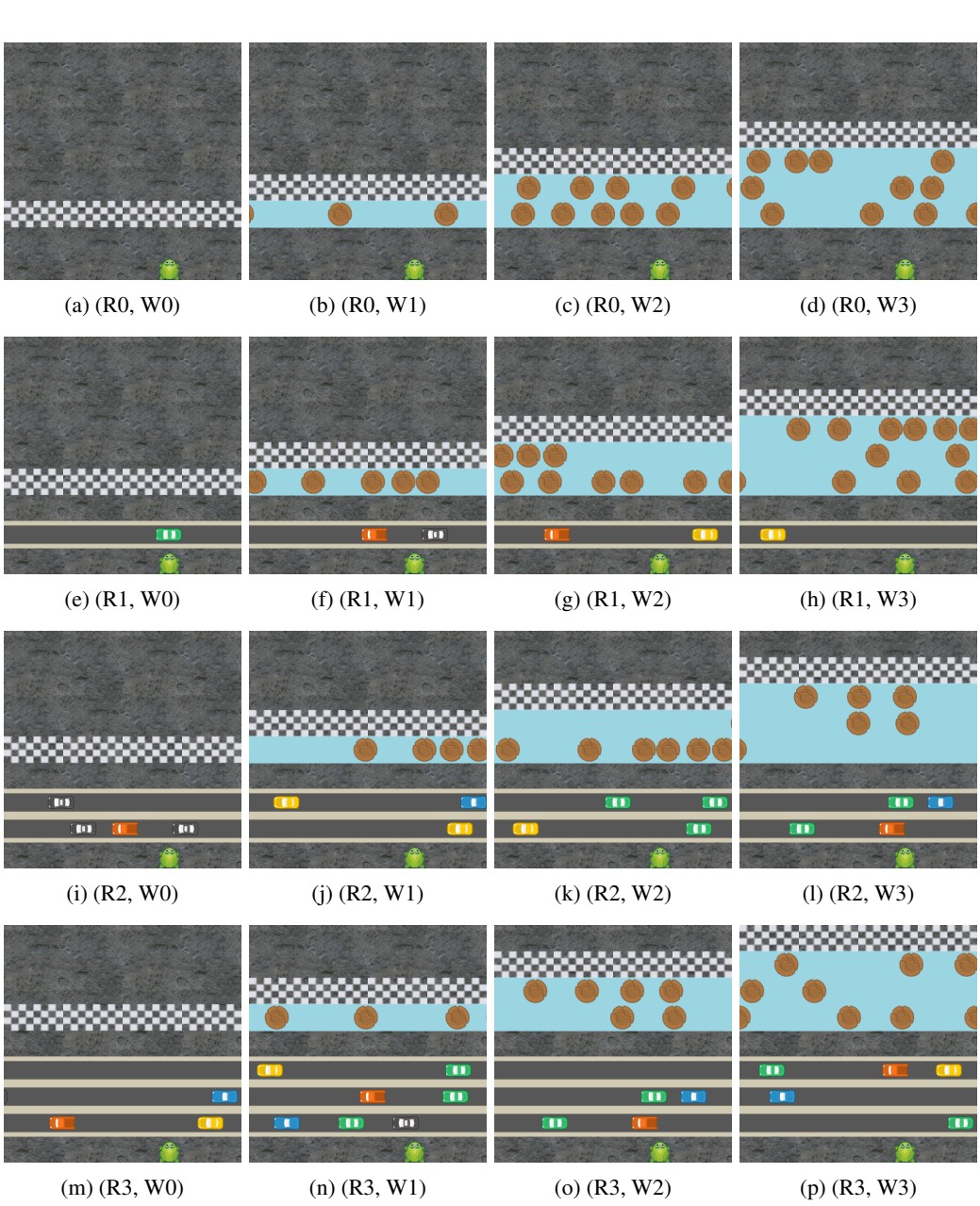

Figure 10: Example of variations in initial scenes for Leaper based on selection of environment parameters. Each figure represents an example task within the task distribution corresponding to the chosen environment parameters. For example, (h) represents a task with $\theta_1 = 1$ road lane and $\theta_2 = 3$ water lanes. All scenes are based on level seed 0.

Table 2: Hyperparameters used for experiments. Note that for CURATE, $N_r$ can take different values depending on whether it is the initial curriculum update or not.

| Hyperparameter | MiniGrid MultiRoom | Procgen Leaper |
|---|---|---|
| Discount factor $\gamma$ | 0.995 | 0.999 |
| $\lambda_{GAE}$ | 0.95 | 0.95 |
| Rollout length | 256 | 512 |
| Epochs | 5 | 3 |
| Minibatches per epoch | 1 | 16 |
| Clip range | 0.2 | 0.2 |
| Number of parallel environments $N_v$ | 32 | 64 |
| Return normalization | no | yes |
| Entropy bonus coefficient | 0.0 | 0.01 |
| Value loss coefficient | 0.5 | 0.5 |
| Max gradient norm | 0.5 | 0.5 |
| Adam learning rate | 0.0001 | 0.0005 |
| Adam $\epsilon$ | 0.00001 | 0.00001 |
| Recurrent agent | yes | no |
| Action space dimensionality $|\mathcal{A}|$ | 7 | 15 |
| Episode length | 80 | 500 |
| Reward threshold $R_S$ | 0.7 | 8.0 |
| Min. number of target episodes per eval. | 128 | 128 |
| Task space dimensionality $|\Theta|$ | 1 | 2 |
| Task space for $\Theta_1$ | [1, 4] | [0, 3] |
| Task space for $\Theta_2$ | n/a | [0, 3] |
| Max. train frames $f_{max}$ ($\times 10^6$) | 100.000 | 150.000 |
| Replay rate | 0.5 | 0.5 |
| PLR prioritization | rank | rank |
| Temperature $\beta$ | 0.3 | 0.1 |
| Staleness coefficient $\rho$ | 0.3 | 0.1 |
| Replay buffer size | 4000 | 4000 |
| Scoring function loss | positive value | L1 value |
| Edit rate | 1.0 | 1.0 |
| Replay rate | 0.8 | 0.8 |
| Number of edits | 3 | 3 |
| Edit method | random | random |
| Levels edited | easy | easy |
| Number rounds $N_r$ | 4/2 | 4/2 |
| Samples per round $N_s$ | 8 | 16 |
| Regularization hyperparameter $\lambda_\theta$ | auto | auto |
| Distance hyperparameter $\lambda_d$ | auto | auto |
| REPS relative entropy bound $\epsilon$ | 0.75 | 0.75 |
| REPS minimum temperature $\eta$ | 0.05 | 0.05 |
| Max. update frames $\Delta f_{sync}$ ($\times 10^6$) | 1.049 | 4.194 |

Table 3: Procgen Curriculum Suite games by task space dimensionality.

| $|\Theta| = 1$ | $|\Theta| = 2$ | $|\Theta| = 3$ | $|\Theta| = 4$ |
|---|---|---|---|
| BigFish | Climber | BossFight | FruitBot (hard) |
| CaveFlyer | CoinRun | FruitBot (easy) | StarPilot |
| Chaser | Heist | Plunder (hard) | |
| Dodgeball | Leaper | | |
| Jumper | Miner | | |
| Maze | Ninja | | |
| | Plunder (easy) | | |

Table 4: Procgen Curriculum Suite games by maximum episode length. A majority (11) of games have a maximum episode length $T$ of 1000. Two games each use $T = 500$ and $T = 4000$. The only game with $T = 6000$ is BigFish.

| $T = 500$ | $T = 1000$ | $T = 4000$ | $T = 6000$ |
|-----------|------------|------------|------------|
| Maze | CaveFlyer | BossFight | BigFish |
| Leaper | Chaser | Plunder | |
| | Climber | | |
| | CoinRun | | |
| | Dodgeball | | |
| | FruitBot | | |
| | Heist | | |
| | Jumper | | |
| | Miner | | |
| | Ninja | | |
| | StarPilot | | |

most useful or informative to use for curriculum learning. Thus, we suggest that the Procgen Curriculum Suite is distinguished in this regard by the definition of task spaces, so that other approaches in curriculum learning can be benchmarked using these task spaces. However, the greater breadth of control that C-Procgen offers is also useful for future work, such as learning the task spaces. Thus, although an overlap exists in capabilities, both frameworks offer important contributions to the curriculum learning literature.

Table 4 summarizes the games by maximum episode length.

### E.1 TASK SPACE DEFINITION

Table 5 (1D), Tab. 6 (2D), Tab. 7 (3D), and Tab. 8 (4D) present the task space $\Theta$ for each game based on dimensionality, along with a description of the task space parameters. Note that games may have differing task spaces between the easy and hard distribution modes. Usually, only the minimum and maximum values change between distribution modes, but two games have differing dimensionality due to the introduction of new mechanics (FruitBot: locked gates, Plunder: panels).

Task spaces were designed with several desiderata. First, task spaces were defined such that the hardest levels that occur at $\theta_t = \max(\Theta)$ can still be encountered in the standard version of Procgen. Put differently, the task spaces cannot yield levels that are more difficult than would have been generated normally for a particular distribution mode. Second, we designed the tasks spaces to have a diversity of dimensionality. For games where some options exist for which parameters should be used, we selected them based on balancing the overall dimensionality distribution. Third, some games, such as BigFish, had no obvious candidates for the parameters of the level procedural generation, so we occasionally define new sources of variation that can be controlled through the task space parameters. In the case of BigFish, we control the number of fish needed to solve the level. Fourth, we design the task space axes such that they are difficulty aligned, where increasing the value of the task space parameters leads to harder levels (in expectation). For cases where the variable in question in inversely proportional to difficulty, we reparameterize the variable such that it becomes proportional to difficulty. Fifth, in setting the lower bounds of the task spaces, we aimed to match the same distribution of easy levels that would have been encountered normally. Where possible, we avoided making the easiest levels trivially easy. There are some exceptions, e.g., StarPilot at very low finish line spawn times can be solved independent of agent action.

### E.2 TERMINAL REWARD MODE

For the original 16 games, only six (CoinRun, Heist, Jumper, Leaper, Maze, Ninja) provide episode rewards only upon successfully completing a level. The 10 other games (BigFish, BossFight, CaveFlyer, Chaser, Climber, Dodgeball, FruitBot, Miner, Plunder, StarPilot) provide intermediate feedback of varying frequency. Some feedback can still be quite sparse (e.g., destroying targets in CaveFlyer) or relatively dense (e.g., collecting orbs in Chaser). This intermediate feedback may be

Table 5: Definition of task spaces for games in the Procgen Curriculum Suite that have a one-dimensional task space ($|\Theta| = 1$). E $\downarrow$ and E $\uparrow$ mean the lower and upper bound respectively in easy mode, and H $\downarrow$ and H $\uparrow$ mean the lower and upper bound respectively in hard mode.

| | E $\downarrow$ | E $\uparrow$ | H $\downarrow$ | H $\uparrow$ | Description |
|---|---|---|---|---|---|
| | | | **BigFish** | | |
| $\theta_1$ | 1 | 30 | 1 | 30 | Number of fish needed to be eaten in order to complete the level. |
| | | | **CaveFlyer** | | |
| $\theta_1$ | 0 | 3 | 0 | 3 | Number of objects per chunk. The first objects are asteroids, the second objects are targets, and the third objects are enemy spaceships. |
| | | | **Chaser** | | |
| $\theta_1$ | 0 | 3 | 0 | 3 | Number of enemies. |
| | | | **Dodgeball** | | |
| $\theta_1$ | 3 | 6 | 3 | 6 | Number of enemies. |
| | | | **Jumper** | | |
| $\theta_1$ | 0 | 20 | 0 | 20 | Spike spawn probability in percentage. |
| | | | **Maze** | | |
| $\theta_1$ | 0 | 6 | 0 | 11 | Size of the maze. |

sufficiently informative such that exploration is no longer a concern, and the agent can train directly in the target tasks without a curricula. Therefore, for these 10 games, we introduce a *terminal reward mode*, which "sparsifies" these games such that a reward is only provided upon successfully completing the level. To avoid confusion, we append the word "Terminal" when referring to the terminal reward version of a game (e.g., Miner Terminal), or simply "-T" for brevity (e.g., Miner-T).

Generally, all other game mechanics are unchanged when in terminal reward mode, with the exception of FruitBot. Normally, in FruitBot, eating a fruit yields $+1$ reward, and eating a food item that isn't fruit yields a $-4$ penalty. An agent that is maximizing the sum of discounted returns would learn to eat fruit and avoid non-fruit. However, for FruitBot Terminal, the agent should only receive the completion bonus upon reaching the end of the level, regardless of the fruit consumed. Therefore, to preserve the incentive structure in FruitBot Terminal to eat fruit and avoid non-fruit, we introduce a health point system. The agent starts with three health points that are displayed in the top-right of the observation frame so that this information is observable to the agent. Eating a fruit adds a health point, up to a maximum of three. Eating a food item that isn't fruit removes a health point. If the agent reaches zero health points, the episode ends.

# F ANALYSIS OF CURATE CURRICULA

Below describe a comprehensive analysis of CURATE curricula.

## F.1 CURATE CURRICULA: STARTING IN EASIER TASKS

We find that for both MultiRoom and Leaper, the initial update for CURATE yields the best initial set of tasks through the combination of optimizing for competence (Eq. 5) and minimizing difficulty (Eq. 6). This is an important step, as the agent must learn behaviors quickly training from scratch, and tasks that are too difficult will lead to inefficient learning. For MultiRoom, for all 10 trials, the curricula starts in tasks generally consisting of one room (App. D.1, Fig. 9a), the easiest distribution of tasks ($\theta_i$ mean: $1.056 \pm 0.073$, $\theta_i$ median: $1.000 \pm 0.103$). Interestingly, for Leaper, we find

Table 6: Definition of task spaces for games in the Procgen Curriculum Suite that have a two-dimensional task space ($|\Theta| = 2$). E $\downarrow$ and E $\uparrow$ mean the lower and upper bound respectively in easy mode, and H $\downarrow$ and H $\uparrow$ mean the lower and upper bound respectively in hard mode.

| | E $\downarrow$ | E $\uparrow$ | H $\downarrow$ | H $\uparrow$ | Description |
|---|---|---|---|---|---|
| | | | **Climber** | | |
| $\theta_1$ | 1 | 10 | 1 | 10 | Number of platforms. |
| $\theta_2$ | 0 | 20 | 0 | 50 | Enemy spawn probability in percentage. |
| | | | **CoinRun** | | |
| $\theta_1$ | 1 | 3 | 1 | 3 | Level difficulty. |
| $\theta_2$ | 1 | 5 | 1 | 5 | Number of sections within a level. |
| | | | **Heist** | | |
| $\theta_1$ | 0 | 2 | 0 | 4 | Level difficulty. |
| $\theta_2$ | 0 | 3 | 0 | 3 | Number of locks/keys that prevent agent progress. |
| | | | **Leaper** | | |
| $\theta_1$ | 0 | 3 | 0 | 5 | Number of road lanes. |
| $\theta_2$ | 0 | 3 | 0 | 5 | Number of water lanes. |
| | | | **Miner** | | |
| $\theta_1$ | 0 | 3 | 0 | 12 | Number of diamonds. |
| $\theta_2$ | 0 | 20 | 0 | 80 | Number of boulders. |
| | | | **Ninja** | | |
| $\theta_1$ | 1 | 3 | 1 | 3 | Level difficulty. |
| $\theta_2$ | 1 | 5 | 1 | 5 | Number of sections within a level. |
| | | | **Plunder Easy** | | |
| $\theta_1$ | 1 | 20 | - | - | Number of ships to defeat to complete the level. |
| $\theta_2$ | 1 | 10 | - | - | Juice penalty when defeating a friendly ship. |

Table 7: Definition of task spaces for games in the Procgen Curriculum Suite that have a three-dimensional task space ($|\Theta| = 3$). E $\downarrow$ and E $\uparrow$ mean the lower and upper bound respectively in easy mode, and H $\downarrow$ and H $\uparrow$ mean the lower and upper bound respectively in hard mode.

| | E $\downarrow$ | E $\uparrow$ | H $\downarrow$ | H $\uparrow$ | Description |
|---|---|---|---|---|---|
| | | | **BossFight** | | |
| $\theta_1$ | 1 | 9 | 1 | 9 | Health points of the boss per round. |
| $\theta_2$ | 1 | 5 | 1 | 5 | Number of rounds for the fight. |
| $\theta_3$ | 2 | 3 | 2 | 5 | Duration of invulnerability at the beginning of each round. |
| | | | **FruitBot Easy** | | |
| $\theta_1$ | 1 | 5 | - | - | Number of walls. |
| $\theta_2$ | 0 | 60 | - | - | Inverse of gap distribution used for the walls ($80\% - \theta_2$). Difficulty is inverted such that increasing $\theta_2$ yields smaller wall gaps and more difficult levels. |
| $\theta_3$ | 0 | 10 | - | - | Number of bad food items. |
| | | | **Plunder Hard** | | |
| $\theta_1$ | - | - | 1 | 20 | Number of ships to defeat to complete the level. |
| $\theta_2$ | - | - | 1 | 10 | Juice penalty when defeating a friendly ship. |
| $\theta_3$ | - | - | 0 | 3 | Number of panels that block the agent's line of fire. |

Table 8: Definition of task spaces for games in the Procgen Curriculum Suite that have a four-dimensional task space ($|\Theta| = 4$). E $\downarrow$ and E $\uparrow$ mean the lower and upper bound respectively in easy mode, and H $\downarrow$ and H $\uparrow$ mean the lower and upper bound respectively in hard mode.

| | E $\downarrow$ | E $\uparrow$ | H $\downarrow$ | H $\uparrow$ | Description |
|---|---|---|---|---|---|
| | | | **FruitBot Hard:** $|\Theta| = 4$ | | |
| $\theta_1$ | - | - | 1 | 10 | Number of walls. |
| $\theta_2$ | - | - | 0 | 70 | Inverse of gap distribution used for the walls ($80\% - \theta_2$). Difficulty is inverted such that increasing $\theta_2$ yields smaller wall gaps and more difficult levels. |
| $\theta_3$ | - | - | 0 | 10 | Number of bad food items. |
| $\theta_4$ | - | - | 0 | 5 | Probability that a wall will be locked, in steps of 2.5%. |
| | | | **StarPilot:** $|\Theta| = 4$ | | |
| $\theta_1$ | 1 | 500 | 1 | 500 | Finish line spawn time. |
| $\theta_2$ | 1 | 20 | 1 | 20 | Inverse of minimum time between enemies ($30 - \theta_2$). Difficulty is inverted such that increasing $\theta_2$ yields less time between enemies and more difficult levels. |
| $\theta_3$ | 1 | 5 | 1 | 5 | Maximum group size for flyer enemies. |
| $\theta_4$ | 1 | 90 | 1 | 90 | Inverse of minimum time between flyer shots ($100 - \theta_4$). Difficulty is inverted such that increasing $\theta_4$ yields less time between flyer shots and more difficult levels. |

that the initial set of tasks is not tasks without road or water lanes ((R0, W0), App. D.2, Fig. 10a). These tasks are trivially easy, and the agent can solve them without any policy upgrades. Instead, 5 out of 6 trials started the agent with tasks mostly consisting of one road lane and zero water lanes ((R1, W0), App. D.2, Fig. 10e), and one trial blended (R1, W0) tasks with tasks consisting of zero road lanes and one water lane (R0, W1) (App. D.2, Fig. 10b). We observe that (R1, W0) tasks are usually easier for the agent and thus offer higher returns, leading to a preference for curricula with these starting tasks. The trial that also samples from (R0, W1) is a reflection that, due to stochastic initialization of the policy $\pi$, agents may prefer different starting tasks. Thus, the curricula offered by CURATE are specialized for the agent's starting competence.

## F.2 CURATE CURRICULA: NARROW TASK DISTRIBUTIONS

We find that for both MultiRoom and Leaper, the task distributions offered by CURATE are narrow, leading to effective learning by prioritizing tasks that match the agent's competence. We calculate metrics for the effective fraction of the task space that is covered by CURATE curricula throughout training. For MultiRoom, we find that the curriculum volume fraction is around 36% (mean: $36.015\% \pm 0.961\%$, $36.142\% \pm 0.424\%$). Given that the task space is bounded between [1, 4], we observe that CURATE curricula focuses on distribution widths about the size of one room. In two dimensions, the task space becomes larger, but CURATE is nonetheless able to maintain a small fraction: about 3.8% (mean: $3.829\% \pm 0.256\%$, median: $3.877\% \pm 0.155\%$).

## F.3 CURATE CURRICULA: EASIEST-TO-HARDEST PROGRESSION

The importance of an easy-to-hard task progression has been shown in prior work (e.g., Li et al. (2023)). We find that the task progression of CURATE curricula are approximately easiest-to-hardest, rising from the competence learning objective (Eq. 5). For MultiRoom, curricula generally progress in sequential order. In Leaper, we see some evidence of curriculum regression, i.e. regressing to easier tasks, but our best results occur with little to no regression.

## F.4 CURATE CURRICULA: TASK-DIRECTED PROGRESSION

In one-dimensional task spaces, solving tasks in an easy-to-hard fashion naturally leads to the target tasks. However, for tasks spaces with multiple dimensions, progressing from easy to hard becomes more challenging, as there are multiple paths the agent could take that could still yield easy-to-hard progression. Thus, for multidimensional task spaces, the ability for curricula to remain *task-directed* is key, as it drives the agent towards the overall goal of solving the target tasks. We find evidence that the orthogonal distance loss (Eq. 7) is beneficial for inducing a task-directed easy-to-hard progression, which empirically extends the findings of Li et al. (2023) to multidimensional task spaces. Visually, Fig. 3b shows that the curricula tend to remain close to $\vec{v}_{\theta_t - \theta_i}$. Empirically, we found that without the orthogonal distance loss, curricula tended to meander through the task space (even if progressing easy-to-hard), making learning more inefficient.

# G CURATE ABLATION EXPERIMENTS

CURATE's learning objectives are necessary for success. The competence $\mathcal{R}_{comp}$ is needed to induce the curriculum progression. The difficulty loss $\mathcal{L}_{diff}$ provides a "safety net" for environments where all tasks are too difficult, leading to a bias towards earlier levels independent of evaluation returns. The orthogonal distance loss $\mathcal{L}_{dist}$ imbues task-directedness, which we find critical in certain domains, such as the hard mode of PCS Leaper (PCS Leaper Hard). As shown in Tab. 9, we find that removing $\mathcal{L}_{dist}$ causes a significant performance degradation, halving the success rate. An analysis of the curriculum trajectories suggests that without the task-directedness provided by the orthogonal distance loss, curricula would diverge towards the edges of the task space. Training in these parts of the task space may hinder generalization, as they could lead to an agent only learning to generalize over a subset of environment parameters. Thus, advancement in the curriculum is hindered, or prevented all together, since the agent lacks generalization over all axes.

Table 9: Sample efficiency for PCS Leaper Hard in terms of student PPO updates required to solve $\mathcal{M}_{\theta_t}$ within the maximum allowable steps ($200 \times 10^6$). Summary statistics for 12 trials are shown in terms of Success Rate, Mean: mean steps with $\pm$ one standard deviation (STD), Median: median steps with $\pm$ one interquartile range (IQR), Min, and Max. Trials that do not solve the task still count towards summary statistics. Student PPO updates are $\times 10^3$. The best approach within 1 STD/IQR is **bolded**.

| Method | Success Rate | Mean | Median | Min | Max |
|---|---|---|---|---|---|
| CURATE | 1.000 (12) | **1.111** $\pm$ 0.214 | **1.076** $\pm$ 0.251 | 0.771 | 1.509 |
| CURATE without $\mathcal{L}_{dist}$ | 0.500 (6) | 1.431 $\pm$ 0.117 | 1.510 $\pm$ 0.181 | 1.184 | 1.525 |

## H  SUPPLEMENTAL RESULTS FOR MINIGRID MULTIROOM

Figure 11 presents a representative curricula and training/target learning curves for each approach. The representative trial for each approach is the closest trial to the median of all 10 trials used for that approach. We show results for CURATE with $R_S = 0.3$ as it was significantly better performaning than with $R_S = 0.7$.

### H.1  DOMAIN RANDOMIZATION EXPERIMENT WITH EVALUATION ON ALL TASKS

We conducted an experiment where we evaluate on all tasks after each PPO update, not just the target tasks. Figure 12 shows the results. Generally, we see that the evaluation returns on tasks that are nearby to those used for training increase inversely proportional to their difficulty. For example, an agent that is training on $\theta_2$ tasks will see some increase in evaluation return when evaluated on $\theta_3$ tasks and $\theta_4$ tasks due to transfer learning. However, this improvement is less for $\theta_4$ than $\theta_3$, as $\theta_4$ is more difficult than $\theta_3$. This property is leveraged by CURATE, so that as training tasks become solved, CURATE can select the easiest nearby tasks that are unsolved to progress in the curriculum.

## I  SUPPLEMENTAL RESULTS FOR PROCGEN CURRICULUM SUITE LEAPER

The learning curve and curricula for a representative trial for each approach are visualized in Figs. 13–14

## J  SUPPLEMENTAL RESULTS FOR THE PROCGEN CURRICULUM SUITE

Figure 15 shows the training and target learning curves for the composition of 14 games within the Procgen Curriculum Suite. These 14 games are BigFish Terminal (Fig. 16), BossFight Terminal (Fig. 17), CaveFlyer Terminal (Fig. 18), Climber Terminal (Fig. 19), CoinRun (Fig. 20), FruitBot Terminal (Fig. 21), Heist (Fig. 22), Jumper (Fig. 23), Leaper (Fig. 24), Maze (Fig. 25), Miner Terminal (Fig. 26), Ninja (Fig. 27), Plunder Terminal (Fig. 28), and StarPilot Terminal (Fig. 29). Chaser Terminal and Dodgeball Terminal were not included in the 14-game composition due to their difficulty; neither CURATE nor Incremental received any rewards during training, even for the easiest tasks.

## K  SUPPLEMENTAL RESULTS FOR BIPEDALWALKER

Figures 30–37 show the test return curves for the BipedalWalker domain.

## L  TASK SOLVED THRESHOLD SENSITIVITY EXPERIMENTS

How precisely does the task solved threshold $R_S$ need to be specified, and does its selection impact the relative performance of the investigated methods? These are the research questions we answered in a set of experiments for MiniGrid MultiRoom (Sec. L.1) and PCS Ninja (Sec. L.2).

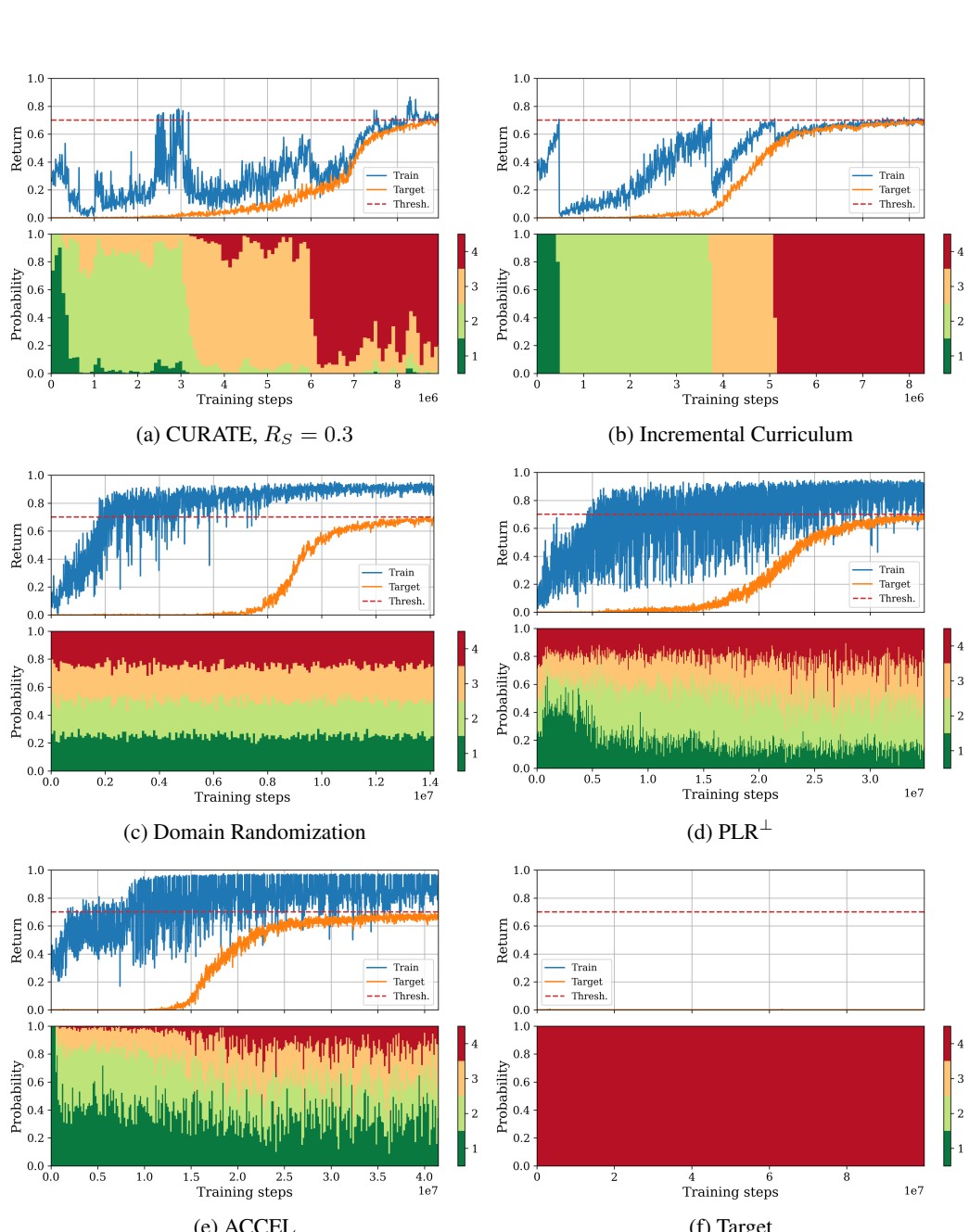

Figure 11: Representative curriculum learning time histories for each approach in MiniGrid Multi-Room. Each time history shows the trial that is closest to the median performance of all 10 trials for each approach. The top figure shows the time history of the return, shown for the training environments and the target task. The bottom figure shows the time history of the curriculum, with time average discretization of 10 updates to better show long-term trends.

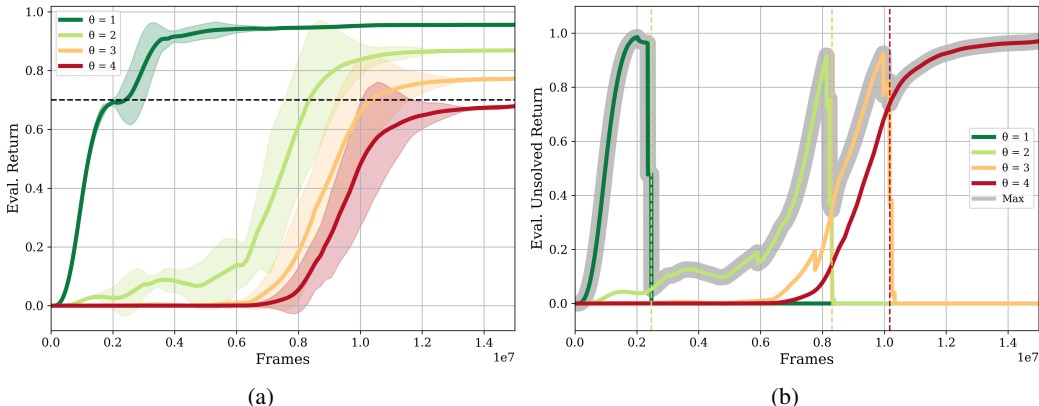

(a)                   (b)

Figure 12: (a) Even when using a random curriculum, we observe that tasks that are nearby those used for training will exhibit performance increases (measured by evaluation return) that are inversely proportional to their difficulty. Thus, the temporal ordering of task mastery (i.e., when a task is solved) occurs by difficulty. (b) Using a measure of task competence in unsolved tasks (Eq. 5), a natural easy-to-hard task ordering emerges. CURATE uses this task competence measure to induce an approximately easiest-to-hardest curriculum that outperforms curricula that do not exploit this property. Experiments are from the MiniGrid MultiRoom domain (Sec. 5).

Our experiments suggest that both CURATE and IC are generally insensitive to the precise selection of $R_S$, and methods more-or-less keep the same general ranking, with some exceptions. This is advantageous, as it improves the utility of CURATE (and IC) by not requiring highly precise selection. The selection of $R_S$ can thus be made simply on what level of capability is needed for a particular domain or use case.

Furthermore, our experimental evidence suggests that for some domains, like MultiRoom, using a lower task solved threshold $R_S$ during training than what is used for the target tasks can significantly boost sample efficiency. For the case of MultiRoom, this modification can overcome "bottlenecks" in the curriculum and progress the agent along faster to reaching and training in the target tasks. However, for other domains, such as PCS Ninja, this finding does not hold, and we find that keeping the same value of $R_S$ for both the training tasks and the target tasks is best. These findings suggest opportunities to better manage the selection of $R_S$ may lead to improvements. The task reward threshold does not have to be static during training, and it can potentially even be learned online by trading off between agent mastery of training tasks and rapid advancement in the curriculum. However, we leave investigations of such opportunities for future work.

## L.1 MULTIROOM TASK SOLVED THRESHOLD SENSITIVITY

Results for using a fixed value of $R_S$ throughout training are shown in Fig. 38. We find that the relative ordering is generally consistent below $R_S = 0.5$, but this order changes for higher values of $R_S$. However, we find that Incremental usually performs best, followed by CURATE and Domain Randomization, which have similar levels of performance. Note that the variance for CURATE increases significantly for $R_S$ values above 0.5.

If $R_S$ can differ between training and target tasks, as shown in Fig. 39, we find that CURATE can achieve significant performance improvements, matching Incremental. Generally, we find that the lower the value of $R_S$ used by CURATE for training tasks, the higher the performance increase. This finding may arise because a lower threshold allows the agent to progress more rapidly through the curriculum, where it can then devote its training budget to solving the target tasks. We find that the selection of $R_S$ used for Incremental does not lead to statistically significant differences.

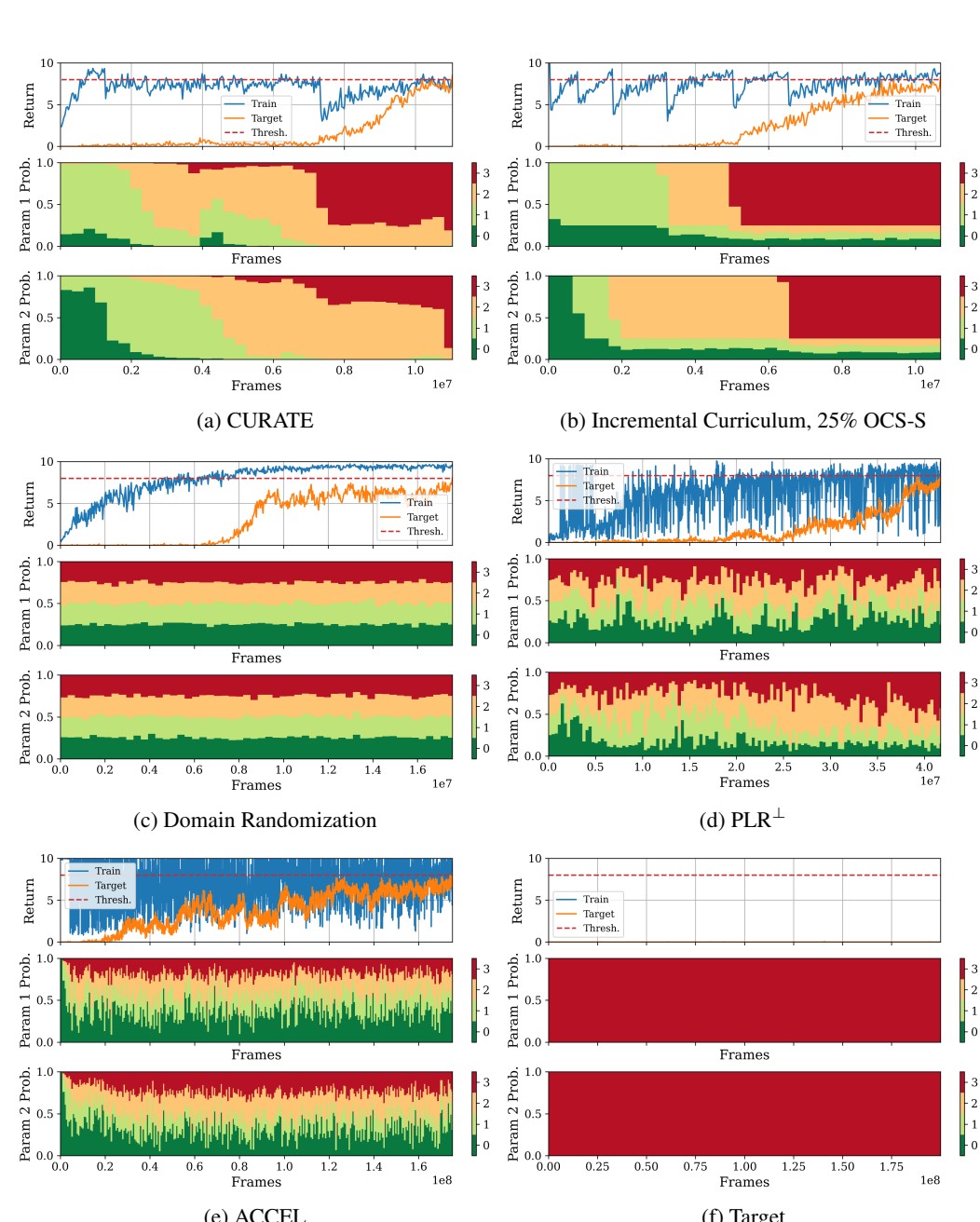

Figure 13: Representative curriculum learning time histories for each approach in Procgen Curriculum Suite Leaper. Each time history shows the trial that is closest to the median performance of all 6 trials for each approach. The top figure shows the time history of the return, shown for the training environments and the target task. The bottom figures show the time history of the curriculum, with time average discretization of 10 updates to better show long-term trends.

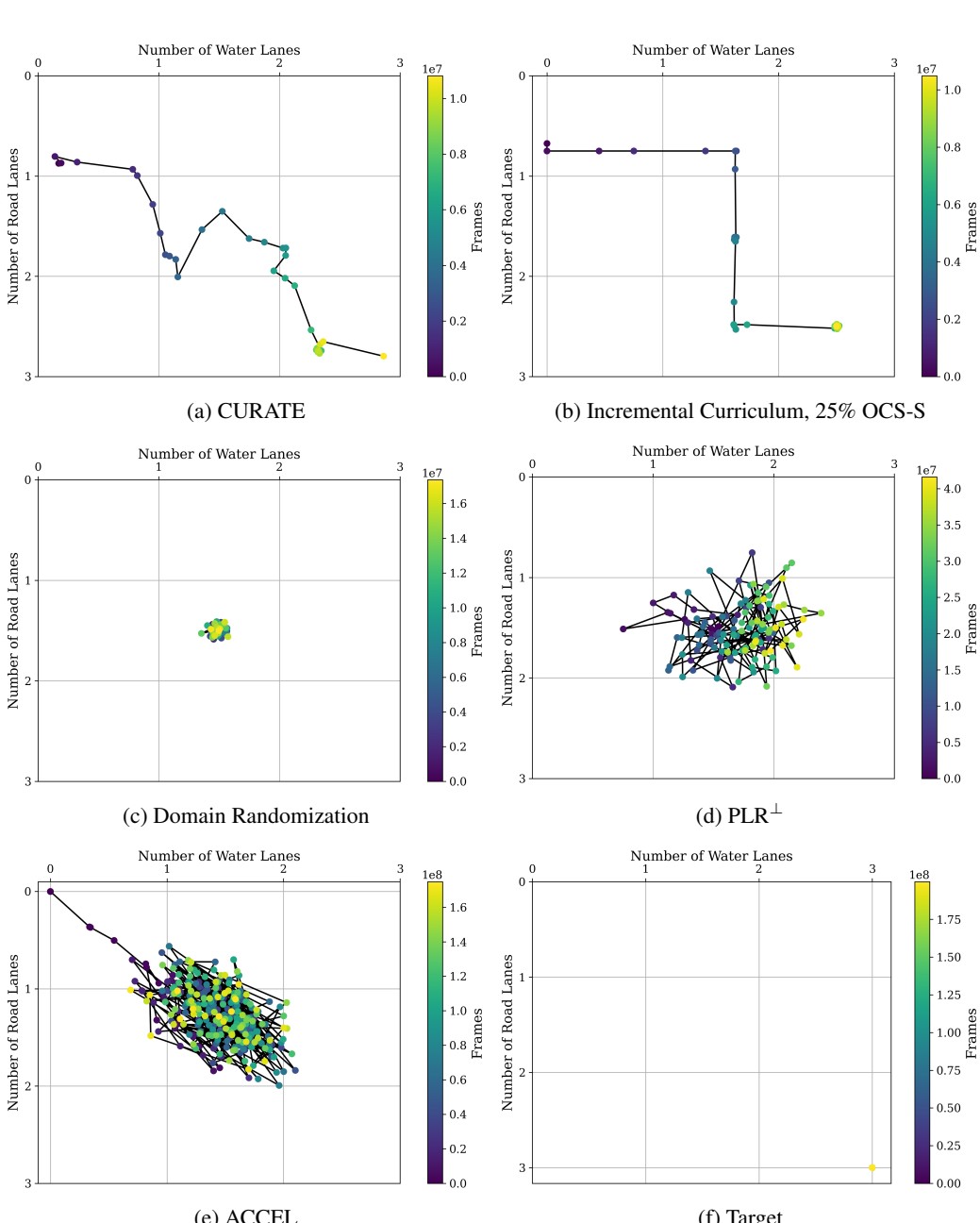

Figure 14: Representative curriculum for each approach in Procgen Curriculum Suite Leaper as represented by the mean environment parameters of the training tasks with time average discretization of 10 updates to better show long-term trends. The trial chosen for the curriculum is the closest to the median performance of all 6 trials of each approach. Note that the colorbar for each figure has a different maximum value.

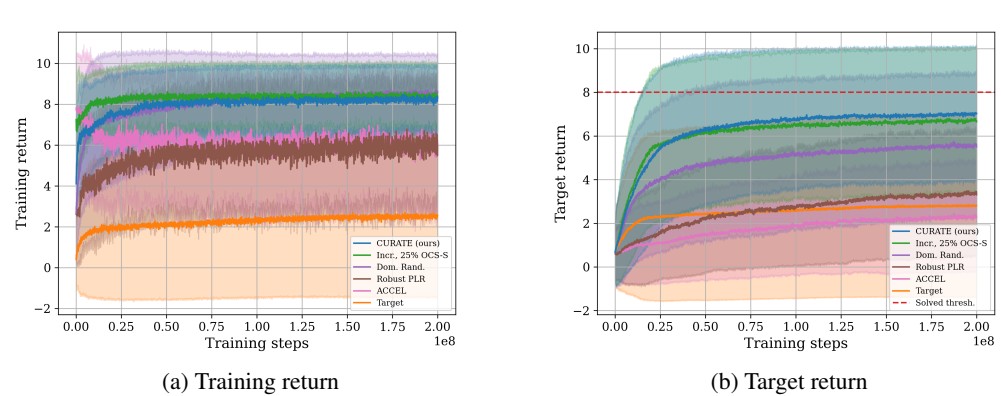

(a) Training return        (b) Target return

Figure 15: Composition of 14 of 16 Procgen Curriculum Suite games for the (a) training return curve and (b) target return curve.

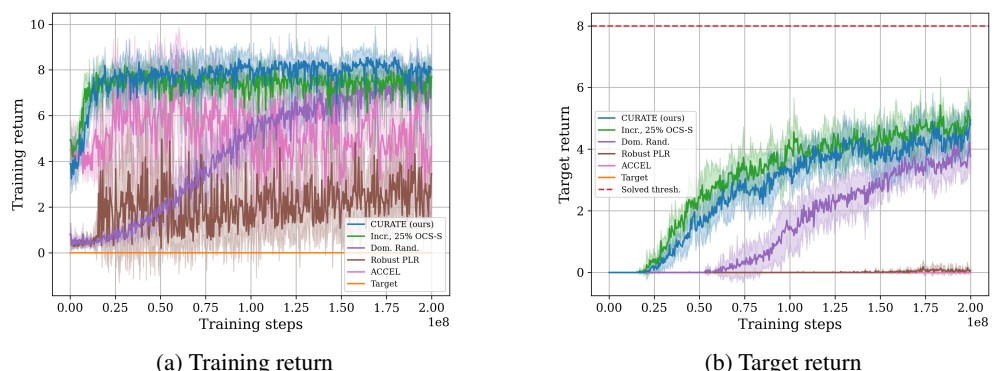

(a) Training return        (b) Target return

Figure 16: PCS BigFish Terminal results: (a) training return curve and (b) target return curve.

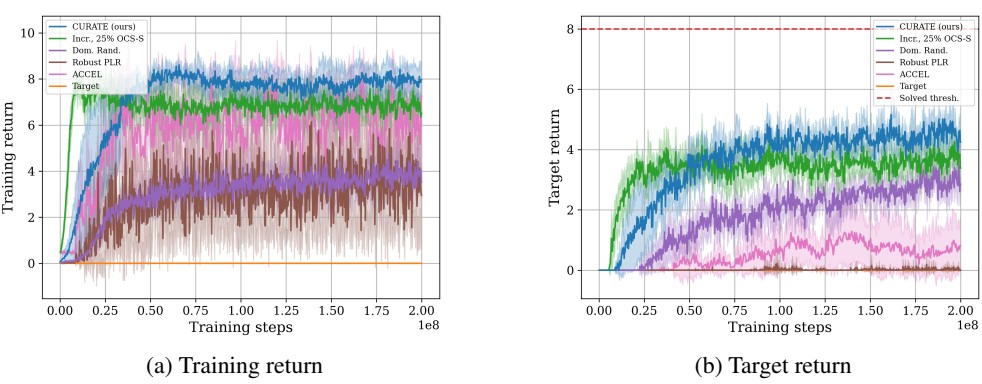

(a) Training return        (b) Target return

Figure 17: PCS BossFight Terminal results: (a) training return curve and (b) target return curve.

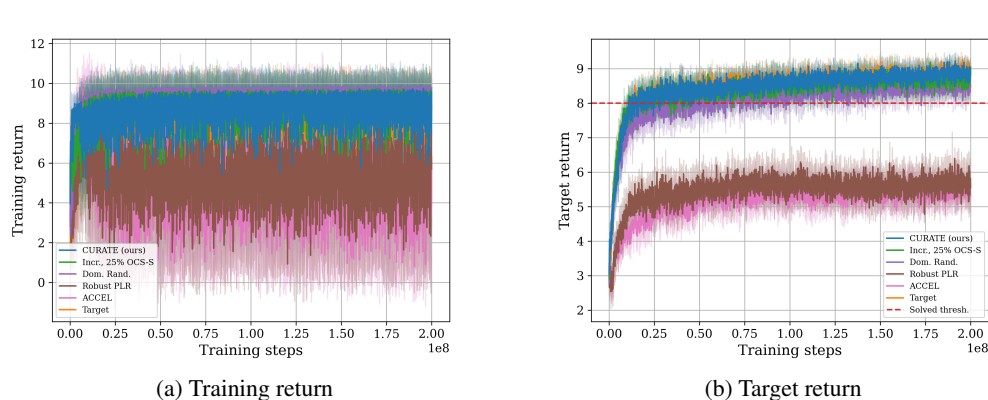

(a) Training return            (b) Target return

Figure 18: PCS CaveFlyer Terminal results: (a) training return curve and (b) target return curve.

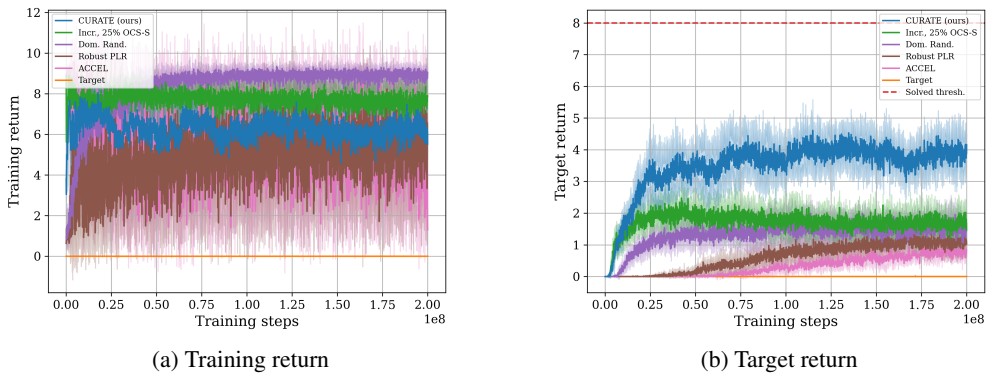

(a) Training return            (b) Target return

Figure 19: PCS Climber Terminal results: (a) training return curve and (b) target return curve.

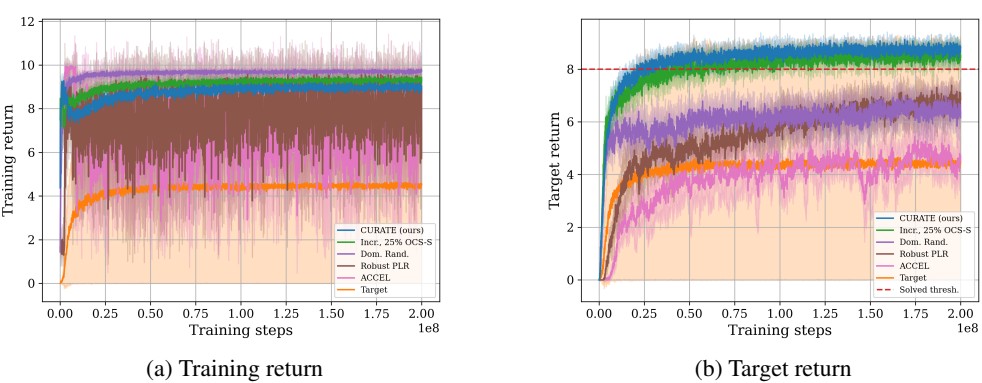

(a) Training return            (b) Target return

Figure 20: PCS CoinRun results: (a) training return curve and (b) target return curve.

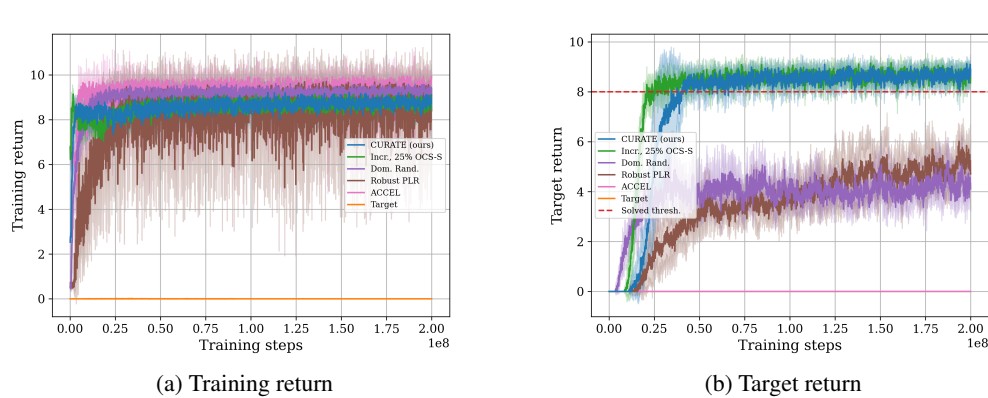

(a) Training return

(b) Target return

Figure 21: PCS FruitBot Terminal results: (a) training return curve and (b) target return curve.

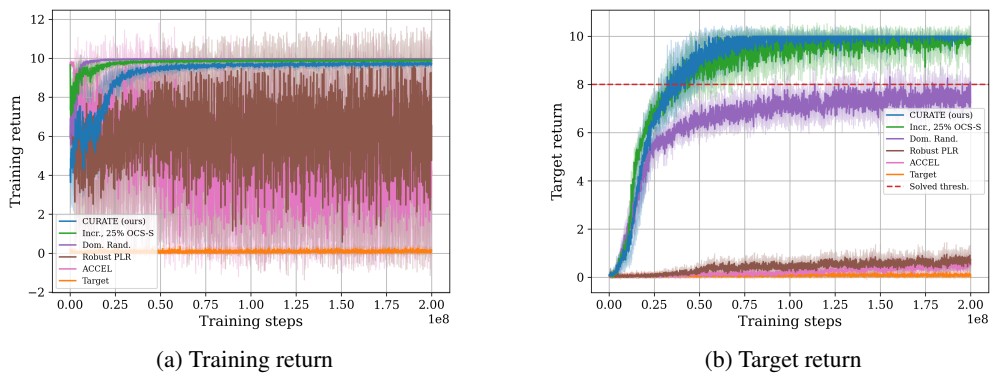

(a) Training return

(b) Target return

Figure 22: PCS Heist results: (a) training return curve and (b) target return curve.

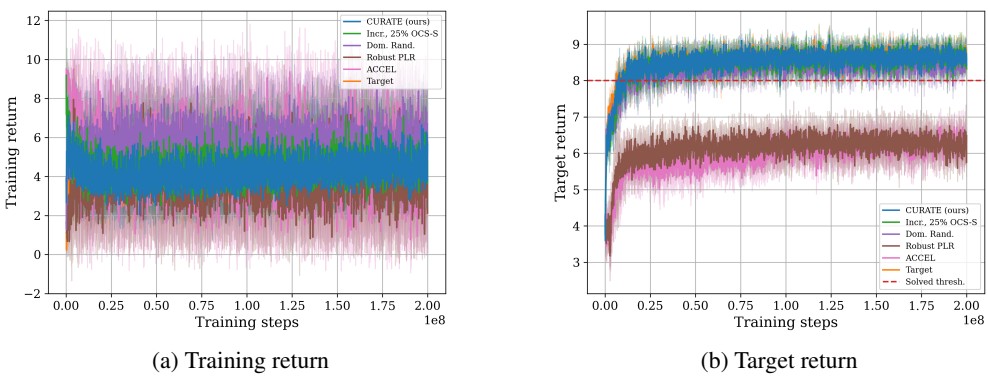

(a) Training return

(b) Target return

Figure 23: PCS Jumper results: (a) training return curve and (b) target return curve.

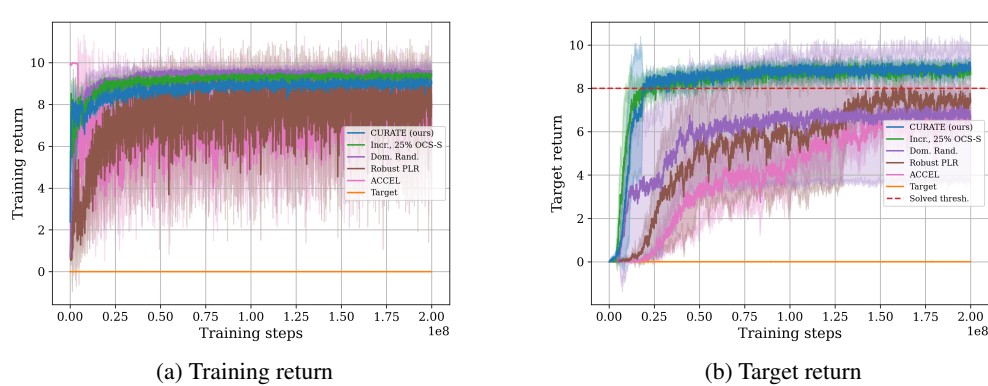

(a) Training return

(b) Target return

Figure 24: PCS Leaper results: (a) training return curve and (b) target return curve.

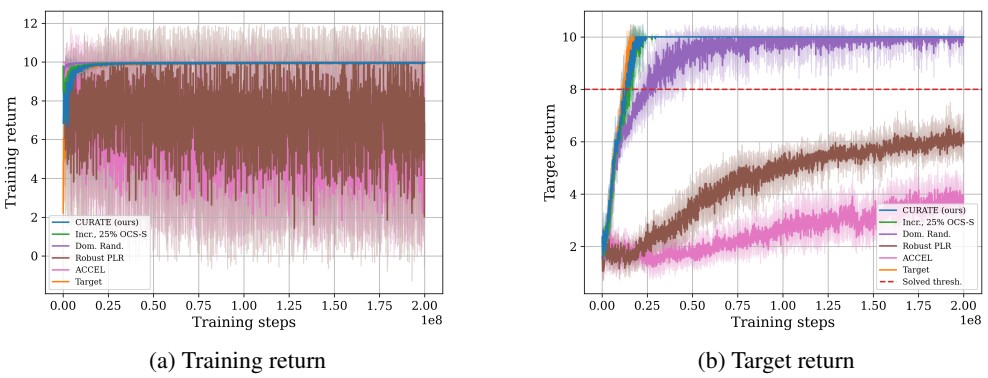

(a) Training return

(b) Target return

Figure 25: PCS Maze results: (a) training return curve and (b) target return curve.

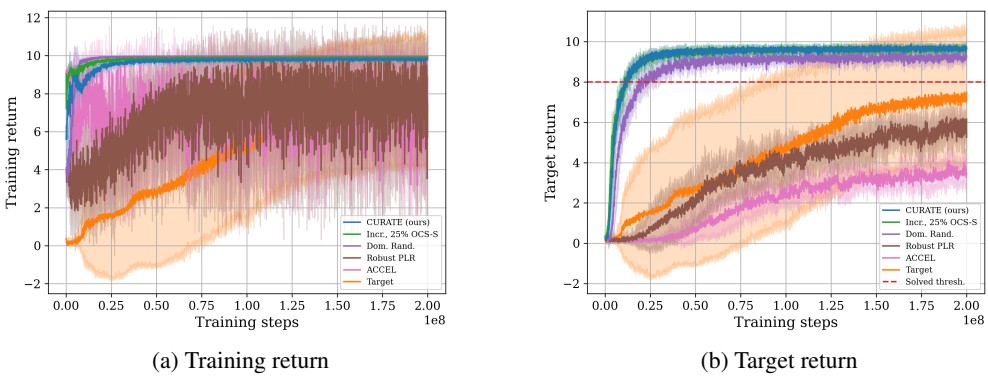

(a) Training return

(b) Target return

Figure 26: PCS Miner Terminal results: (a) training return curve and (b) target return curve.

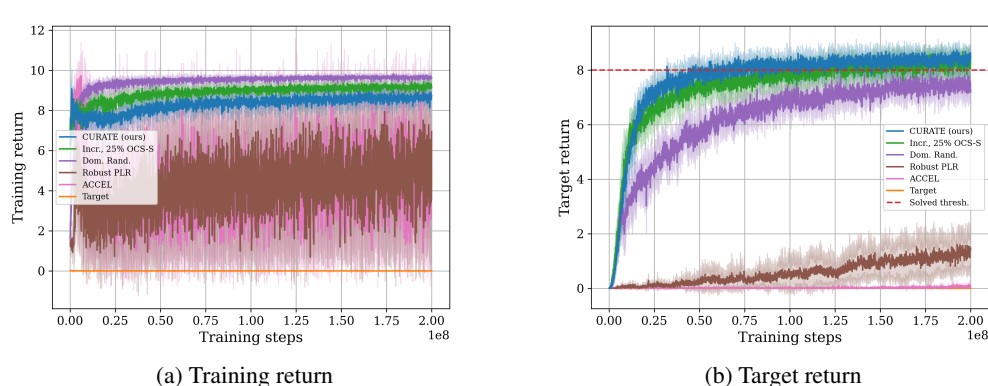

(a) Training return

(b) Target return

Figure 27: PCS Ninja results: (a) training return curve and (b) target return curve.

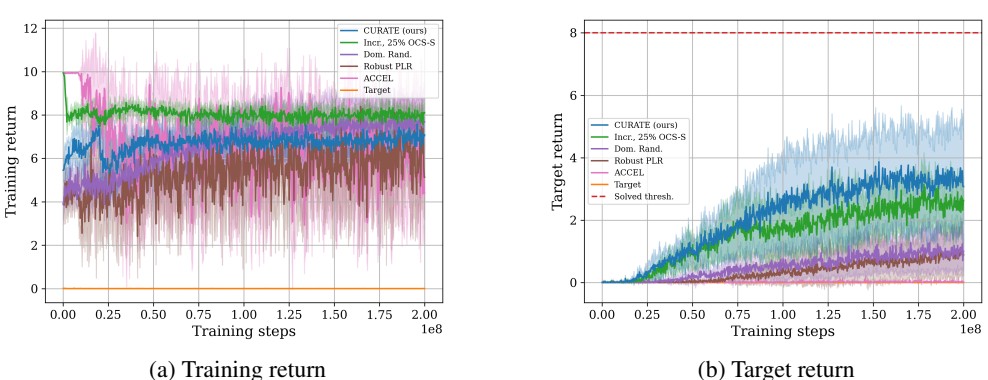

(a) Training return

(b) Target return

Figure 28: PCS Plunder Terminal results: (a) training return curve and (b) target return curve.

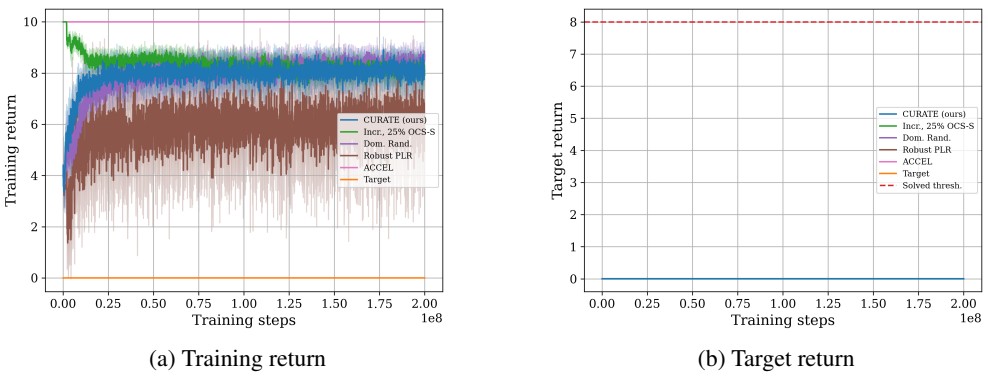

(a) Training return

(b) Target return

Figure 29: PCS StarPilot Terminal results: (a) training return curve and (b) target return curve.

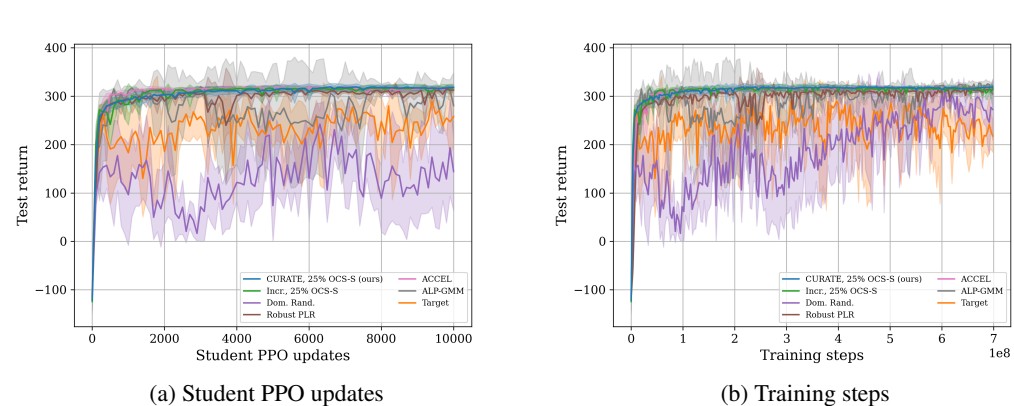

(a) Student PPO updates

(b) Training steps

Figure 30: Test returns for the `BipedalWalker` environment in terms of (a) student PPO updates and (b) training steps.

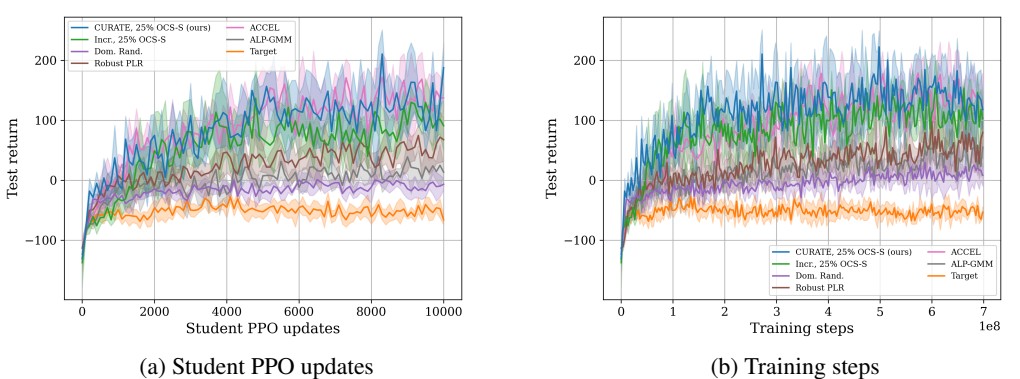

(a) Student PPO updates

(b) Training steps

Figure 31: Test returns for the `BipedalWalker-Hardcore` environment in terms of (a) student PPO updates and (b) training steps.

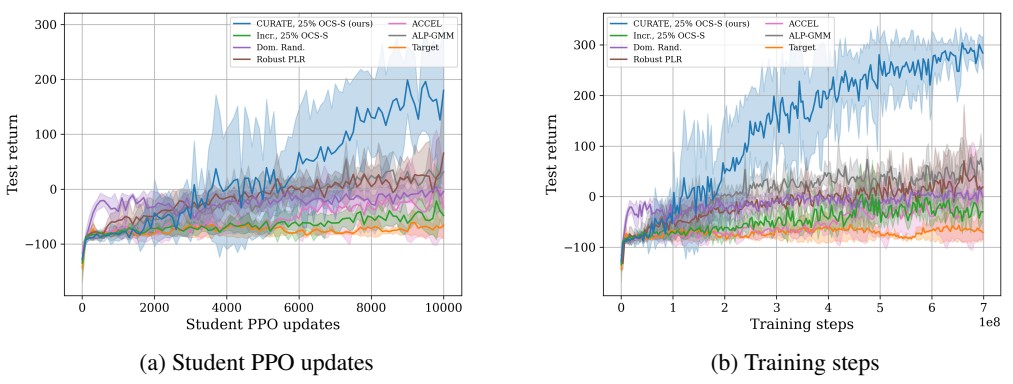

(a) Student PPO updates

(b) Training steps

Figure 32: Test returns for the `PitGap` environment in terms of (a) student PPO updates and (b) training steps.

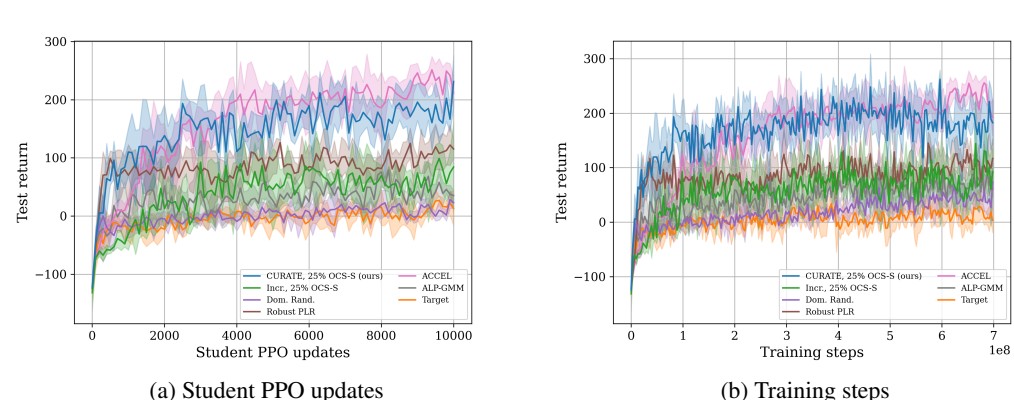

(a) Student PPO updates

(b) Training steps

Figure 33: Test returns for the `Roughness` environment in terms of (a) student PPO updates and (b) training steps.

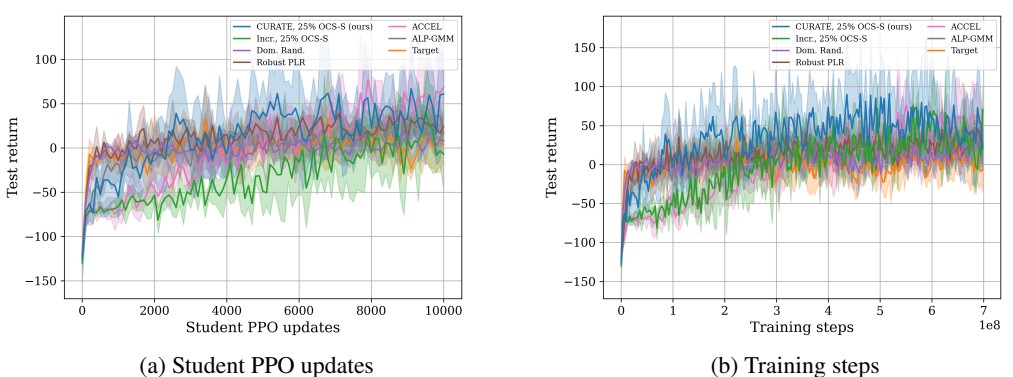

(a) Student PPO updates

(b) Training steps

Figure 34: Test returns for the `Stairs` environment in terms of (a) student PPO updates and (b) training steps.

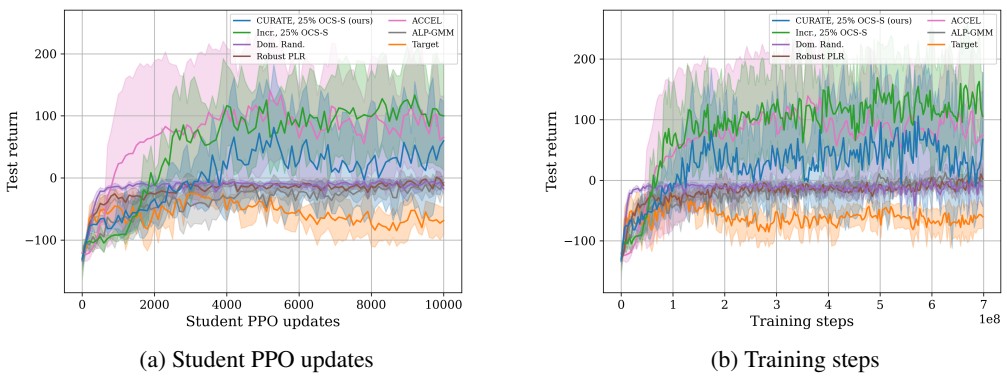

(a) Student PPO updates

(b) Training steps

Figure 35: Test returns for the `Stump` environment in terms of (a) student PPO updates and (b) training steps.

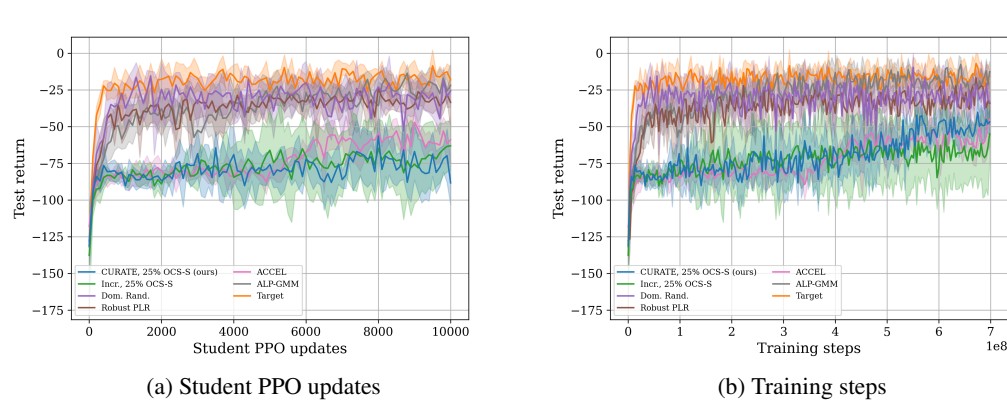

(a) Student PPO updates      (b) Training steps

Figure 36: Test returns for the `BipedalWalker-Max` environment in terms of (a) student PPO updates and (b) training steps. This is the target environment for the BipedalWalker domain.

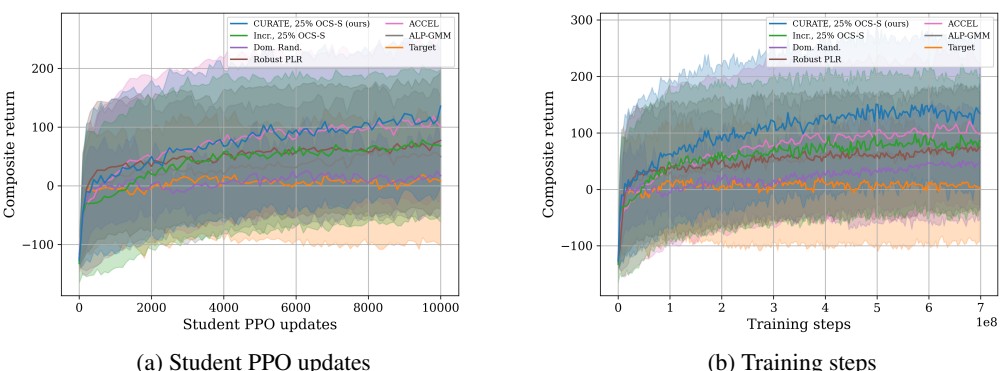

(a) Student PPO updates      (b) Training steps

Figure 37: Composite test returns for all seven test environments in the BipedalWalker domains in terms of (a) student PPO updates and (b) training steps.

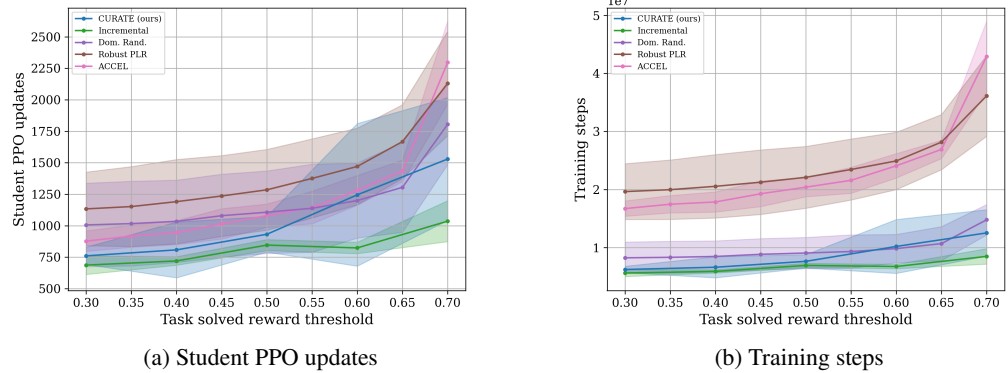

(a) Student PPO updates      (b) Training steps

Figure 38: Reward sensitivity experiment for MiniGrid MultiRoom that varies the task solved return threshold $R_S$ in terms of (a) student PPO updates and (b) training steps. For CURATE and Incremental, the selected value of $R_S$ is used for training and is the same as the value used for the target tasks.

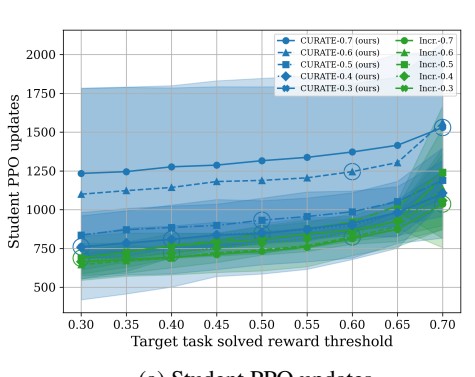 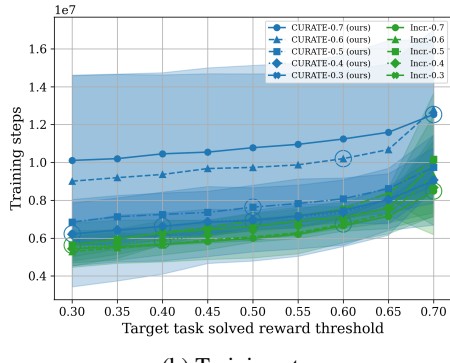

(a) Student PPO updates

(b) Training steps

Figure 39: Reward sensitivity experiment for MiniGrid MultiRoom that has separate task solved return thresholds $R_S$ for both training and target tasks in terms of (a) student PPO updates and (b) training steps. The value of $R_S$ used for training is shown in the legend (e.g., CURATE-0.3 means CURATE with $R_S = 3$). The value of $R_S$ on the $x$-axis specifies the threshold used to determine if the target tasks are solved, independent of the choice of $R_S$ used for training. Configurations where training and target tasks have the same value of $R_S$ are annotated with a circle.

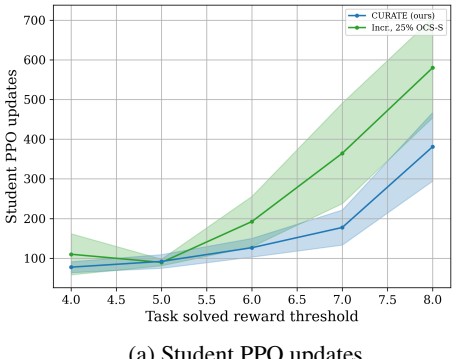 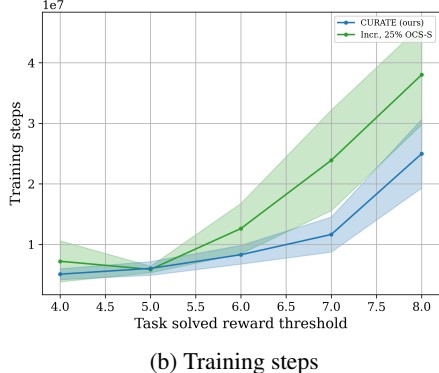

(a) Student PPO updates

(b) Training steps

Figure 40: Reward sensitivity experiment for PCS Ninja that varies the task solved return threshold $R_S$ in terms of (a) student PPO updates and (b) training steps. For CURATE and Incremental, the selected value of $R_S$ is used for training and is the same as the value used for the target tasks.

## L.2 PCS NINJA TASK SOLVED THRESHOLD SENSITIVITY

Figure 40 shows the results for a fixed value of $R_S$. Unlike in MultiRoom, we find that CURATE is generally superior to Incremental over a change of $R_S$, although the difference narrows for smaller $R_S$.

When $R_S$ can differ between the training and target tasks, we find that CURATE still holds an advantage over Incremental, but primarily for larger values of $R_S$ used for the target tasks. At low values of $R_S$ used for the target tasks, this difference is small. Besides CURATE with $R_S = 8$, we do not find evidence that changing $R_S$ used for training significantly affects performance. However, for Incremental, we find limited evidence that lower values of $R_S$ used for training hurts performance, suggesting that higher values of $R_S$ for training are better.

In both Fig. 40 and Fig. 41, only CURATE and Incremental are shown, as these two methods were significantly better than other methods.

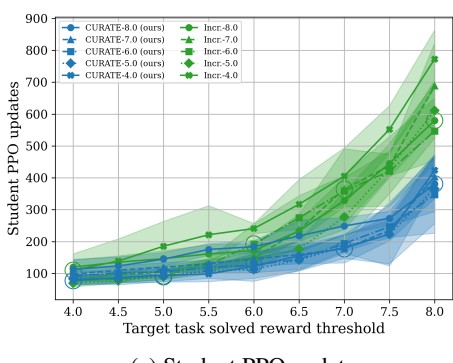 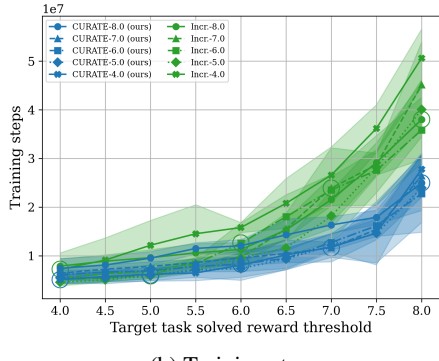

(a) Student PPO updates              (b) Training steps

Figure 41: Reward sensitivity experiment for PCS Ninja that has separate task solved return thresholds $R_S$ for both training and target tasks in terms of (a) student PPO updates and (b) training steps. The value of $R_S$ used for training is shown in the legend (e.g., CURATE-0.3 means CURATE with $R_S = 3$). The value of $R_S$ on the $x$-axis specifies the threshold used to determine if the target tasks are solved, independent of the choice of $R_S$ used for training. Configurations where training and target tasks have the same value of $R_S$ are annotated with a circle.

## M    USE OF LARGE LANGUAGE MODELS

We acknowledge that a Large Language Model (LLM) was occasionally used as an assist tool to augment the retrieval and discovery of related work of this manuscript beyond the primary retrieval and discovery by the coauthors. All related works discovered by the LLM were individually examined to ensure consistency and correctness of the retrieval. Integration of the relevant discovered works into the related works section was completed by the coauthors without LLM guidance. None of the content of this manuscript was generated, curated, or edited by an LLM.

