# OpenReview forum: "CURATE: Automatic Curriculum Learning for Reinforcement Learning Agents through Competence-Based Curriculum Policy Search"
_ICLR.cc/2026/Conference — Submitted to ICLR 2026_

### Official Review · Reviewer_X7fM · 2025-10-31

**Soundness:** 2
**Presentation:** 1
**Contribution:** 2
**Rating:** 2
**Confidence:** 5

**Summary:**

This paper presents CURATE, an automated curriculum generation method for sparse RL domains. Using relative entropy policy search, CURATE searches for a policy in the task space that prioritizes the easiest unsolved tasks as the policy improves. This induces an approximately easiest-to-hardest curriculum, where difficulty is inversely proportional to the discounted return.  Experiments on MiniGrid MultiRoom and Procgen Leaper demonstrate that CURATE improves sample efficiency in comparison to domain randomization and some UED methods. The paper also introduces a benchmark for systematic evaluation of curriculum learning methods.

**Strengths:**

- The paper has a detailed related work covering a wide range of curriculum learning works.
- The proposed method seems novel in its framing of exploration by exploitation for curriculum learning for RL.
- The domains of interest in the experiments seem difficult to tackle due to their sparse-reward nature.

**Weaknesses:**

- The paper is difficult to follow, the method section has a circular structure, and is confusing.
- The notion of difficulty is limited to a niche group of sparse reward domains where the number of steps to reach the goal is directly associated with the difficulty. Although mentioned in the limitations section, this is a clear obstacle against the practicality of the proposed approach.
- Despite claiming that the authors scale existing ideas to multidimensional settings, they only do it up to two-dimensional task spaces.
- There is no ablation for the impact of different reward components for training the curriculum policy.

**Questions:**

- What ablations would demonstrate the use of having different rewards for different stages?
- Is there a reason why ALP-GMM cannot handle multi-dimensional task spaces?
- Could you please give me an example domain where the assumption on the difficulty ordering doesn't hold, and explain how CURATE would perform there?

---

> ### Author Response · Authors · 2025-11-25
> **Our response to your initial review (addressing weaknesses)**
>
> We provide our thanks for your time in reading our submission. We are happy to hear our related work, novelty, and domains of interest were considered strong points. We also thank you for sharing your weaknesses and limitations, which we view as opportunities to strengthen the work and improve your review score.
>
> Before we continue, we wanted to share that we expect a new version of the paper within 4 days that may address some concerns. This new version will also include more Procgen experiments besides Leaper.
>
> Below are our responses to the weaknesses.
>
> ## W1: Difficult to Follow
>
> We will add figures, such as an overview figure, to help with clarity and to make the paper easier to follow.
>
> ## W2: Niche Application
>
> Although we acknowledge that the choice of assumptions – that the task space is parameterized with axes correlating with task difficulty – narrows its applicability, these assumptions empower CURATE to achieve state-of-the-art performance relative to UED/naive curriculum baselines that don’t require human expertise. In this light, we still view CURATE as an important contribution towards overall progress in automatic curriculum learning. Moreover, we are currently investigating how the task space can be learned, which would relax this assumption. However, this extension is significant and non-trivial, so we believe it is best served to investigate in a separate, follow-up work based on our current study with CURATE. Additionally, we note that the use of a curriculum (whether learned by CURATE or another method) may not be appropriate for all domains. For example, if dense feedback is available, it is unclear whether a curriculum would be more sample efficient than directly training in the target tasks. Therefore, we believe that the perception of CURATE being only applicable for a niche set of domains may also apply to other curriculum learning works as well.
>
> ## W3: Demonstrations for Multidimensional Task Spaces
>
> In our forthcoming paper version, we will show experimental results for higher dimensional task spaces. We hope that our future paper addresses these concerns. We note that the largest dimensionality in the Procgen Curriculum Suite is 4-dimensional (StarPilot).
>
> ## W4: Ablations
>
> We plan to run experiments to ablate the learning objective components for CURATE. (See also our response to Q1 below).
>
> We will address your questions in a separate comment below.

---

> > ### Author Response · Authors · 2025-11-25
> > **Responses to Questions 1-3**
> >
> > ## Q1: Ablations
> >
> > As mentioned in our response to W4, we plan to run ablations to show the effect of each learning objective component on the overall performance of CURATE. Note that the ablation for the orthogonal distance learning objective will only be shown for Procgen, as it only applies to multidimensional task spaces.
> >
> > ## Q2: ALP-GMM
> >
> > We find conflicting evidence in the literature regarding how well ALP-GMM [1] can scale to multidimensional task spaces. For example, in the ACCEL paper [2], which uses ALP-GMM as a baseline, describes that it does not scale: “Finally, we see ALP-GMM performs poorly when the design space is increased from 2D (as in Portelas et al. (2019)) to 8D.” [2]. Because ALP-GMM is designed for continuous task spaces, we do not consider it as a baseline for MultiRoom or Procgen. However, we are currently investigating the use of CURATE in a continuous parameter domain, and in this setting, we will also investigate the use of ALP-GMM as a baseline against CURATE.
> >
> > ## Q3: Assumption of Difficulty Ordering
> >
> > In the ACCEL paper, there is a domain where a room is designed with 0 - 60 blocks (Section 4.2 in the ACCEL paper [1]). In this “cluttered room” domain, we find that increasing the number of blocks in the room counterintuitively decreases the difficulty, by reducing the branching factor to navigate the room. This goes against the design assumption for CURATE, so although it is able to train the agent to solve the target task, the learned curricula are not optimal and it performs worse than Robust PLR and ACCEL. However, such a domain would be a good opportunity to learn a task parameterization that does have a structure that positively correlates with task difficulty, which we are currently pursuing (see our response to W2). Moreover, we emphasize that for domains that meet CURATE’s assumptions, we would expect state-of-the-art performance.
> >
> > [1] R. Portelas et al., “Teacher Algorithms for Curriculum Learning of Deep RL in Continuously Parameterized Environments”, CoRL 2020.
> >
> > [2] J. Parker-Holder and M. Jiang et al., “Evolving Curricula with Regret-Based Environment Design”, ICML 2022.

---

### Official Review · Reviewer_YzWB · 2025-10-31

**Soundness:** 2
**Presentation:** 1
**Contribution:** 2
**Rating:** 2
**Confidence:** 4

**Summary:**

The paper introduces CURATE, an automatic curriculum learning method for reinforcement learning tasks. The approach learns a curriculum policy and updates it using REPS to shift the task distribution toward the easiest unsolved tasks for the agent, resulting in an emergent easiest-to-hardest training progression. However, the approach requires a known and ordered task space to function. Experiments on MiniGrid MultiRoom and Procgen Leaper show that CURATE improves sample efficiency and outperforms several baselines.

**Strengths:**

- The paper tackles an important  challenge in reinforcement learning, how to automate curriculum generation to improve training efficiency in sparse-reward settings.
- The paper adapts the Procgen environment for curriculum-based evaluation. While only three environments are used (and mainly one discussed in the main text), these setups could still serve as useful benchmarks for the community and support more consistent evaluation of curriculum learning methods.

**Weaknesses:**

W1: The paper is difficult to follow. The description of the proposed method in the main text is mostly narrative, intermixed with mathematical formulas that are not well connected to the algorithmic intuition. The pseudocode in the appendix is poorly formatted and hard to read. Furthermore, the paper lacks any theoretical justification or analysis of why the proposed approach should converge or be optimal.

W2: Although the authors acknowledge this limitation, the requirement of having a known and ordered task space is a major assumption. CURATE depends on explicit task parameters that can be continuously ordered by difficulty, an assumption that many other baseline methods do not exploit. This significantly narrows the generality of the approach.

W3: The evaluation procedure appears to exclude the cost of repeated task evaluations from the total training budget. This underestimates the true computational cost of CURATE compared to baselines. Moreover, for this kind of problem setup, evaluation alone can sometimes be sufficient to train an agent, for example, in Evolution Strategies (ES), where learning is driven entirely by repeated evaluations of policy performance without a separate training phase. This makes the comparison potentially unfair, as CURATE benefits from frequent evaluations but seems not include them in its reported training budget.

W4: The evaluation of the proposed approach requires the definition of a solved threshold, which in most cases means that we already need to know how to solve the problem in order to define it. This requirement limits the range of domains where the approach can be applied.

W5: The testbed consists of only two environments: MiniGrid MultiRoom and the modified-for-curriculum Procgen Leaper. This is a very small set of tasks, and the task spaces themselves are quite limited (e.g., 4 possible rooms in MiniGrid and a two-dimensional task space with 3 road lanes and 3 water lanes for Leaper). Overall, the testbed is too restricted to demonstrate scalability. Moreover, the paper ignores several benchmarks from curriculum learning research, such as those used in the ACCEL paper.

W6: It seems that the approach is not applicable to hierarchical curriculum setups (e.g., MineCraft or Crafter), where preventing catastrophic forgetting is crucial. These environments typically require mechanisms for skill retention or replay, as presented, for example, in PLR. CURATE does not appear to include such mechanisms.

**Questions:**

Q1: How were the baselines tuned or adapted to the tasks being solved?
For example, in PLR replay there are several possible prioritization methods (e.g., policy_entropy, least_confidence, min_margin, gae, value_l1, etc.). Which specific configuration was used in the experiments? Additionally, what does the “PLR prioritization - rank” entry in Table 3 of the appendix refer to?

Q2: Does the evaluation budget include the cost of all curriculum evaluations used? If not, can the authors provide an estimate of how many additional environment steps these evaluations require?

Q3: Can CURATE be adapted to environments with continuous task parameters, where task difficulty changes smoothly (e.g., obstacle density or agent's speed)? How would the Gaussian curriculum policy and REPS update handle such cases?

---

> ### Author Response · Authors · 2025-11-25
> **Our response to your initial review**
>
> We provide our thanks in reviewing our submission. We are happy to hear strong marks for our motivation and the Procgen Curriculum Suite. We agree on both points, especially towards using the Procgen Curriculum Suite to improve benchmarking. We also provide our thanks for your limitations and weaknesses. We view them as opportunities to strengthen the work and improve your review score.
>
> Before we continue, we expect a new version of the paper within 4 days that may clear up some concerns. This new paper version will contain results on other Procgen games besides Leaper.
>
> ## W1: Clarity
>
> We will add figures, such as illustrations of the loss terms and an overview figure, which we hope will address concerns here. We are also looking into studying CURATE from a theoretical perspective for insights we can also add to the paper to help with understanding.
>
> ## W2: Task Space
>
> We acknowledge that CURATE requires that a task space parameterization with axes positively correlated with difficulty is known a priori. However, we suggest that the existence of such a parameterization is not a strong assumption, as a natural difficulty progression may exist in many domains. So, the specific assumption that CURATE requires is that such a parameterization is known at training time. Although this is an assumption that other baselines do not use, if these assumptions hold, we demonstrate that CURATE shows state-of-the-art performance compared to UED/naive curriculum baselines that don’t require human expertise. So, even with this assumption, we believe CURATE is nonetheless an important contribution to the literature. Furthermore, we are currently investigating how this assumption can be relaxed by learning a task space that addresses CURATE’s assumptions. However, this extension is significant and non-trivial, as it entails 1) identifying what the right parameter variations are, and 2) how to order tasks within this space. For this reason, we believe relaxing this assumption would be best served in a follow-up work that can thoroughly investigate how this is learned.
>
> ## W3: Evaluation Procedure
>
> We will address this concern in a separate comment below.
>
> ## W4: Solved Threshold
>
> We suggest that the definition of such a threshold does not necessarily mean it is known _how_ to solve the problem — just that it is known what level of success is considered acceptable. We suggest that this is a much smaller assumption than if it needed to be known how the task is solved. For domains where the reward threshold corresponds to a percentage of solved tasks, this can be quantified in an intuitive way (e.g., solved at least 95% of the time). We also note that for many tasks in RL, the community has agreed what an acceptable solved threshold is, e.g., for CartPole, the agent needs to receive an average of at least 195 over 100 episodes. In this light, CURATE simply integrates this knowledge into the learning process. We also note that we are planning an experiment for the new version of the paper that examines the sensitivity of this threshold.
>
> ## W5: Limited Experimentation
>
> In the next version of the paper, we will have more experiments in the Procgen domain, which we hope allays concerns here. We acknowledge that there are other benchmarks used in the curriculum learning literature, such as those in the ACCEL paper [1]  (MiniHack, cluttered room, BipedalWalker), but we note the use of MiniGrid and Procgen are in distribution (as investigated in the PLR [2], DCD [3], and ACCEL [1] papers). For benchmarks from the ACCEL paper, our initial investigation of the cluttered room domain (Section 4.2 in the ACCEL paper [1]) suggested that the task difficulty counterintuitively decreases with the number of introduced blocks, as this reduces the branching factor for navigating the maze. Thus, a core assumption of CURATE does not hold, and Robust PLR and ACCEL will outperform CURATE in this domain. Our claims for CURATE are only for domains in which the task axes positively correlate with difficulty, which excludes cluttered room. For BipedalWalker, we are currently investigating this domain, although we note it is somewhat different than the other domains. There are dense rewards, and the initial difficulty comes not from the terrain, but from keeping the walker upright. We have tried to sparsify this domain by only providing reward when the walker solves the tasks, but even in the easiest tasks, the walker does not stay upright without the dense reward shaping that is customary in this domain. Thus, we cannot say for certain that CURATE will work in this domain, but we are actively looking into it.
>
> Below, we provide separate comments for your questions.
>
> [1] J. Parker-Holder and M. Jiang et al., “Evolving Curricula with Regret-Based Environment Design”, ICML 2022.
>
> [2] M. Jiang et al., “Prioritized Level Replay”, ICML 2021.
>
> [3] M. Jiang and M. Dennis et al., “Replay-Guided Adversarial Environment Design”, NeurIPS 2021.

---

> > ### Author Response · Authors · 2025-11-25
> > **Response to Question 1**
> >
> > ## Q1: Tuning the Baselines
> >
> > We mention this in Section D.3 in the appendix. Generally, we use the same hyperparameters as specified in the domains used for PLR [1] or ACCEL [2]. We choose these hyperparameters because they were chosen by the authors of these papers as the best hyperparameters for their respective approaches in these domains.
> >
> > The hyperparameters for Robust PLR are the same as those used in the DCD [3] paper for MultiRoom and the same as those used in the PLR paper [1] for Procgen. The term “PLR prioritization - rank” refers to how differences in PLR scores translate to differences in prioritization using the rank of the level score (please see end of Section 3.1 in the PLR paper, for convenience, this is explained as: “We make the design choice of using rank prioritization, for which h(Si) = 1/rank(Si), where rank(Si) is the rank of level score Si among all scores sorted in descending order.”). The specific PLR configuration is shown in Table 3: replay rate is 0.5, prioritization is rank, temperature is 0.3 for MultiRoom and 0.1 for Procgen, staleness coefficient is 0.3 for MultiRoom and 0.1 for Procgen, replay buffer size is 4000, and the scoring loss function is positive value for MultiRoom and L1 value for Procgen.
> >
> > The hyperparameters from ACCEL come from the ACCEL paper [1] for MultiRoom (a MiniGrid domain), except that we performed a hyperparameter search for the number of edits and found that 3 edits worked best. We chose 3 edits also for Procgen as well as this number is also used for other domains, e.g. BipedalWalker.
> >
> > For Incremental, we chose the step size as 1 as it was both the minimum value for these discrete domains and a reasonable choice given the size of the task spaces.
> >
> > For CURATE, we selected the hyperparameters through careful analysis, experimentation, and study of the various tradeoffs of these parameters. Towards its use as an automatic curriculum learning algorithm, CURATE provides automatic calculation of several hyperparameters (see Section B.2) to help with hyperparameter tuning.
> >
> > [1] M. Jiang et al., “Prioritized Level Replay”, ICML 2021.
> >
> > [2] J. Parker-Holder and M. Jiang et al., “Evolving Curricula with Regret-Based Environment Design”, ICML 2022.
> >
> > [3] M. Jiang and M. Dennis et al., “Replay-Guided Adversarial Environment Design”, NeurIPS 2021.

---

> > ### Author Response · Authors · 2025-11-25
> > **Response to Question 2**
> >
> > ## Q2: Evaluation Budget
> >
> > As we mention in our limitations section, we designed CURATE around the assumption that task evaluations are neither limited nor costly. So, our experimental results do not include the cost for evaluations used in the curriculum learning updates. We note that if evaluations are costly, then extensions to CURATE are certainly possible such that the Gaussian distribution can be updated with samples collected from training, rather than separately collecting those from evaluation. In future versions of the paper, we will report the number of evaluation episodes required for CURATE. However, we note that what this translates to in terms of environment steps is very domain specific and highly variable. For domains with a few number of steps, it is likely that this cost will be small relative to the overall number of training steps. For domains with a large number of steps, this may be quite large, so for such domains, we recommend extending CURATE as previously mentioned to use training samples to update the Gaussian distribution. In this way, we believe that CURATE can still be used in cases where evaluations are costly, just with a slightly different approach.

---

> > ### Author Response · Authors · 2025-11-25
> > **Response to Question 3**
> >
> > ## Q3: Continuous Task Parameters
> >
> > As long as the task axes are positively correlated with difficulty, we expect that CURATE should work for continuous task parameters/spaces. For continuous task spaces, we can dispense with the discretization of the Gaussian distribution over the task space, and instead, directly sample from the Gaussian distribution for the training tasks. So, in some sense, it is more straightforward to use CURATE with continuous task spaces, and curriculum progression/REPS behaves similarly as in discrete domains. To this end, we are currently investigating the use of CURATE in a continuous task parameter domain, and if we can obtain these results by the rebuttal, we will include them in the paper. Even if we do not obtain these by the end of the rebuttal, we commit to providing this in the camera ready paper.

---

### Official Review · Reviewer_JQLZ · 2025-10-31

**Soundness:** 2
**Presentation:** 2
**Contribution:** 2
**Rating:** 2
**Confidence:** 5

**Summary:**

The paper introduces CURATE, an automatic curriculum learning algorithm for reinforcement learning agents. CURATE frames curriculum generation as a policy search problem in task space, where a Gaussian curriculum policy selects tasks based on the agent’s current competence. By rewarding performance on “unsolved but easiest” tasks and updating the task distribution using Relative Entropy Policy Search (REPS), the method aims to produce an approximately easiest-to-hardest progression. Experiments on MiniGrid MultiRoom and Procgen Leaper compare CURATE with domain randomization, handcrafted incremental curricula, and automatic approaches such as Robust PLR and ACCEL. Results show that CURATE can learn task-directed curricula and achieve better sample efficiency than the unsupervised baselines.

**Strengths:**

1. Proposes a clear formulation of curriculum policy search based on competence, with explicit loss terms and an implementable algorithm.

2. Includes quantitative comparisons with both non-learning and learning-based curriculum methods.

3. Uses two distinct RL domains (grid-based and image-based) to demonstrate the approach.

4. Provides full pseudocode and implementation details, which supports reproducibility.

**Weaknesses:**

1. Unclear novelty and positioning - The abstract and introduction do not clearly articulate what distinguishes CURATE from prior teacher-student or learning-progress methods (e.g., ALP-GMM, bandit-based curricula).

2. Lack of theoretical grounding - The main insight that “performance on nearby tasks increases inversely with difficulty” is presented empirically, without discussion of the assumptions or limits under which it holds.

3. Potential local minima - Because CURATE samples from tasks the agent already performs well on, it is unclear how it avoids getting stuck in competence plateaus or repetitive easy tasks.

4. Assumption of monotonic difficulty - The claim that “easier tasks yield higher returns” may not generalize.

5. Reward threshold - R_S is critical to defining competence but is only described in the appendix. Its determination, sensitivity, and generality are not discussed in the main text.

6. Key objective terms deferred to appendix – Loss components L_diff and L_dist are central to how CURATE functions but are explained only in Appendix B.1, with minimal intuition in the main body.

7. Limited experimental scope - Only two domains are evaluated, both with low-dimensional, discrete tasks, making it difficult to claim broader applicability.

8. Missing ablations and trajectory comparisons – There is no ablation of loss terms or analysis of the actual task trajectories compared to other methods.

9. Ambiguous statement about “best initial tasks” – The paper claims CURATE finds the “best” initial task set (line 365) without defining what constitutes “best” or whether it is benchmarked against any ground truth.

**Questions:**

1. How is R_S chosen in practice, and how sensitive is performance to this value?

2. What prevents CURATE from remaining in a local optimum of easy tasks that yield consistently high returns?

3. Under what formal assumptions does the “nearby task improvement” insight hold true?

4. Why were the key curriculum objective terms (L_diff, L_dist) placed in the appendix, and how do they affect learning if removed or modified?

5. Can you provide quantitative comparisons of the task trajectories (sequence or spread of sampled tasks) between CURATE and baselines such as ACCEL or PLR⊥?

6. Would the method still perform effectively in task spaces that are not strictly monotonic in difficulty or where difficulty is non-separable across dimensions?

---

> ### Author Response · Authors · 2025-11-25
> **Our response to your initial review**
>
> Thank you for your time in reviewing our submission. We are happy to hear strong marks for our formulation, experimental analysis, and implementation details. We also thank you for your thoughts on weaknesses and limitations. We hope to use them as opportunities to strengthen our work and improve your review score.
>
> Before we address your concerns, we want to share that we expect a new version of the paper within 4 days that may clear up some concerns. Among other improvements, this new version will contain results on other Procgen games besides Leaper, which we hope to address some concerns (e.g., limited experimental scope).
>
> Below are our comments to the identified weaknesses.
>
> ## W1: Unclear Novelty
>
> We believe CURATE’s novelty lies in the use of the reward solved threshold and competence learning objective, which induces an easiest-to-hardest curriculum progression. Our experiment in Appendix E.2 is designed to show this. In future versions of the paper, we will make this novelty more clear.
>
> ## W2: Lack of Theoretical Grounding
>
> We acknowledge that, at present, our study is primarily empirical. However, CURATE works in a principled way, under the assumptions such as 1) task space axes positively correlating with difficulty and 2) adjacent tasks are sufficiently similar, such that learning one task improves performance in adjacent tasks due to transfer learning. These two assumptions are required for the competence learning objective to induce a curriculum, along with the fact that the tasks are solvable. We will update the paper to make these assumptions more clear. We will also pursue theoretical insights and understanding and will add them to the paper when we obtain them.
>
> ## W3: Nearby Tasks
>
> As mentioned in W2, we believe it holds true under 1) task space axes positively correlating with difficulty and 2) adjacent tasks are sufficiently similar. A key question is how task similarity can be quantified. Some ideas include 1) cosine similarity of encoded image observations of the tasks or 2) the local gradient of evaluation reward relative to environment parameters (e.g., if the agent receives X evaluation reward in 2-room mazes, it will receive Y evaluation reward in 3-room mazes and Z evaluation reward in 4-room mazes, where X > Y > Z). We believe quantifying similarity would unlock theoretical insights, which we are interested in exploring and adding to the paper.
>
> ## W4: Monotonic Difficulty
>
> We would suggest that it is not the assumption of a task space of monotonic difficulty that is CURATE’s strongest assumption, rather, it is that CURATE assumes that this task space is known a priori. We believe that for many tasks, a natural difficulty ordering does exist – it’s just that CURATE does assume this is known. We are currently pursuing work in learning the task space, which would address this limitation. However, this extension is significant and nontrivial, as it requires 1) identification of the right axes and 2) ordering tasks within these axes. We believe that the contributions so far with CURATE, although with relatively stronger assumptions, allow for an important contribution towards progress in automatic curriculum learning.
>
> ## W5: Solved Threshold
>
> In the upcoming new version of the paper, we will include an experiment that shows the effect of varying the reward threshold. We will also provide more discussion as well.
>
> ## W6: Objectives
>
> In the upcoming new version of the paper, we will include illustrations for the loss terms and describe them more thoroughly in the main text.
>
> ## W7: Limited Experimental Scope
>
> In the upcoming new version of the paper, we will include results for more Procgen games, which we hope addresses this limitation.
>
> ## W8: No Ablations
>
> We plan to run experiments to show the sensitivity of the learning objectives.
>
> ## W9: Best Tasks
>
> By best tasks, we mean that they are a set of tasks that have the highest curriculum return (e.g., easiest that have not yet been solved). This is not always the easiest tasks: in Leaper, for example, due to stochastic network initialization, many agents will start in tasks with (1 road lane, 0 water lanes), but some may start in tasks with (0 road lanes, 1 water lane). In both cases, tasks with (0 road lanes, 0 water lanes) are trivially easy and already solvable. Thus, it is difficult to quantify ground truth in general for the best starting tasks for multidimensional task spaces, because in principle, different starting conditions could work as it ultimately depends on the initial proficiency of the agent. However, for unidimensional task spaces, this is more straightforward. In these cases, it would be the first set of tasks that are not yet solvable. In MultiRoom, this is the set of tasks with 1 room, which we find that CURATE successfully starts in over 10 trials (the mean of the starting distribution is 1.056).
>
> We address your questions in a separate comments below.

---

> > ### Author Response · Authors · 2025-11-25
> > **Response to Questions 1-3**
> >
> > ## Q1: Reward Solved Threshold
> >
> > Usually, to select $R_S$, we run domain randomization until it converges to find the maximum achievable target evaluation value. Then, we set $R_S$ just under that value. In our forthcoming Procgen experiments, we run all games with terminal rewards only, and use the same threshold (8) for Leaper as for the other games. In that case, we needn’t figure out what is best for each game individually. For domains where $R_S$ represents a percentage of successful episodes, this can simply mean defining what is the acceptable threshold for successfully solving the tasks. We note that the selection of $R_S$ does not necessarily require knowledge of how the tasks are solved, just at what value the obtained return is considered acceptable. Regarding sensitivity, our next version of the paper will include an experiment that illustrates this sensitivity.
> >
> > ## Q2: Local Optima
> >
> > As long as the RL agent can still improve and eventually solve the current task distribution, then CURATE should not get stuck in local optima. If the RL agent can in principle solve any task within the task space given enough time, then we would expect CURATE to induce a curriculum that progresses the agent to eventually solving the target tasks, so we do not expect it to get stuck in easier tasks. However, we have noticed in our forthcoming Procgen experiments that some domains are sufficiently difficult that the agent cannot further improve its return past a certain point. This means that curriculum advancement is no longer possible, and the curriculum stalls out before reaching the target task. While this affects CURATE, it also affects all other baselines equally, including the Incremental baseline. So we do not see this as a particular reflection on CURATE, rather, the difficulty of the tasks.
> >
> > ## Q3: Formal Assumptions for Nearby Task Improvement
> >
> > As mentioned in our response to W2 and W3, we believe these assumptions are that 1) the task space axes positively correlate with difficulty and 2) adjacent tasks are sufficiently similar. We plan to look into a more theoretically rigorous definition of “sufficiently similar” and can update the paper with our findings.

---

> > ### Author Response · Authors · 2025-11-25
> > **Response to Questions 4-6**
> >
> > ## Q4: Loss Terms
> >
> > The curriculum learning objective explanations were placed in the appendix not because they are unimportant, but because we were somewhat limited on space. In our next version of the paper, we will put more discussion in the main paper, including figures. We also plan to run ablation experiments that analyze the effect of these terms.
> >
> > ## Q5: Task Trajectories
> >
> > We have some illustrations of curriculum trajectories in Figure 12 in the appendix. Typically, we find that Robust PLR and ACCEL tend to stall out and have limited progression into the task space as compared to CURATE and Incremental. This is because Robust PLR and ACCEL are optimizing for training return, which tends to favor easier tasks as they naturally have higher return. This is in contrast to CURATE, which uses the competence learning objective that yields no curriculum return for solved tasks, which induces a progression to harder tasks. In Figure 12, we show only one representative trial (the median trial) as the curricula are often multimodal and do not show well to average across trials.
> >
> > ## Q6: Task Spaces that are not Monotonic and Non-Separable Difficulty
> >
> > For task spaces that are not monotonic/separable in difficulty, we observe that CURATE may eventually induce a curriculum that can solve the task, but it likely will not be optimal. In our initial experiments with the cluttered room domain (Section 4.2 in the ACCEL paper [1]), we find that the principal axis of difficulty is actually not the number of blocks placed in the room, which counterintuitively makes the tasks easier by having a smaller branching factor for navigating the room. In this domain, Robust PLR and ACCEL will do better than CURATE. However, one promising direction that we are exploring is the ability to learn a task space parameterization that addresses the assumptions required for CURATE. We would expect that if a monotonic task space can be learned with the right assumptions, CURATE should perform very well, even if the provided task space definition is non-monotonic. Because this extension is non-trivial, we plan to investigate it in a separate paper; we believe the contributions with CURATE make important progress in the area to stand alone.
> >
> > [1] J. Parker-Holder and M. Jiang et al., “Evolving Curricula with Regret-Based Environment Design”, ICML 2022.

---

### Official Review · Reviewer_WGga · 2025-11-01

**Soundness:** 2
**Presentation:** 2
**Contribution:** 1
**Rating:** 2
**Confidence:** 3

**Summary:**

The paper proposes a curriculum learning approach for RL tasks where the curriculum is evaluated based on task difficulty.

**Strengths:**

* The proposed method is intuitive.

**Weaknesses:**

* The framework has a strong assumption that the environments are fully parameterized within space $\Theta$. This raises practical concerns about whether such an assumption holds for generic decision-making tasks. More non-game examples and also clarifications on the limitations of the method would help address the concern.
* Critical values such as initial $\mu_\theta, \Sigma_\theta$ are treated as hyperparameters to be tuned. Either ablations on showing robustness of these hyperparameters or discussions on how to choose them would make the framework more practical for general tasks.
* Results are validated on simulated tasks either with low-dimensional task space, or with human-engineered procedural environment distributions, with oracal access to the procedural parameters to generate the environments being benchmarked. These simulated domains are much simpler than real-world or general decision-making tasks.

**Questions:**

* The environment distribution seems to be a Gaussian with mean $\mu_\theta$ and covariance $\Sigma_\theta$, although I didn't find it clearly defined in the manuscript (but it's possible I missed it).

---

> ### Author Response · Authors · 2025-11-25
> **Our response to your initial review (1/2)**
>
> Thank you for your time to review our submission. We are also happy to hear that you felt that CURATE was intuitive. We also thank you for sharing your weaknesses and limitations, and we hope to use these as opportunities to strengthen our work and improve your review score.
>
> Before we address your limitations and questions, we want to share that we expect a new version of the paper within 4 days that may clear up some concerns. Among other improvements, this new version will contain results on other Procgen games besides Leaper, which we hope to address concerns (e.g., with limited experimentation or clarity).
>
> Below are our comments.
>
> ## W1: Fully Parameterized Environments
>
> We recognize that CURATE requires certain assumptions of the task space. However, we note the existence of underlying environment parameters has been previously used for the UED framework [1] (e.g., ACCEL [2]) and ALP-GMM [3] frameworks. Moreover, for other non-game settings, such as simulators, results depend on the value of specific parameters, so we do feel the existence of such parameters is not a strong assumption. Therefore, the narrower assumption that CURATE uses is just that 1) the axes of the task space positively correlate with task difficulty and 2) these axes are known to the agent. We would suggest that the first assumption, the existence of such a task space, is not a strong assumption, as many tasks have some natural progression of difficulty. This is true across many domains more broadly than RL, such as the effective design of curricula for human education and effective engagement in interactive digital media, such as video games. Thus, we suggest that it is specifically the second assumption, that these axes are known, to be the specific narrower assumption that CURATE uses. While this may not be known for all domains, in domains which it is known, our results suggest CURATE provides state-of-the-art performance over UED/naive curriculum baselines that don’t require human expertise. Furthermore, we are currently pursuing follow-up work to learn this task space, thereby addressing this assumption. However, the extension to learn the task space is non-trivial and itself a significant contribution, because 1) the axes need to be identified and learned and 2) tasks need to be properly ordered within these axes. We feel that under the curriculum assumptions of CURATE, we have made important progress towards automatic curriculum learning, while providing an opportunity to study relaxing these assumptions in subsequent work.
>
> Additionally, we note that the use of environments that are “game-like” are common in the UED/curriculum learning literature with RL agents. For example, UED [1] uses MiniGrid exclusively; ACCEL [2] uses MiniHack, MiniGrid, and BipedalWalker; ALP-GMM [3] uses versions of BipedalWalker; and PLR [4] uses MiniGrid and Procgen. So, we feel that our chosen domains are in distribution to what is commonly used in the literature. Our focus is on the use of RL agents, not sequential decision-making agents in general. If there is a specific example of an environment that you feel would be useful to show CURATE for, please let us know and we will look into the viability of CURATE.
>
> ## W2: Hyperparameters
>
> As noted in Section 4.3 and Appendix B.2, many of the hyperparameters for CURATE are automatically calculated based on the task space definition $\Theta$. Therefore, we do not need specific tuning for many hyperparameters. For example, in our forthcoming Procgen experiments, we use the same automatic hyperparameter calculation procedure and the same hyperparameters otherwise. Specifically, the hyperparameters that you mention, the initial values of $\mu_\theta$ and $\Sigma_\theta$, are in fact automatically inferred and do not need to be tuned (see Appendix B.2, point 1). CURATE uses an optimization procedure that finds the Gaussian distribution that most optimally approximates a uniform distribution. Furthermore, we would suggest that most algorithms in the UED/curriculum learning literature require tuning of hyperparameters, e.g., ACCEL [2] requires the number and frequency of edits to be specified. In this light, CURATE distinguishes itself from other methods, in that some of our hyperparameters are automatically calculated. Per your suggestion, in our new version of the paper, we will also include a discussion of how to tune the hyperparameters that are not automatically calculated.
>
> [1] M. Dennis and N. Jaques et al., “Emergent Complexity and Zero-shot Transfer via Unsupervised Environment Design”, NeurIPS 2020.
>
> [2] J. Parker-Holder and M. Jiang et al., “Evolving Curricula with Regret-Based Environment Design”, ICML 2022.
>
> [3] R. Portelas et al., “Teacher Algorithms for Curriculum Learning of Deep RL in Continuously Parameterized Environments”, CoRL 2020.
>
> [4] M. Jiang et al., “Prioritized Level Replay”, ICML 2021.

---

> > ### Author Response · Authors · 2025-11-25
> > **Our response to your initial review (2/2)**
> >
> > ## W3: Experiments
> >
> > As mentioned in our response to W1, the use of “game-like” domains is common and in distribution for literature in the UED/curriculum learning space. Thus, we believe this criticism is not specific to CURATE, but rather the field as a whole, and we ask to be evaluated based on other literature within this area. Evaluating in the real world is also significantly more costly and expensive and requires access to compatible hardware, and progress towards automatic curriculum learning with CURATE is still achievable with the domains we have chosen. Moreover, in our upcoming new version of the paper, we will have much more extensive experiments for other Procgen games.
> >
> > ## Q1: Environment Distribution
> >
> > You are correct, our representation for the environment distribution used by CURATE is a Gaussian distribution. This distribution is what is being learned by $\pi_c$.

---

### Official Review · Reviewer_sBQV · 2025-11-01

**Soundness:** 3
**Presentation:** 3
**Contribution:** 3
**Rating:** 6
**Confidence:** 4

**Summary:**

This paper presents CURATE, an automatic curriculum learning algorithm for reinforcement learning. CURATE conducts policy search in task space, dynamically adapting the curriculum based on the agent’s competence to progress from easier to harder tasks. Evaluated on MiniGrid MultiRoom and the Procgen Leaper environments, CURATE achieves higher sample efficiency than other baselines.

**Strengths:**

1. CURATE is conceptually well-motivated, formulating curriculum design as a policy search over task distributions guided by agent competence.
2. The paper provides a careful qualitative and quantitative examination of the learned curricula, including how the agent starts with easier tasks, maintains narrow and focused task distributions, and progresses in a generally easiest-to-hardest trajectory. The visualizations  effectively illustrate the evolution and path of the curriculum in the task space.
3. The paper introduces the Procgen Curriculum Suite, a new environment designed to evaluate curriculum learning. Given the lack of effective tools for rapid assessment of curriculum methods on parameterized tasks, this contribution appears promising.

**Weaknesses:**

1. A key limitation of CURATE is its reliance on a fully defined task space with clear difficulty gradients, limiting its use in many RL environments that lack this or need significant effort to specify. This limitation is noted but not fully explored (see Questions 1, 2).
2. The baseline selection is not entirely fair. ACCEL and PLR are designed for generalization across many tasks (retaining and revisiting previously learned tasks throughout training) and lack prior knowledge of task difficulty until attempting to learn each task. These constraints inherently limit their sample efficiency compared to CURATE, which assumes difficulty is known a priori. A more equitable comparison would be against TSCL, which evaluates performance across all tasks and is designed for environments with a small number of tasks. This would balance CURATE's advantage of having prior difficulty information against TSCL's ability to evaluate all tasks .
3. CURATE is only evaluated on two relatively simple environments with few parameters, which were specifically selected for this method. It would be more informative to test CURATE on widely used benchmarks for curriculum learning, such as BipedalWalker, which includes multiple configurable parameters and has been used to assess POET and ACCEL. Broader evaluation would better demonstrate CURATE’s generalizability and robustness.
4. The description of the method lacks clarity and is difficult to understand on the first reading. As a result, readers often need to revisit the pseudocode for better understanding. Including a visual diagram of the method would make the section much clearer and improve the overall presentation.

References
- [ACCEL] Parker-Holder, J., Jiang, M., Dennis, M., Samvelyan, M., Foerster, J., Grefenstette, E. and Rocktäschel, T., 2022, June. Evolving curricula with regret-based environment design. In International Conference on Machine Learning (pp. 17473-17498). PMLR.
- [PLR] Jiang, M., Grefenstette, E. and Rocktäschel, T., 2021, July. Prioritized level replay. In International Conference on Machine Learning (pp. 4940-4950). PMLR.
- [ TSCL ] Tambet Matiisen, Avital Oliver, Taco Cohen, and John Schulman. Teacher-Student Curriculum Learning. IEEE Transactions on Neural Networks and Learning Systems, 31(9):3732–3740, 2019.
- [ POET ] Wang, R., Lehman, J., Clune, J. and Stanley, K.O., 2019, July. Poet: open-ended coevolution of environments and their optimized solutions. In _Proceedings of the genetic and evolutionary computation conference_ (pp. 142-151).

**Questions:**

Questions on task parametrization
1. How could the constraint linking task complexity to the parameter value be addressed?
2. If the principal axes are not perfectly disentangled, how robust is CURATE? Does it degrade gracefully or collapse?

Questions on environments

3. What criteria were used to select the parameters for task configuration in Procgen?
4. Have you experimented with the other curriculum-based version of Procgen (C-Procgen)? In what key aspects does  Procgen Curriculum Suite differ from theirs?
5. Why were the experiments not performed in a standard environment where other curricula were evaluated? Is this decision connected to the method’s main limitation concerning task parametrization?

Questions on curriculum performance

6. If the environment has a large task space, does the model forget simpler tasks over time since the method lacks a built-in mechanism to revisit them?

References
- [C-Procgen] Tan, Z., Wang, K., Wang, X., 2023. C-Procgen: Empowering Procgen with Controllable Contexts. [https://doi.org/10.48550/arXiv.2311.07312](https://doi.org/10.48550/arXiv.2311.07312)

---

> ### Author Response · Authors · 2025-11-25
> **Our response to your initial review (addressing weaknesses)**
>
> Thank you for your time to review our submission. We are also happy to hear that you felt that CURATE had strengths in being well-motivated, there was effective discussion and visualization, and that the Procgen Curriculum Suite would be an effective contribution to benchmarking curriculum learning. We agree on all of these points. We also thank you for sharing your weaknesses and limitations, and we hope to use these as opportunities to strengthen our work. We are happy to hear that you currently feel CURATE is above the acceptance threshold, and we are open to using your feedback to further improve your score.
>
> Before we proceed, we expect to upload a new version of the paper within 4 days that may clear up some concerns. Among other improvements, this new version will contain results on other Procgen games besides Leaper, which we hope to address some concerns.
>
> Below are our comments to the identified weaknesses.
>
> ## W1: Task Space
> We recognize that CURATE requires certain assumptions of the task space. However, we note the existence of underlying environment parameters has been previously used for the UED framework [1] (e.g., ACCEL [2]) and ALP-GMM [3] frameworks. Therefore, the narrower assumption that CURATE uses is just that 1) the axes of the task space positively correlate with task difficulty and 2) these axes are known to the agent. We would suggest that the first assumption, the existence of such a task space, is not a strong assumption, as many tasks have some natural progression of difficulty. We recognize then it is specifically it is the second assumption, that these axes are known, to be the specific narrower assumption that CURATE uses. While this may not be known for all domains, in domains which it is known, our results suggest CURATE provides state-of-the-art performance over UED/naive curriculum baselines that don’t require human expertise. Furthermore, we are currently pursuing follow-up work to learn this task space (see our response to Q1).
>
> ## W2: Baselines
> Thank you for this recommendation, and we agree that evaluating against TSCL [4] would be a good baseline for MultiRoom. Evaluating in all tasks should be possible given that the number of tasks is small in this domain. For the Procgen domain, while Leaper has 16 tasks, other games have much larger spaces (e.g., our defined task space for Climber has 210 tasks [[1,10] platforms and [0,20] enemy spawn probability]), which makes exhaustive evaluation less tractable. Therefore, we propose using the Graves algorithm [5] instead of TSCL for Procgen, which is similar in some respects to TSCL but may be more amenable to large task spaces since it only imposes a probability distribution over tasks via a multi-armed bandit (1 arm == 1 task), not that the tasks necessarily be exhaustively evaluated. However, we believe that in larger tasks spaces, Graves will not work as well as CURATE, because exploration for Graves requires random sampling uniformly over tasks (which becomes sample efficient as the number of tasks increases), whereas CURATE is able to learn an effective starting distribution initially to get the agent learning quickly. We will try to add these baselines during the rebuttal, but we cannot say for certain if we can with the limited time. However, we do commit to adding 1) TSCL for MultiRoom and 2) Graves for Procgen in the camera-ready version.
>
> ## W3: Evaluation Breadth
> Our upcoming version of the paper will have results for the rest of the Procgen Curriculum Suite, which we hope addresses some concerns here. We are also looking into the possibility of evaluating against BipedalWalker, but we note there are some differences between that domain and the domains we have studied. BipedalWalker is a dense reward domain, and much of the initial difficulty comes not from the terrain, but from keeping the walker upright. Moreover, we looked into sparsifying the reward as in MultiRoom and Procgen, but the agent receives no reward at all due to the difficulty. We are currently looking into this and will share updates as we have them.
>
> ## W4: Clarity:
> Thank you for this concern and for sharing the recommendation for figures. For the next version of the paper, we will add a figure that provides an overview of the method as well as a figure that illustrates the loss terms.
>
> Our responses to your questions are posted in individual comments below.
>
> [1] M. Dennis and N. Jaques et al., “Emergent Complexity and Zero-shot Transfer via Unsupervised Environment Design”, NeurIPS 2020.
>
> [2] J. Parker-Holder and M. Jiang et al., “Evolving Curricula with Regret-Based Environment Design”, ICML 2022.
>
> [3] R. Portelas et al., “Teacher Algorithms for Curriculum Learning of Deep RL in Continuously Parameterized Environments”, CoRL 2020.
>
> [4] T. Matiisen et al.,  “Teacher–Student Curriculum Learning”, IEEE Transactions on Neural Networks and Learning Systems, 2020.
>
> [5] A. Graves et al. “Automated Curriculum Learning for Neural Networks”, ICML 2017.

---

> > ### Author Response · Authors · 2025-11-25
> > **Response to Task Parameterization Questions (Question 1 and Question 2)**
> >
> > ## Q1: Addressing Complexity-Value Constraint
> >
> > This is an important area of future research that we are currently pursuing. The existence of an underlying parameterization of the task spaces is not itself novel (e.g., a parameterized task space exists in the frameworks of the UED [1] and ALP-GMM [2] papers), but CURATE does require the task space to have certain properties above those specified by the task space definition used in ALP-GMM: that 1) the axes positively correlate with difficulty and 2) the axes are known to the agent. We believe there is promise in learning a latent representation for the task space to address both these assumptions, and within this latent representation, the task ordering could be established without direct knowledge of the environment parameters. However, because this extension is significant and itself non-trivial to learn, we believe it is best to devote investigation of that study in a separate, follow-up paper.
> >
> > ## Q2: Axes Disentanglement
> > Regarding the principal axes, we refer to “disentangled” to mean that the principal axes are orthogonal and aligned with the major factors of variation. The representation of the curriculum policy for CURATE is a multivariate Gaussian distribution, so for the policy side, the principal axes just need to be linearly independent (although they must still positively correlate with task difficulty for the curriculum progression to work). So, we believe the behavior of CURATE for non-disentangled, positively-difficulty-correlated axes comes down to how REPS [3] works in these settings. The REPS paper does not appear to indicate any particular assumptions other than the standard used for reinforcement learning, e.g., usually that elements such as state vectors are themselves disentangled. We currently believe that if a failure exists, it would be a graceful degradation correlating to the degree to which the principal axes are not orthogonal. If there is more interest in this thread, please let us know and we can continue to investigate. We can also look into running an experiment to show this empirically as well.
> >
> > [1] M. Dennis and N. Jaques et al., “Emergent Complexity and Zero-shot Transfer via Unsupervised Environment Design”, NeurIPS 2020.
> >
> > [2] R. Portelas et al., “Teacher Algorithms for Curriculum Learning of Deep RL in Continuously Parameterized Environments”, CoRL 2020.
> >
> > [3] J. Peters et al., “Relative Entropy Policy Search“, AAAI, 2010.

---

> > ### Author Response · Authors · 2025-11-25
> > **Response to Environment Questions (Questions 3, 4, and 5)**
> >
> > ## Q3: Criteria for Procgen
> > We will explain this more thoroughly in the next version of the paper. In short, we examined each of the 16 Procgen games to study the major factors used in procedural generation that affect difficulty, and usually we constructed the task spaces using those factors. In some cases, we introduced new sources of variation to realize a task space. An example of this is BigFish, which did not have a clear difficulty variation normally, so we introduced an axis to control the number of fish needed to eat to solve the level. Normally this is a fixed value of 30, but we introduce an axis here from [1,30] to introduce the task space. In cases where multiple axes are possible, we also chose axes in such a way to have a diversity of games with differing task dimensionality throughout the Procgen Curriculum Suite. In total, our games have dimensionality ranging from 1-dimensional to 4-dimensional.
> >
> > ## Q4: C-Procgen
> > Thank you for bringing C-Procgen [1] to our attention. This seems like a very useful line of work, and we will cite this in our paper. Intuitively, the Procgen Curriculum Suite is related in many respects to C-Procgen, by exposing certain factors of the procedural generation to be set. We also observe that one feature of the Procgen Curriculum Suite, the possibility of only allowing terminal rewards (thereby sparsifying the games), also is possible with C-Procgen. Therefore, we suggest that the major difference is the specific use of the Procgen Curriculum Suite for curriculum learning by proposing certain parameterized axes to be used for benchmarking. Although the related work [2] investigates curriculum learning experiments, they only propose new, expanded contexts for one game (Leaper) (Sec. 5.2 [2]), which we note has the same parameterization as ours. Given the breadth of possible parameter adjustments with C-Procgen, it would not be clear which ones in particular should be varied for curriculum learning for games beyond Leaper. Therefore, the Procgen Curriculum Suite addresses this by introducing the definition of which axes should be used for benchmarking across all games. That being said, while the goals of C-Procgen and the Procgen Curriculum Suite overlap, they are also distinct in other regards, for example, C-Procgen exposes many more factors of variation. So, we believe there is room in the literature for both works. The Procgen Curriculum Suite can be used to benchmark an approach for certain defined task spaces, whereas curriculum learning works that focus on learning the task space (as in our current ongoing extension work to CURATE) may use both the Procgen Curriculum Suite or C-Procgen for this. So, we believe both are important contributions that can both improve curriculum learning in different, complementary ways.
> >
> > ## Q5: Experimentation for other environments
> > We note that in our upcoming version of the paper, we will have more Procgen experiments, which we hope allays some concerns here. You are correct in that other domains within the curriculum learning literature are used. We are currently investigating the application of CURATE for continuous control domains, such as BipedalWalker. We note that BipedalWalker is slightly different from our other domains insofar that the rewards are dense, and the initial difficulty comes not from the terrain, but keeping the walker upright, which is a problem regardless of terrain. Our initial investigations suggested that sparsifying the reward like our other domains means that no learning progress is made at all due to the difficulty of keeping the walker upright without reward shaping. However, we will continue to investigate this to understand the feasibility of CURATE in this domain.
> >
> > [1] Z. Tan and K. Wang et al., “C-Procgen: Empowering Procgen with Controllable Contexts”, arXiv:2311.07312, 2023.
> >
> > [2] Z. Tan and K. Wang, et al., “Implicit Curriculum in Procgen Made Explicit”, NeurIPS 2024.

---

> > ### Author Response · Authors · 2025-11-25
> > **Response to Question on Curriculum Performance (Question 6)**
> >
> > ## Q6: Large task spaces and forgetting
> >
> > For certain domains, forgetting becomes more of a concern, as it is still the case that forgetting remains an open problem in machine learning. For smaller domains, like MultiRoom and Leaper, we rarely see evidence of forgetting when using CURATE, in part because of the ability to preserve some diversity through maintaining a temporally-varying distribution of tasks. However, if forgetting is an issue for certain domains, CURATE is perfectly compatible with integrating strategies for off-curriculum sampling to mitigate forgetting. For example, we have run experiments for Leaper to investigate the use of say, 25% of the time, sample from other tasks (e.g., all tasks or previously solved tasks), leaving the rest (75%) to be sampled from tasks from the CURATE distribution. This will prevent forgetting, but our initial investigation suggested this reduced sample efficiency, possibly because a non-trivial amount of training budget is being allocated to simply not forget, rather than advance the curriculum. In the next version of the paper, we will explain that CURATE is compatible with such off-curriculum strategies, and we will also look into adding an appendix experiment.

---

### Author Response · Authors · 2025-12-03
**Our closing comments (1/3)**

Dear Area Chair,

We thank you for your time and effort in reviewing our paper and discussion. With the ICLR rebuttal coming to a close, we wanted to update you on the latest of the paper.

We want to thank our reviewers again for their helpful reviews, including identifying areas of improvement for CURATE. We also appreciate that several aspects of the paper were considered strengths, such as CURATE being conceptually well-motivated and intuitive, the potential impact of the Procgen Curriculum Suite to help with benchmarking progress in curriculum learning, and that we are studying an important and difficult problem.

Following our initial reviews, we have **significantly strengthened** our paper by specifically addressing common reviewer concerns. Please see the revised paper that we have uploaded. We summarize the improvements below and map them to reviewer concerns.

## Experimentation

_Initially:_ We demonstrate CURATE’s performance in two main experiments across two domains: MultiRoom and the Leaper game of Procgen. All five reviewers felt this set of experiments was limited. For example, Reviewer YzWB said that, "The testbed consists of only two environments… This is a very small set of tasks, and the task spaces themselves are quite limited… Overall, the testbed is too restricted to demonstrate scalability. Moreover, the paper ignores several benchmarks from curriculum learning research, such as those used in the ACCEL paper."

_Revised paper:_ We **significantly expand our experimentation**, including three domains: 1) MultiRoom (with more seeds, and we found a configuration where CURATE matches Incremental), 2) nearly all of the Procgen Curriculum Suite (14/16 games, with 2 games not chosen because they were too difficult for any method), and 3) BipedalWalker. In doing so, we address concerns regarding experimentation scope. We also add new ablation and follow-up experiments, which are detailed in a separate comment below.
- With our new experiments, we show CURATE outperforms prior methods for **both discrete and continuous task parameterization spaces** and a **range of task space dimensionality**, from 1D (MultiRoom), 1D - 4D (Procgen), and 8D (BipedalWalker).
- Our results on BipedalWalker show that we can match or exceed all baselines in this area. Moreover, our experimentation on BipedalWalker addresses concerns about not choosing more domains used in the curriculum learning literature, like BipedalWalker. We also note that BipedalWalker was specifically mentioned by Reviewer sBQV as well as Reviewer YzWB mentioning using a domain from the ACCEL paper, which primarily features BipedalWalker.

## Scope of applicability

_Initially:_ We provide evidence for training an RL agent to complete a difficult, target task distribution. Some reviewers felt this applicability was too narrow, such as Reviewer sBQV saying that, "Broader evaluation would better demonstrate CURATE’s generalizability and robustness."

_Revised paper:_ We **broaden CURATE’s applicability** to other use cases. Specifically, in our BipedalWalker experiments, we demonstrate that CURATE can effectively train the agent to complete multiple tasks of interest within a broad task space. In this experiment, we also demonstrate how CURATE is perfectly compatible with strategies to prevent forgetting, a concern raised by some reviewers.

---

> ### Author Response · Authors · 2025-12-03
> **Our closing comments (2/3)**
>
> ## Ablations and Understanding
>
> _Initially:_ Reviewers raised concerns that there were no ablations or rigorous study of R_S, the task solved threshold. For example, Reviewer X7fM said that, “There is no ablation for the impact of different reward components for training the curriculum policy,” and Reviewer JQLZ wrote that “Reward threshold - R_S is critical to defining competence but is only described in the appendix.”
>
> _Revised paper:_ We present **greater understanding** of CURATE through new appendix experiments: a learning objective ablation, and studies for sensitivity to the task solved threshold.
> - For our ablation experiment, we demonstrate the necessity of the orthogonal distance loss term in an experiment with Leaper (hard mode).
> - We conduct sensitivity experiments by varying the task solved threshold R_S for MultiRoom and Procgen Ninja. We find that, in some domains like MultiRoom, lowering the task solved threshold for intermediate tasks improves performance, whereas in other domains, like Ninja, a fixed threshold is better. We also find that generally, the ordering of baselines is generally insensitive to the choice of the threshold. In other words, CURATE is insensitive to the precise choice of threshold and thus is effective across a range of values, addressing concerns that this term needed to be set carefully.
>
> ## Clarity
>
> _Initially:_ Reviewers raised concerns about the clarity, such as description of loss terms. Reviewer JQLZ asked, “Why were the key curriculum objective terms (L_diff, L_dist) placed in the appendix,” and Reviewer sBQV recommended including a visual diagram.
>
> _Revised paper:_ We have provided greater clarity of the approach. This includes adding new illustrations for the learning objective terms.
>
> Due to the OpenReview information leak, we completely understand the need to end reviewer engagement early and keep initial scores. Because the reviewer engagement period ended before we could upload a revised paper, unfortunately, we missed out on the opportunity for our reviewers to share whether the revised paper addressed their concerns and improve their initial score accordingly. Nonetheless, we have used the opportunity to make our revised paper **significantly stronger** by addressing many concerns shared across reviewers. We cannot say for certain if the reviewers would have changed their scores, but we believe that many of the concerns, whether in whole or in part, would have been addressed by the new revision.
>
> The last concern that was raised was the assumptions on CURATE’s task space. We address this directly in a separate comment below. For all other concerns, we are happy to look into them and continue to strengthen the paper in time for the camera-ready version.

---

> > ### Author Response · Authors · 2025-12-03
> > **Our closing comments (3/3)**
> >
> > ## CURATE’s Task Space Assumptions Lead to Impressive Performance
> >
> > We acknowledge that most reviewers raised concerns regarding the assumptions that CURATE makes on the task space. However, we suggest that these assumptions facilitate CURATE’s **strong performance that match or exceed prior methods**, so our approach nonetheless would be an important contribution to the literature. Furthermore, the strength of the assumptions may be less than what they initially seem.
> >
> > Specifically, CURATE requires that 1) the task space is parameterized by environment variables; 2) the principal axes of the task space positively correlate with difficulty; and 3) the principal axes of the task space are known to the agent. Assumption 1 is a core assumption of work in this area, e.g., the UED framework and ALP-GMM. We suggest that Assumption 2 holds for many domains, in the sense that there exists some task space parameterization with this property. Therefore, we suggest it is the combination of Assumptions 2 and 3 – that the task space is difficulty-aligned and is available to the agent – that is the strongest, as other algorithms do not use this. However, we note that, in principle, privileged information is already used for other algorithms that are important to the field: consider that ACCEL requires that the initial starting task distribution is given.
> >
> > We also state our assumptions in the paper, and we do not claim that CURATE will work outside these assumptions.
> >
> > We agree with the reviewers that addressing these assumptions would strengthen the work, and indeed, we are currently investigating this. However, this extension is significant and non-trivial. First, the task space must be learned from data, which is itself a notable contribution that may require its own in-depth study. Second, tasks must be ordered from data by difficulty, and to do this, difficulty must be properly quantified, then learned. In light of these reasons, we believe it is best to devote investigation to address CURATE’s task space assumptions in a separate, follow-up paper, building from the foundation we set with CURATE.

---

### Meta-Review · Area_Chair_mQFa · 2025-12-21

**Summary:**

Reviewers raised concerns regarding (1) the lack of experiments in diverse environments, (2) the difficulty of understanding the method on first reading, and (3) the strength of the underlying assumptions. The authors addressed the first concern by adding experiments in additional environments. However, the latter two concerns remain largely unresolved.

Despite the revisions, the method still requires substantial back-and-forth to understand, suggesting that clarity and presentation remain an issue. More importantly, reviewers pointed out that the assumption that the task space can be disentangled into parameters that positively correlate with task difficulty, together with the availability of expert knowledge about this task space (e.g., knowing task-solving thresholds a priori), is strong and provides an advantage over the baselines. Notably, under these assumptions, the hand-designed incremental curriculum (IC) method is sufficient to achieve strong performance. The paper does not provide convincing evidence that CURATE offers a clear advantage over this oracle baseline.

Overall, while the paper shows potential, these issues significantly weaken the contribution in its current form. The work would benefit from improved clarity and a relaxation or more thorough justification of its assumptions.

**Reviewer Concerns:**

### Addressed
* **Lack of experiments**.
The authors provided additional experiments in more diverse environments, including BipedalWalker, which has been used in prior work and also covers the continuous-control setting. These additions have addressed concerns about evaluation breadth.

### Outstanding
* **Strong assumptions about the task space**. Reviewers consistently noted that these assumptions are strong and provide highly privileged information to the agent. The rebuttal does not sufficiently address how CURATE would perform when these assumptions do not hold.
* **Clarity and presentation**. While the authors added new illustrations to explain the learning objective, the figures remain difficult to interpret and are accompanied by limited explanatory captions. As noted by Reviewer sBQV, readers often need to repeatedly revisit the pseudocode to understand the method, and this issue persists in the revised version.

**Reviewer Scores:**

* Reviewer sBQV (initial score: 6). This reviewer raised concerns about experimental coverage and clarity. The additional experiments likely address part of the evaluation concern, but clarity issues remain. I would expect the reviewer to maintain their original positive score.

* Reviewer WGga (initial score: 2). Although some concerns were partially addressed in the rebuttal, the core issue regarding the strength of the assumptions remains unresolved. I would expect a modest increase in score after discussion, but not a reversal of the overall assessment.

* Reviewer JQLZ, Reviewer YzWB, Reviewer X7fM (initial scores: 2, 2, 2). These reviewers expressed strong concerns about both clarity and the underlying assumptions. While the rebuttal provides some additional experiments, these fundamental issues remain. Given the reviewers' high confidence levels (5, 4, and 5), I would not expect a meaningful score increase following the rebuttal.

---

### Decision · Program_Chairs · 2026-01-26

Reject